# Phase separation properties of RPA combine high-affinity ssDNA binding with dynamic condensate functions at telomeres

Vincent Spegg[1], Andreas Panagopoulos ®[1], Merula Stout ®[1], Aswini Krishnan[1], Giordano Reginato[2,3], Ralph Imhof[1], Bernd Roschitzki ®[4], Petr Cejka[2,3] & Matthias Altmeyer ®[1]✉

RPA has been shown to protect single-stranded DNA (ssDNA) intermediates from instability and breakage. RPA binds ssDNA with sub-nanomolar affinity, yet dynamic turnover is required for downstream ssDNA transactions. How ultrahigh-affinity binding and dynamic turnover are achieved simultaneously is not well understood. Here we reveal that RPA has a strong propensity to assemble into dynamic condensates. In solution, purified RPA phase separates into liquid droplets with fusion and surface wetting behavior. Phase separation is stimulated by sub-stoichiometric amounts of ssDNA, but not RNA or double-stranded DNA, and ssDNA gets selectively enriched in RPA condensates. We find the RPA2 subunit required for condensation and multi-site phosphorylation of the RPA2 N-terminal intrinsically disordered region to regulate RPA self-interaction. Functionally, quantitative proximity proteomics links RPA condensation to telomere clustering and integrity in cancer cells. Collectively, our results suggest that RPA-coated ssDNA is contained in dynamic RPA condensates whose properties are important for genome organization and stability.

The DNA structure, with nucleotide complementarity as the basis of the double helix, confers chemical stability to the genome. At the same time, it comprises a backup copy of the nucleotide base sequence, conferring information stability through redundancy. Single-stranded DNA (ssDNA) lacks these features and is inherently fragile. It is prone to form secondary structures, such as hairpins and G-quadruplexes, it is more easily attacked by nucleases, and nucleotide removal or base damage can lead to irreversible loss of sequence information and permanently alter the genome.

ssDNA occurs as an intermediate of DNA-repair processes, for example during DNA double-strand break (DSB) repair by homologous recombination (HR), nucleotide excision repair (NER) of lesions induced by ultraviolet (UV) light, and during base-excision repair (BER)

of oxidized bases[1]. Furthermore, short stretches of ssDNA are exposed when DNA is unwound during transcription and DNA replication[2]. Under conditions of replication stress, a hallmark of cancer, levels of ssDNA are elevated, which can exhaust replication capacity and cause massive DNA damage[3,4].

In eukaryotic cells, replication protein A (RPA) is the main factor that binds and protects ssDNA[5]. RPA is a conserved heterotrimeric protein complex composed of the 70-kDa subunit RPA1 (RPA70), the 32-kDa subunit RPA2 (RPA32), and the 14-kDa subunit RPA3 (RPA14). Multiple oligonucleotide binding folds (OB folds) in RPA confer sub-nanomolar affinity to ssDNA, with a reported dissociation constant ($K_D$) as low as $1 \times 10^{-10}$ to $1 \times 10^{-11}$ M[6,7]. Owing to its function of binding and protecting short and long stretches of ssDNA in the eukaryotic cell nucleus,

[1]Department of Molecular Mechanisms of Disease, University of Zurich (UZH), Zurich, Switzerland. [2]Institute for Research in Biomedicine, Faculty of Biomedical Sciences, Università della Svizzera italiana (USI), Bellinzona, Switzerland. [3]Department of Biology, Institute of Biochemistry, Eidgenössische Technische Hochschule (ETH), Zurich, Switzerland. [4]Functional Genomics Center Zurich, University of Zurich (UZH) & Eidgenössische Technische Hochschule (ETH), Zurich, Switzerland. ✉e-mail: matthias.altmeyer@uzh.ch

irrespective of their sequence and genomic location, and owing to its universal role in DNA metabolism, RPA is essential for cell survival. Non-lethal RPA mutations result in DNA-repair defects and genome instability, and RPA haploinsufficiency causes greatly enhanced tumor formation and shortened lifespan[8].

One RPA heterotrimer binds to approximately 30 nucleotides of ssDNA with ultrahigh affinity[2,6,9]. RPA coating of ssDNA has been considered to resemble stoichiometric binding, like tightly packed 'beads on a string.' However, individual RPA domains dynamically exchange on ssDNA, and diffusion-driven sliding of RPA along ssDNA has been observed[10–12]. RPA, when bound to ssDNA, also functions as an interaction and activation platform for a variety of signaling molecules and genome caretakers, including the replication stress response factors ATRIP–ATR, ETAA1, and PRIMPOL[2]. In the context of HR, the tumor suppressor BRCA2 is needed to displace RPA from ssDNA in exchange for RAD51. RPA not only has up to $1 \times 10^5$-fold higher affinity to ssDNA than RAD51, but it is also highly abundant: there are approximately 3–5 million molecules per cell[13], equivalent to a concentration in the 10 μM range in the nucleoplasm, and its local concentration is likely even higher when it is bound to ssDNA. Considering RPA's high abundance and high affinity for ssDNA, how ssDNA is handed off from RPA to different downstream effectors remains poorly understood.

In bacteria, the essential ssDNA-binding protein SSB has recently been reported to phase separate in vitro and in bacterial extracts[14]. In yeast, however, the ssDNA-binding protein Rfa1 showed less dynamic behavior and tighter ssDNA binding than did Rad52, for which condensation behavior consistent with phase separation had been proposed[15,16]. Here, we reveal that mammalian RPA has a strong intrinsic propensity to form highly dynamic condensates both in vitro and in the nucleus of living cells. In vitro, phase separation of the purified RPA trimer is specifically enhanced by ssDNA, which in turn gets selectively enriched in RPA droplets. We show that the RPA subunit RPA2 is critical for phase separation and, using structure prediction and systematic site-directed mutagenesis in conjunction with light-inducible RPA optoDroplet formation, reveal that the intrinsically disordered amino terminus of RPA2 regulates condensation properties. This disordered sequence stretch of RPA2 is subjected to multi-site phosphorylation, and mutations that mimic RPA hyper-phosphorylation disrupt RPA phase separation in vitro and abolish cellular RPA optoDroplet formation. Finally, using label-free quantitative proteomics, we identify heterotypic interactions that respond to intracellular RPA condensation and link RPA's clustering capacity to telomere maintenance by alternative lengthening of telomeres (ALT) in cancer.

## Results

### RPA forms optoDroplets and dynamic DNA-repair condensates

Using an optogenetic tool to interrogate protein properties associated with phase separation in living cells[17,18], we identified surprisingly strong light-induced clustering of human RPA2 (Fig. 1a,b and Extended Data Fig. 1a–c). The prion-like N-terminal domain of FUS and an oligomerization-prone Cry2 mutant[19], which were included as two positive controls, also showed strong optoDroplet formation (Extended Data Fig. 1a–c). Other DNA-damage response (DDR) factors showed weaker or no light-induced clustering when over-expressed as Cry2-mCherry fusions, and neither expression of the Cry2-mCherry module alone (empty) nor as a fusion with dimerization-prone glutathione *S*-transferase (GST) resulted in optoDroplet formation (Extended Data Fig. 1a–c). We conclude that light-induced seeding requires self-assembly-driven amplification of protein condensation to cause discernable optoDroplet formation, and that RPA2 carries these features. Although other DDR factors did not show these features in the optoDroplet system, we do not exclude the possibility of their dynamic clustering at endogenous expression levels and in other cellular contexts.

RPA2 optoDroplets, once induced, remained stable for several minutes in the absence of blue light, with a half-life of around 5 minutes (Fig. 1c). The RPA2 optoDroplets showed a high degree of mobility (Supplementary Video 1) and frequently underwent droplet fusions (Fig. 1d). Furthermore, RPA2 optoDroplets could be induced multiple times sequentially in the same cell in a switch-like manner (Fig. 1e). Together, these results suggest that RPA2 has an intrinsic and hitherto unexplored propensity for dynamic self-assembly. As RPA2 forms a stable trimeric protein complex together with RPA1 and RPA3, we generated a polycistronic construct to express all three RPA subunits in stoichiometric amounts in cells. When expressed as a trimeric complex (tRPA), light-controlled clustering was observed within seconds after induction with blue light (Extended Data Fig. 2a–e), indicating that RPA condensation was not due to isolated expression of one of its subunits.

To exclude that light-induced clustering of RPA was due to protein over-expression, we analyzed nuclear RPA levels in transfected cells by multicolor high-content microscopy (Extended Data Fig. 3a). By differentiating cells according to their Cry2-mCherry-RPA2 expression levels into transfected (positive) and untransfected (negative) cells, we identified cells in both categories with similar RPA2 expression on the basis of RPA2 antibody staining. We observed that the transfected cells formed RPA2 optoDroplets despite the RPA2 concentration being close to the endogenous level (Extended Data Fig. 3b). Moreover, a short interfering RNA (siRNA) against RPA1 lowered endogenous RPA levels (measured by RPA1 and RPA2 co-staining) and allowed us to look at Cry2-mCherry-RPA2-transfected cells, in which RPA2 levels (after RPA depletion by siRNA) were comparable to endogenous RPA2 levels (cells transfected with non-targeting negative control siRNA (siControl) served as a reference). These cells also showed optoDroplet formation, excluding supraphysiological RPA expression as a cause of light-induced RPA clustering (Extended Data Fig. 3c).

Next, we analyzed intracellular RPA condensation in cells stably expressing fluorescently labeled tRPA[20]. RPA formed nuclear foci in these cells, and time-lapse experiments revealed fusion and occasional fission events after replication stress (Extended Data Fig. 4a). Similarly, replication-stress-induced RPA foci fusions and rare fission events were observed at physiological protein concentrations in cells in which the endogenous *RPA1* locus had been engineered, using CRISPR–Cas9, to express an mScarlet-RPA1 fusion protein (Extended Data Fig. 4b–d)[21].

RPA foci appeared to be of spherical shape in widefield and confocal microscopy images, a feature that was also observed when nuclear RPA ensembles were imaged at higher resolution by stimulated emission depletion (STED) microscopy (Extended Data Fig. 4e). Fluorescence recovery after photobleaching (FRAP) experiments to evaluate RPA exchange at individual RPA foci showed fast recovery rates, with half-recovery times below 10 seconds, both in unchallenged cells and in cells exposed to replication-stress- and DNA-damage-inducing treatments (Extended Data Fig. 4f–h). These half-recovery times at endogenous RPA foci and at ATR-kinase-inhibitor- and ionizing radiation (IR)-induced DNA lesions are in the same range as was measured for other proteins with a high propensity to phase separate, including FUS, hnRNPA1, and DDX4 (ref. [22]). Moreover, endogenous and replication-stress-induced RPA foci were sensitive to a short treatment with low concentrations of the aliphatic alcohol 1,6-hexanediol, which interferes with weak hydrophobic interactions (Extended Data Fig. 4i,j).

### Phase separation of the RPA complex is stimulated by ssDNA

Collectively, these findings raised the possibility that RPA forms condensates owing to protein properties linked to phase separation. To further explore this possibility, we expressed and purified the untagged human tRPA complex over three sequential columns[23–25], resulting in highly pure tRPA fractions (Fig. 2a and Supplementary Fig. 1a,b). The purified tRPA complex readily formed dynamic liquid droplets in vitro (Fig. 2b, Supplementary Fig. 1c, and Supplementary Video 2), and their formation was enhanced by increasing concentrations of

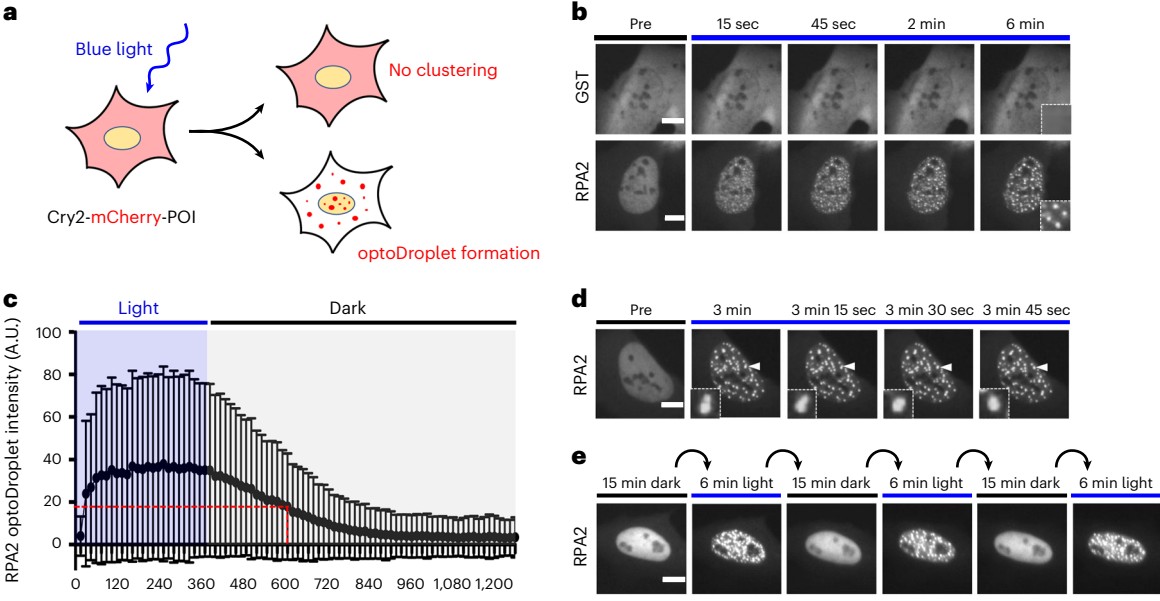

**Fig. 1 | RPA2 forms dynamic intracellular optoDroplets. a**, Schematic of blue-light-induced clustering of Cry2-mCherry fusion proteins (proteins of interest, POI) into microscopically discernible optoDroplets[17,18]. **b**, Time-resolved optoDroplet formation of RPA2, with GST fused to Cry2-mCherry included as negative control. Representative stills from live-cell microscopy are shown. **c**, Lifetime of blue-light-induced RPA2 optoDroplets. Formation of optoDroplets was induced and optoDroplet dissolution was followed in the absence of blue light. Average optoDroplet intensities and s.d. for $n = 240$ cells are shown. **d**, RPA2 optoDroplets are highly dynamic and show optoDroplet fusions. Representative stills from live-cell microscopy are shown. See Supplementary Video 1. **e**, Reversibility and repeatable inducibility of RPA2 optoDroplets by cycles of blue-light activation for 6 minutes, followed by 15 minutes of recovery in the dark. Scale bars, 10 μm.

the crowding agent PEG-8000 (Supplementary Fig. 1d). RPA droplets displayed gravity flow resulting in surface wetting at plate bottoms (Fig. 2b and Supplementary Video 2), and frequently underwent droplet fusions (Fig. 2c), behaviors consistent with phase separation[26,27]. Freshly formed RPA droplets were initially mobile, but then settled and only partially formed again upon remixing (Supplementary Fig. 1e). RPA droplets that had formed in vitro were dissolved by 1,6-hexanediol (Supplementary Fig. 1f) and by millimolar concentrations of ATP, which has previously been reported to act as a biological hydrotrope that solubilizes aggregation-prone proteins in aqueous solutions[28] (Supplementary Fig. 1g). Taken together, these results demonstrate that the human trimeric RPA complex has a propensity to phase separate and form liquid droplets in vitro.

Given the prominent role of RPA as a ssDNA-binding protein, we went on to test whether ssDNA affects the phase-separation behavior of RPA. Strikingly, ssDNA, but not sequence-matched annealed double-stranded DNA (dsDNA), greatly enhanced RPA phase separation (Fig. 2d and Extended Data Fig. 5a,b). Although short ssDNA oligomers of 10–12 nucleotides in length did not measurably induce RPA phase separation, ssDNA molecules that were 20–50 nucleotides long were sufficient to robustly trigger RPA droplet formation and render RPA solutions turbid (Extended Data Fig. 5c,d). These findings are consistent with the biochemically characterized ssDNA binding preference of RPA[2,6] and suggest that RPA–ssDNA binding regulates RPA phase-separation behavior. Of note, titration of both RPA and ssDNA demonstrated that sub-stoichiometric amounts of 40-nucleotide-long ssDNA are sufficient to trigger RPA phase separation at physiological RPA concentrations (Fig. 2e,f). Labeling of purified RPA and ssDNA with fluorescent dyes revealed that ssDNA not only enhanced RPA phase separation, but also was specifically enriched in RPA-containing droplets (Fig. 2g,h and Extended Data Fig. 5e). Matched RNA, on the other hand, did not induce RPA phase separation and was not enriched in RPA droplets (Extended Data Fig. 5f,g). Without RPA, ssDNA did not

form discernible liquid droplets under these conditions (Extended Data Fig. 5h), and ssDNA-containing RPA droplets were dissolved by 1,6-hexanediol (Extended Data Fig. 5i). Taken together, ssDNA, but not dsDNA or ssRNA, induces RPA phase separation in vitro and is enriched in RPA condensates.

## Multi-site phosphorylation regulates RPA self-interaction

Next, to elucidate the mechanism and regulation of RPA phase separation, we turned again to the cellular optoDroplet system for the following reasons: the induction of optoDroplets by blue light not only provides a means to reversibly trigger protein condensation in the natural environment of living cells, but also to compare individual proteins, protein domains, and point mutants with respect to their propensity to form biomolecular condensates. When we compared the individual RPA subunits RPA1, RPA2, and RPA3 side by side, RPA2 showed the most pronounced light-induced clustering in cells (Fig. 3a–c), suggesting that RPA2 is the main driver of RPA assembly into light-induced condensates. To address whether RPA self-assembly occurs in the absence of blue-light-induced seeding, we immunoprecipitated green fluorescent protein (GFP)-RPA2 from benzonase-treated lysates of a stable GFP-RPA2 cell line and tested whether other RPA subunits are enriched together with GFP-RPA2. Remarkably, GFP-RPA2 pulled down not only its constitutive interaction partner RPA1, but also endogenous RPA2 (Fig. 3d).

RPA2 has two globular domains connected by a short linker, and an extended N-terminal intrinsically disordered region (N-IDR) with multiple phosphorylation sites[29] (Fig. 3e–h). AlphaFold predicted the N-IDR to be flexible and unconstrained by the two globular domains of RPA2 (Fig. 3e), with a high expected position error for the first 40–45 amino acids and a low expected position error for the two globular domains (Fig. 3f). Two sequence-based prediction tools for protein folding and disorder, GlobPlot[30] and PONDR (Predictor of Natural Disordered Regions)[31], agreed on the N-IDR being unstructured (Fig. 3g,h).

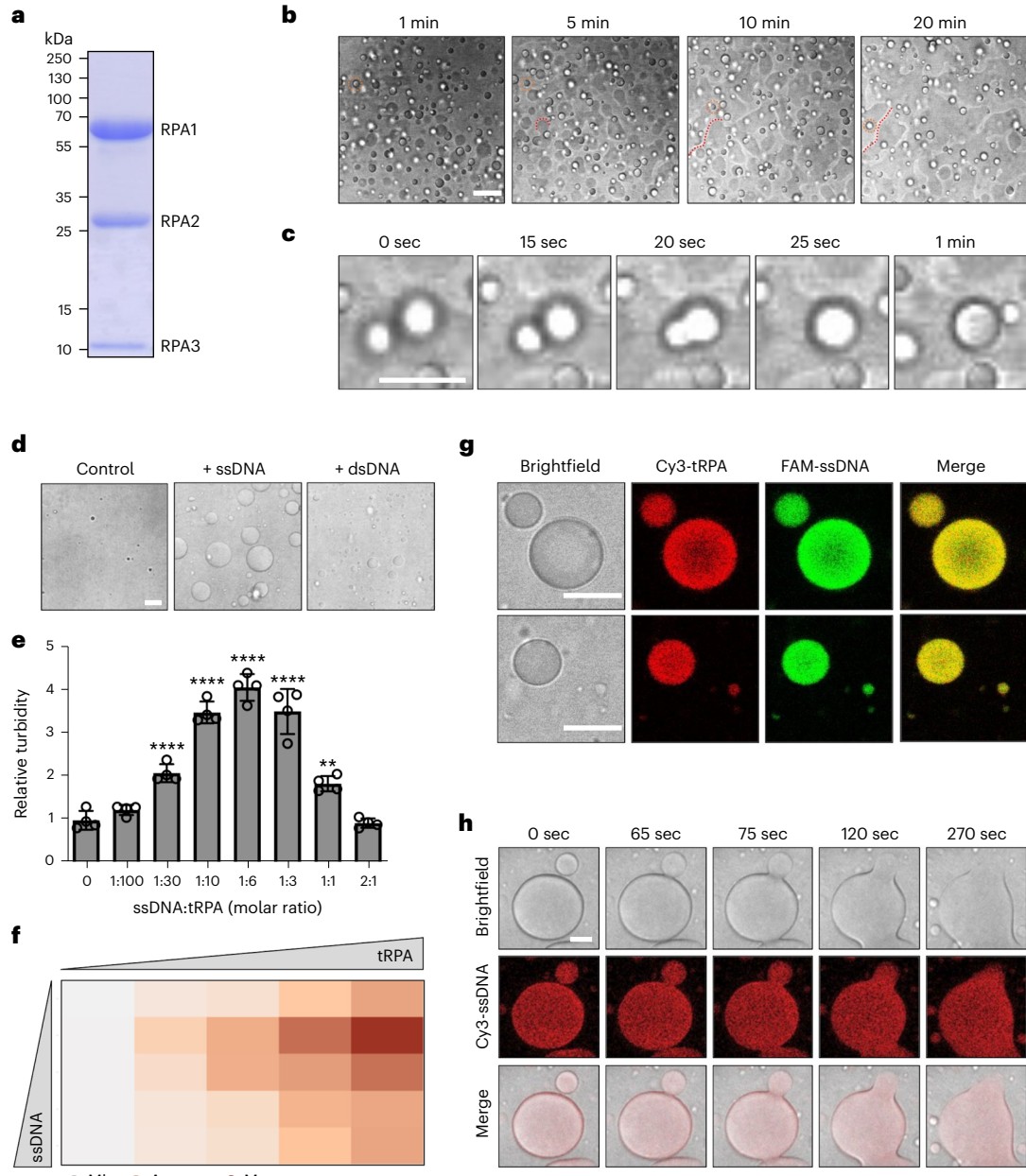

**Fig. 2 | Purified heterotrimeric RPA forms liquid droplets in vitro.**
**a**, Coomassie staining of the purified trimeric human RPA complex separated by sodium dodecyl sulfate–polyacrylamide gel electrophoresis (SDS–PAGE). **b**, Spontaneous formation of spherical liquid droplets and surface wetting by the purified RPA complex. Representative stills from time-lapse microscopy are shown. For illustration purposes, one RPA droplet per image is marked by an orange dotted circle, and examples of surface wetting at the plate bottom are marked by a red dotted curved line. See Supplementary Video 2 and Methods for details. **c**, Example of RPA droplet fusion from time-lapse microscopy as in **b**. **d**, Stimulation of RPA condensation by ssDNA. The purified trimeric RPA complex was incubated with equimolar amounts of 40-nucleotide-oligomer ssDNA (Supplementary Table 3), or sequence-matched dsDNA, as indicated, and RPA droplet formation was analyzed. **e**, Turbidity measurements of purified

trimeric RPA incubated with different molar ratios of 40-nucleotide-oligomer ssDNA. Turbidity measurements with $n = 4$ replicates were performed and normalized to control. Averages and s.d. are shown. One-way analysis of variance (ANOVA) with Dunnett's test compared with control, 1:30, 1:10, 1:6, 1:3 ****$P < 0.0001$; 1:1 **$P = 0.001$. **f**, Turbidity phase diagram of RPA versus ssDNA (concentration range: 0 µM, 2.5 µM, 5 µM, 10 µM, and 15 µM for both RPA and 40-nucleotide-oligomer ssDNA). A heat map of average turbidity measurements from $n = 4$ replicates is shown. **g**, Co-assembly of ssDNA into RPA droplets. Cy3-labeled purified RPA was incubated with equimolar 17-nucleotide-oligomer FAM-labeled ssDNA. Two representative example images of RPA-ssDNA droplets are shown. **h**, RPA-ssDNA droplets maintain liquid properties and undergo fusion. Purified RPA was incubated with equimolar 40-nucleotide-oligomer Cy3-labeled ssDNA. Representative stills from time-lapse microscopy are shown. Scale bars, 10 µm.

The N-IDR of human RPA2 and the N-IDR of its yeast homolog Rfa2 share little homology (Extended Data Fig. 6a) and, unlike human RPA2, yeast Rfa2 showed no light-induced clustering in optoDroplet experiments in mammalian cells (Extended Data Fig. 6b–d). We therefore focused our attention on the human N-IDR of RPA2 with its multiple phosphorylation sites. While alanine mutations did not abolish the condensation

behavior of RPA2 in the optoDroplet system, phospho-mimicking aspartate mutations completely abrogated light-inducible RPA2 clustering (Fig. 4a–c). In a series of single, double, triple, and quadruple S/T→D mutants, there was a gradual suppression of light-inducible RPA2 clustering, with the most pronounced clustering defect being observed when multiple phosphorylation sites (S8D, S11D, S12D, S13D,

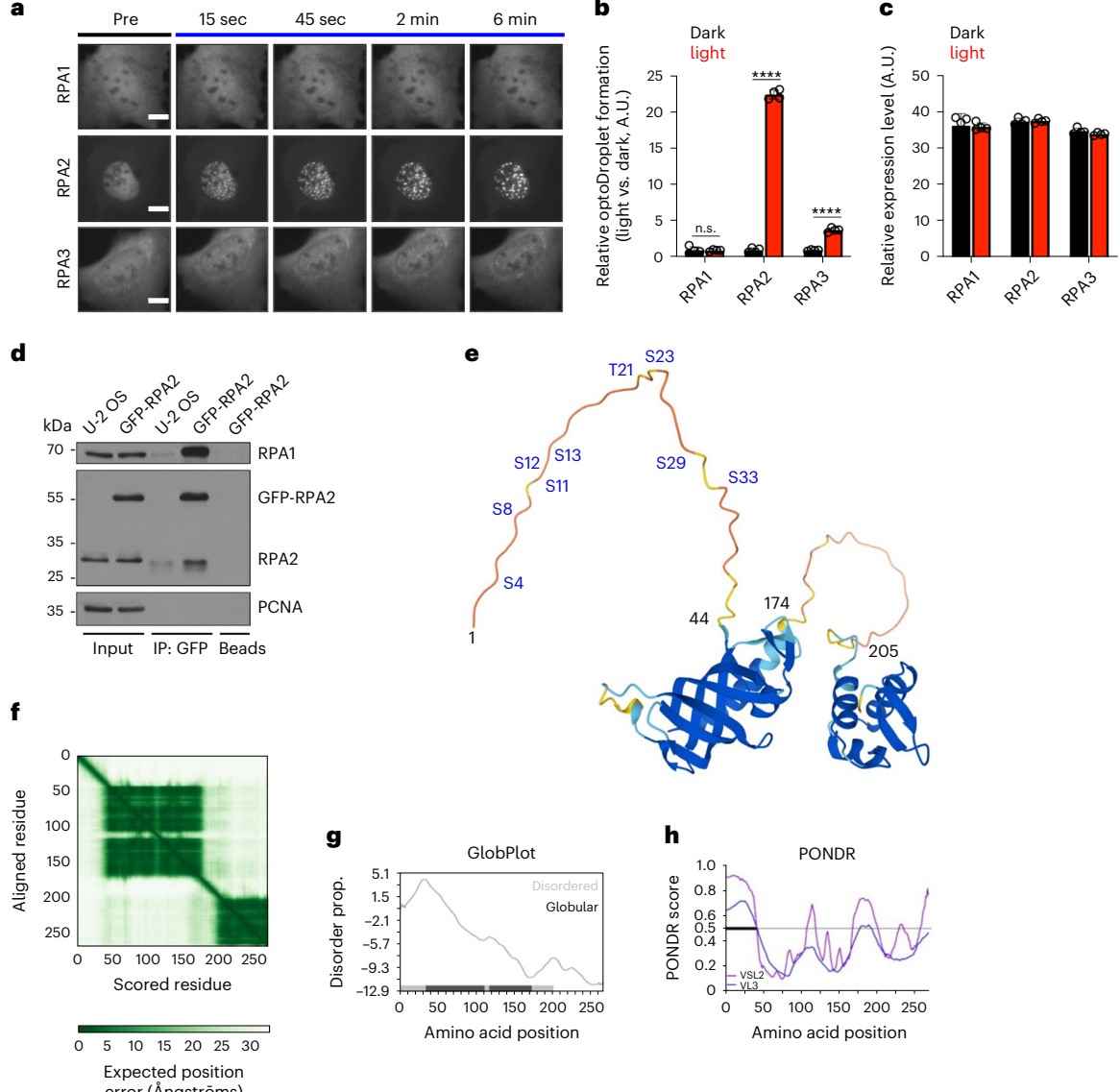

**Fig. 3 | RPA2 is the main driver of RPA clustering. a**, Time-resolved optoDroplet formation of RPA1, RPA2, and RPA3 fused to Cry2-mCherry. Representative stills from live-cell microscopy are shown. Scale bars, 10 μm. **b**, Accumulated optoDroplet intensity per nucleus of single RPA subunits RPA1, RPA2, or RPA3 fused to Cry2-mCherry was analyzed and normalized to the average accumulated optoDroplet intensity of the corresponding dark condition. Two-way ANOVA with Šidák`s test; RPA2, RPA3 ****$P < 0.0001$; RPA1 not significant (n.s.) $P = 0.9998$. **c**, Nuclear mean intensities of Cry2-mCherry in cells analyzed in **b**. **b,c**, Averages and s.d. are shown for $n = 4$ independent samples (cell number: RPA1$_{dark}$ $n_1 = 143$, $n_2 = 168$, $n_3 = 175$, $n_4 = 114$; RPA1$_{light}$ $n_1 = 203$, $n_2 = 200$, $n_3 = 241$, $n_4 = 225$; RPA2$_{dark}$ $n_1 = 937$, $n_2 = 943$, $n_3 = 855$, $n_4 = 768$; RPA2$_{light}$ $n_1 = 1,092$, $n_2 = 947$, $n_3 = 1,137$, $n_4 = 1,024$; RPA3$_{dark}$ $n_1 = 1,093$, $n_2 = 1,231$, $n_3 = 1,005$, $n_4 = 814$; RPA3$_{light}$ $n_1 = 1,199$, $n_2 = 1,447$, $n_3 = 1,643$, $n_4 = 1,424$). A.U., arbitrary units. **d**, GFP co-IP from naive U-2 OS cells and from U-2 OS cells stably expressing GFP-RPA2 to probe for specific RPA2 self-interaction. Endogenous RPA2 and GFP-RPA2 were detected

with an anti-RPA2 antibody. **e**, AlphaFold[66] protein structure prediction for RPA2. Consistent with X-ray crystallography data on the RPA trimerization core, two globular domains, a flexible linker, and an N-IDR were predicted. Putative and confirmed phosphorylation sites within the N-IDR are indicated. **f**, AlphaFold Predicted Aligned Error (PAE) heatmap depicting the distance error for every pair of residues as an estimate of position error for the structure prediction shown in **e**. RPA2 residue numbers run along the vertical and horizontal axes, and the color indicates PAE values for the corresponding pair of residues. **g**, RPA2 protein disorder prediction by GlobPlot 2.3, an online tool plotting the tendency within query proteins for order/globularity and disorder[30]. Disorder propensity is shown on the *y* axis, and amino acid position of RPA2 is on the *x* axis. **h**, RPA2 protein disorder prediction by PONDR (Predictor of Natural Disordered Regions)[31]. PONDR score is shown on the *y* axis, and amino acid position of RPA2 is on the *x* axis. Highly disordered regions are marked by the black bar.

T21D, S23D, S29D, and S33D)[32] were altered (Fig. 4d and Extended Data Fig. 6e–h). In accordance, when we expressed and purified trimeric wild-type (WT) RPA and the complete phospho-mimicking S/T→D mutant, only WT tRPA was able to form liquid droplets in vitro (Fig. 4e,f). When purified tRPA was phosphorylated in vitro by the DDR kinase DNA-PK, which was previously shown to antagonize the self-assembly of FUS into hydrogels through phosphorylation of its low complexity

domain[33], tRPA phase separation was reduced (Fig. 4g,h). Moreover, in the presence of ssDNA, the S/T→D mutant formed ssDNA-containing aggregates in vitro, unlike the dynamic ssDNA-containing droplets formed by WT tRPA (Fig. 4i), and the dynamic exchange of ssDNA, measured by FRAP, was reduced in these aggregates (Extended Data Fig. 6i,j). These results are consistent with previous data showing that phosphosite mutations in RPA2 are compatible with ssDNA binding[32,34],

but they also suggest that the rapid exchange between RPA and ssDNA may be facilitated by dynamic condensate properties.

## RPA condensation properties are linked to telomere maintenance

To explore biological functions potentially associated with dynamic RPA condensation, we generated stable cell lines in which light-induced RPA2 optoDroplet formation, without induction of DNA damage and independent of RPA's known roles in genome maintenance and repair, is coupled to TurboID-mediated proximity labeling (Fig. 5a). Negative control cells expressed TurboID fused to mCherry and the blue-light sensor Cry2, whereas matched test cells expressed TurboID-RPA2 fused to mCherry-Cry2. As anticipated, only the RPA2-expressing cells formed light-inducible condensates (Fig. 5b). Using a comparatively short biotin labeling time of 15 minutes, with and without simultaneous blue-light induction, we performed label-free quantitative proteomics (TurboID-RPA2 light, TurboID-RPA2 dark, TurboID light; six replicates per condition). The proximity labeling coupled to mass spectrometry revealed 186 proteins common to the TurboID-RPA2 light and TurboID-RPA2 dark conditions (Fig. 5c), including all three RPA subunits and several known interactors of RPA (Supplementary Table 1). Importantly, the proximity labeling proteomics also revealed 53 proteins that were significantly enriched in the TurboID-RPA2 light versus TurboID-RPA2 dark conditions ($P \leq 0.05$, fold change $\geq 1.5$). With only four exceptions, these proteins were also specifically enriched in the TurboID-RPA2 light versus TurboID light comparison (Supplementary Table 2). Of note, among the RPA clustering-induced interactions were components of the BLM–TOP3A–RMI (BTR) complex together with its associated partners FANCM and RAD52 (Fig. 5d).

The BTR complex dissolves double Holliday junctions to prevent crossovers during homologous recombination and is recruited to stalled replication forks to promote fork restart[35,36]. Moreover, the BTR complex localizes to telomeres and, together with FANCM and RAD52, is involved in alternative lengthening of telomeres (ALT), a recombination-dependent telomere maintenance pathway used by telomerase-negative cancers to achieve replicative immortality[37–39]. We have previously found that ALT is associated with replication-stress-induced post-mitotic DNA synthesis (post-MiDAS) in G1 cells at genomic regions marked by RPA[21]. Parallel work has shown that ALT is a self-perpetuating process, which involves telomere clustering and RAD52-dependent telomere synthesis in PML bodies[40–42]. We therefore hypothesized that RPA self-assembly into nuclear condensates could be involved in telomere clustering and RAD52-mediated telomere elongation. In support of this possibility, liquid droplets formed in vitro by purified unlabeled trimeric RPA selectively enriched labeled WT RPA, but not a RPA mutant lacking the RPA2 N-IDR (ΔN), and also resulted in partitioning of purified RAD52 into the preformed RPA droplets (Extended Data Fig. 7a–c).

RPA and RAD52 co-localized at the telomeres of ALT-positive cancer cells marked by telomere repeat binding factor 2 (TRF2), and

their association with telomeres was further enhanced by replication stress or when ALT activity was increased by FANCM depletion, as has been demonstrated previously[43,44] (Extended Data Fig. 8a–j). Next, we generated stable cell lines expressing siRNA-resistant WT *RPA2* or mutated *RPA2* encoding the phase-separation-impaired but ssDNA-binding-proficient S/T→D mutant. This allowed for efficient depletion and replacement of endogenous RPA2, despite its essentiality (Supplementary Fig. 2a). The levels of GFP-RPA2 and RAD52 in these cells were unaffected by the RPA2 mutations (Supplementary Fig. 2b), and both cell lines showed comparable cell cycle profiles, similar responses to camptothecin (CPT)-induced DNA damage, and recruitment of RPA to telomeres (Extended Data Fig. 9a–g). However, the accumulated intensity of RPA in CPT-induced nuclear foci at sites of replication-fork stalling was lower in RPA S/T→D cells (Extended Data Fig. 9h). Similarly, the accumulated intensity of RPA in IR-induced foci was reduced in RPA S/T→D cells compared with cells expressing WT RPA (Extended Data Fig. 9i), indicating that RPA assembly was affected. Moreover, RPA S/T→D showed impaired self-interaction in co-immunoprecipitation (co-IP) experiments (Extended Data Fig. 9j), and RPA S/T→D cells had elevated markers of replication-stress-induced telomere fragility, including increased numbers of ALT-associated PML bodies (APBs), more DNA synthesis at telomeres outside S-phase in G2 and in G1, higher levels of ssDNA at telomeres, and more extrachromosomal C-circles (Fig. 6a–e and Supplementary Fig. 3a-e). On the other hand, RAD52 enrichment at stressed telomeres was reduced in RPA S/T→D cells (Fig. 6f,g and Supplementary Fig. 3d). Similar defects were observed in conditions of unrestrained ALT activity due to FANCM depletion (Supplementary Fig. 4a–i).

ALT telomeres are hyper-sensitive to replication stress, and replication stress promotes RPA binding at telomeric ssDNA and telomere clustering for ALT[40,45]. Considering the tight association between RPA and ALT telomeres (Extended Data Fig. 8a–f), we analyzed RPA fusion events in RPA WT and S/T→D mutant cells as a proxy for telomere clustering (Extended Data Fig. 10a). Despite RPA expression levels being indistinguishable in the analyzed WT and S/T→D cells (Extended Data Fig. 10b), the mutant cells showed a defect in replication-stress-induced fusions (Fig. 6h). Conversely, inhibition of the nuclear kinases that are involved in RPA2 phosphorylation in a partly redundant manner (ATM, ATR, DNA-PK, CDK1 and CDK2) resulted in elevated RPA foci fusions (Extended Data Fig. 10c). A double-tagged cell line expressing GFP-RPA2 and TRF2-RFP confirmed that RPA foci fusions coincide in space and time with TRF2 fusions, indicating that they occur at telomeres (Extended Data Fig. 10d,e). In addition to the clustering defects, RPA2 S/T→D cells displayed elevated telomere loss compared with that of WT cells (Fig. 6i and Extended Data Fig. 10f). Together with the optoDroplet and in vitro RPA phase separation data, these results support the notion that the propensity of RPA to form dynamic condensates promotes telomere clustering and facilitates RAD52-mediated telomere maintenance in ALT cancer cells.

---

**Fig. 4 | The phosphorylated N-IDR of RPA2 modulates RPA phase separation properties. a**, Time-resolved optoDroplet formation of RPA2 WT, RPA2 S/T→A, and RPA2 S/T→D fused to Cry2-mCherry. Representative stills from live-cell microscopy are shown. **b**, Accumulated optoDroplet intensity per nucleus of WT RPA2, phosphodeficient RPA2 S/T→A, and phosphomimetic RPA2 S/T→D fused to Cry2-mCherry was analyzed and normalized to the average accumulated optoDroplet intensity of the corresponding dark condition. Two-way ANOVA with Šidák`s test, RPA2 WT, RPA2 S/T→A ****$P < 0.0001$; RPA2 S/T→D n.s. $P = 0.9435$. **c**, Nuclear mean intensities of Cry2-mCherry in cells analyzed in **b**. **b,c**, Averages and s.d. are shown for $n = 4$ independent samples (>350 cells per sample). **d**, Accumulated optoDroplet intensity per nucleus of WT RPA2 and phosphomimetic mutants fused to Cry2-mCherry was analyzed. Averages and s.d. are shown for $n = 4$ independent samples (>15 cells per sample, average 61 cells per sample). Two-way ANOVA with Šidák`s test, RPA2 WT, RPA2-S8D,

RPA2-S11D, RPA2-S12D, RPA2-T21D, RPA2-S23D, RPA2-S33D, RPA2-S4D S8D, RPA2-S23D S33D, RPA2-S8D S33D, RPA2-T21D S33D, RPA2-S11D S12D S13D, RPA2-T21D S29D S33D, RPA2-S8D T21D S33D ****$P < 0.0001$; RPA2-T21D S23D S29D S33D ***$P = 0.0003$; RPA2 S/T→D n.s. $P > 0.9999$. **e**, Coomassie staining of the purified trimeric human RPA complexes (WT and S/T→D) separated by SDS–PAGE. **f**, Formation of liquid droplets by the purified WT RPA complex, but not by the phosphomimetic S/T→D mutant. Insets show magnifications. **g**, In vitro phosphorylation of the purified trimeric WT RPA complex by DNA-PK, with a western blot as a control. **h**, In vitro phosphorylation of the purified trimeric WT RPA complex by DNA-PK impairs RPA phase separation. Representative stills from time-lapse microscopy are shown. **i**, Incubation of purified trimeric Cy3-labeled RPA WT with equimolar 40-nucleotide-oligomer FAM-labeled ssDNA results in liquid droplet formation, whereas RPA S/T→D forms aggregate-like structures. Insets show magnifications. Scale bars, 10 μm.

## Discussion

Principles of phase separation by associative polymers provide a framework to investigate liquid unmixing behavior of proteins in vitro and complex biomolecular condensates in different physiological and pathological conditions in vivo[46–49]. RPA is the main ssDNA-binding protein in mammalian cells and is essential for most DNA transactions, including replication, recombination, and repair. RPA-coated ssDNA is protected from unscheduled nucleolytic cleavage and degradation, secondary structure formation, and spontaneous breakage. Moreover, RPA bound to ssDNA serves as a binding platform for several genome caretakers and coordinates the handover of ssDNA to proteins involved in replication-fork processing, recombination, and repair[2]. Although

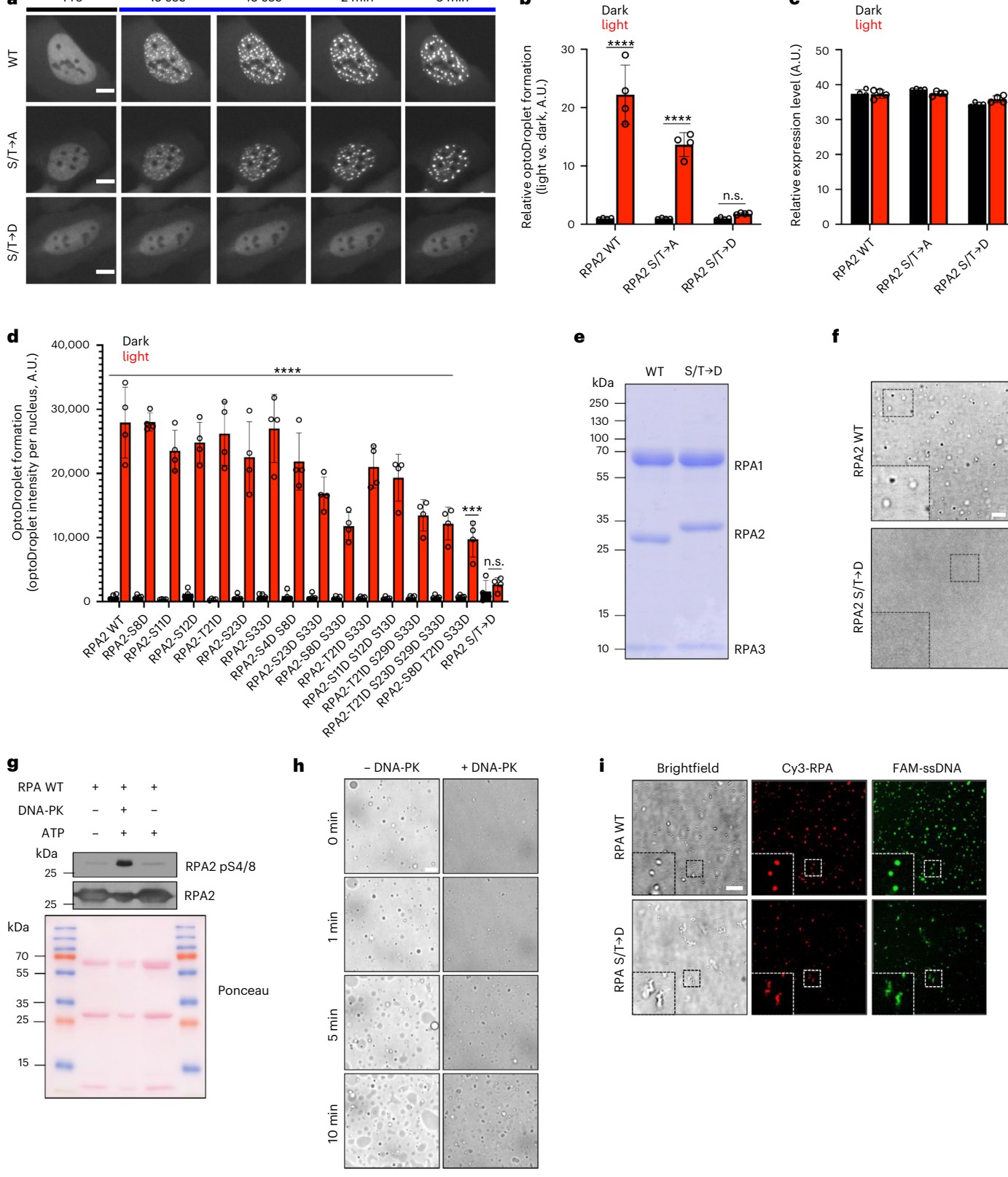

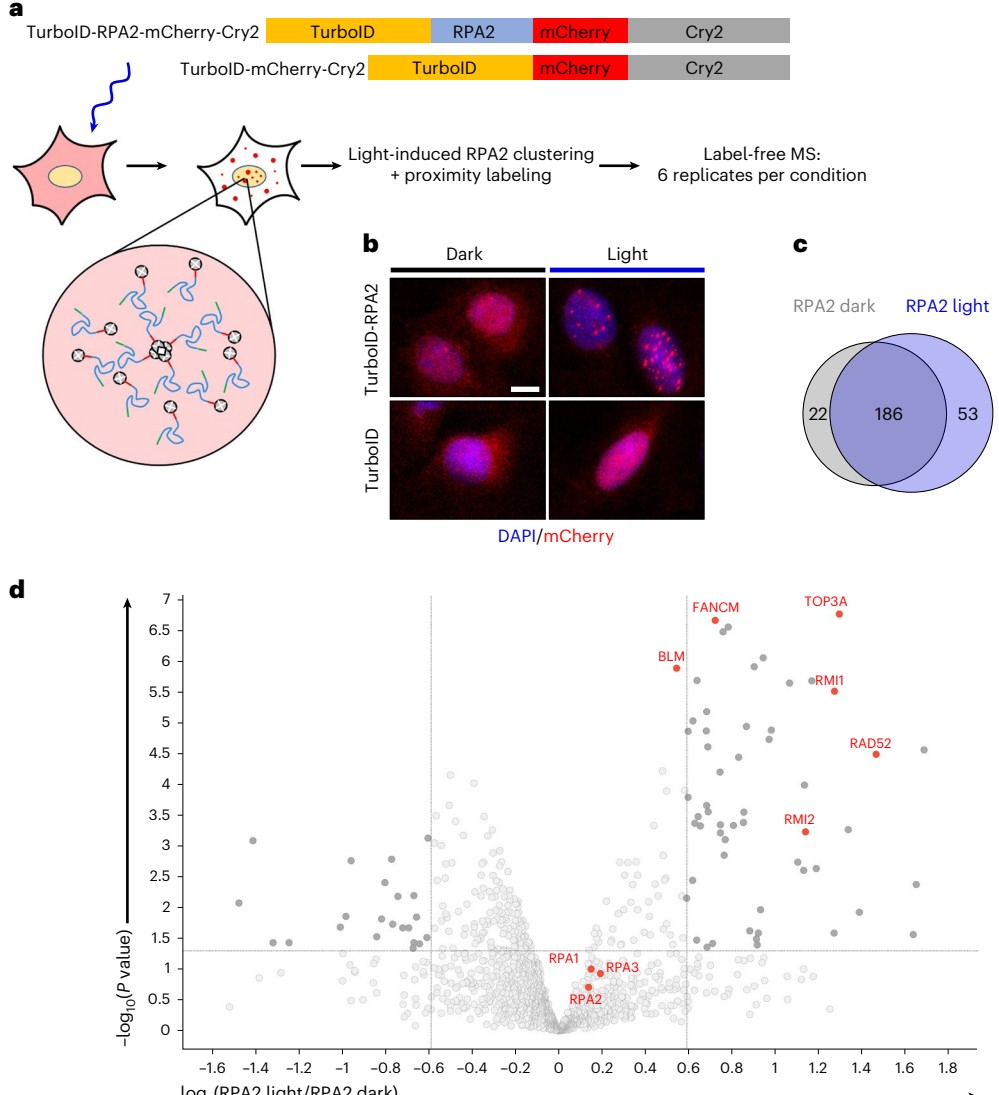

**Fig. 5 | RPA condensation enhances interactions with the BTR complex.**
**a**, Schematic of proximity-labeling mass spectrometry to identify RPA condensation-dependent interactions. A TurboID-mCherry-Cry2 cell line was used as negative control. A corresponding TurboID-RPA2-mCherry-Cry2 cell line was used with and without blue-light induction (15 minutes). Biotin proximity labeling was performed for 15 minutes. Six replicates per condition were analyzed by mass spectrometry for label-free quantification. **b**, Representative fluorescence microscopy images of the indicated cell lines, with and without blue-light induction. Scale bar, 10 μm. **c**, Venn diagram of specifically enriched proteins from proximity labeling mass spectrometry (fold change ≥ 1.5; $P \leq 0.05$) in conditions without light-induced RPA condensation (RPA2 dark) versus with light-induced RPA condensation (RPA2 light). **d**, Volcano plot of differentially enriched proteins in RPA2 light versus dark conditions (fold change ≥ 1.5; $P \leq 0.05$). RPA, BTR complex components, and BTR-associated proteins are highlighted in red. Note that BLM scored just below the fold change threshold. **c**,**d**, $P$ values were assessed by moderated $t$-test from $n = 6$ replicates.

RPA-coated ssDNA has often been viewed and depicted as linear 'beads on a string,' a dynamic exchange of RPA is required to prevent the formation of rigid filaments and allow other proteins to gain access to ssDNA[5,8]. In accordance with previous biochemical and structural studies[50–54], our results confirm that the RPA complex is flexible and binds ssDNA in a highly dynamic manner. Additionally, we found that RPA also self-interacts dynamically, both in vitro and in the cell nucleus. RPA self-interaction is concentration-dependent and can be triggered by molecular seeds, such as in the light-controllable Cry2 system, at endogenous expression levels, or by its physiological binding substrate ssDNA. Of note, sub-stoichiometric amounts of ssDNA were sufficient to trigger RPA phase separation in vitro. Considering that nuclear RPA is in large excess over ssDNA under physiological conditions, these results indicate that ssDNA-seeded self-assembly of RPA into dynamic condensates is likely favored over stoichiometric RPA-ssDNA binding in vivo.

Biochemical experiments using single-molecule ssDNA curtains previously demonstrated that free RPA in solution was required for dynamic exchange of RPA with ssDNA[55,56]. Our results are consistent with these observations and suggest that nuclear RPA condensates provide a reservoir of highly concentrated free RPA in excess over the bound ssDNA, which enables rapid exchange of RPA molecules on the enclosed ssDNA. Dynamic condensate formation, initiated by ssDNA binding and amplified by the dynamic self-interaction properties of RPA, can thus explain how ultrahigh-affinity binding to ssDNA and rapid RPA exchange are both achieved simultaneously (Fig. 6j).

Mechanistically, we found that RPA2 is critical for RPA phase separation in the context of the functional RPA heterotrimer. Within RPA2, we identified the N-IDR to be involved in RPA phase separation. Bacterial SSB was recently reported to form phase-separated liquid condensates in vitro and in *Escherichia coli* cell extracts, dependent on

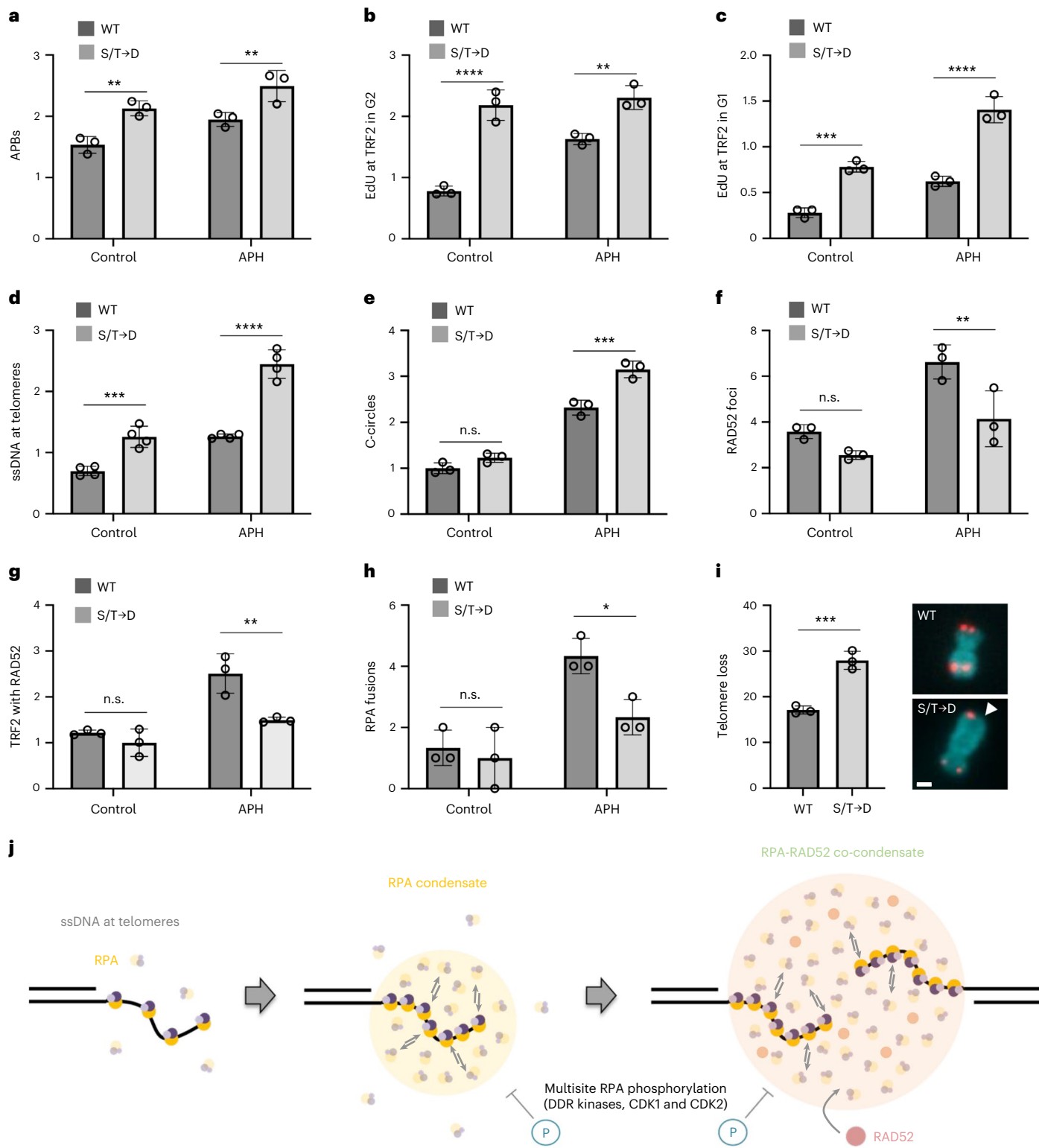

its internal intrinsically disordered linker region[14]. In yeast, however, the RPA subunit Rfa1 (corresponding to human RPA1) exhibits tighter binding to ssDNA and slower motion inside subcellular foci than does the yeast DNA-repair protein Rad52 (BRCA2 in human), suggesting that yeast Rad52, but not Rfa1, displays phase-separation behavior[15,16]. Consistently, yeast Rfa2 (corresponding to human RPA2) failed to form optoDroplets in our experiments, although it remains possible that this would be different in a yeast cell environment. The yeast Rfa2 N-IDR shows only little sequence homology with the human RPA2 N-IDR, and

the larger size of the human genome and the bigger volume of the cell nucleus may have required additional layers of control for the spatial and temporal regulation of subcellular compartmentalization. This requirement could be one of the reasons why human cells have acquired and preserved condensation properties of RPA2 that can be modulated by kinase-dependent phosphorylation of its N-IDR.

RPA phosphorylation has been studied in the context of its ability to bind and melt ssDNA structures and regulate DNA resection, and multiple DDR kinases and CDKs are involved in RPA2

**Fig. 6 | RPA condensation is linked to telomere maintenance. a**, RPA2-depleted U-2 OS cells stably expressing WT GFP-RPA2 (encoded by an siRNA-resistant sequence) or the S/T→D mutant were stained for PML and TRF2 to quantify APBs. Averages and s.d. from $n = 3$ independent samples, 50 cells per replicate. Two-way ANOVA with Šidák`s test, control **$P = 0.0047$, APH **$P = 0.0077$. **b**, Cells treated as in **a** were EdU labeled for cell cycle staging by QIBC, and G2 cells were analyzed for EdU-positive TRF2 foci. Averages and s.d. from $n = 3$ independent samples for 100 G2 cells per replicate. Two-way ANOVA with Šidák`s test, control ****$P < 0.0001$, APH **$P = 0.0024$. **c**, Cells were treated the same as in **b**, but G1 cells were analyzed. Averages and s.d. from $n = 3$ independent samples for 100 G1 cells per replicate. Two-way ANOVA with Šidák`s test, control ***$P = 0.0002$, APH ****$P < 0.0001$. **d**, Cells were treated as in **a** for non-denaturing TelG-FISH. Averages and s.d. from $n = 4$ independent samples (>1,000 cells per sample). Two-way ANOVA with Šidák`s test, control ***$P = 0.0004$, APH ****$P < 0.0001$. **e**, Cells were treated as in **a** for C-circle analysis by quantitative PCR. Averages and s.d.

from $n = 3$ replicates. Two-way ANOVA with Šidák`s test, control n.s. $P = 0.1691$, APH ***$P = 0.0002$. **f**, Cells were treated as in **a** for RAD52 foci analysis. Averages and s.d. from $n = 3$ independent samples (>2,000 cells per sample). Two-way ANOVA with Šidák`s test, control n.s. $P = 0.2346$, APH **$P = 0.0066$. **g**, Cells were treated as in **a** for TRF2-RAD52 co-localization analysis. Averages and s.d. from $n = 3$ independent samples (>75 cells per sample). Two-way ANOVA with Šidák`s test, control n.s. $P = 0.5668$, APH **$P = 0.0032$. **h**, Cells were treated as in **a** for RPA foci fusion analysis from $n = 3$ independent 48-hour time-lapse experiments, 100 cells per replicate. Averages and s.d. are shown. Two-way ANOVA with Šidák`s test, control n.s. $P = 0.8233$, APH *$P = 0.0170$. **i**, Telomere loss in U-2 OS cells expressing GFP-RPA2 WT or S/T→D by metaphase telomere FISH analysis. Averages and s.d. from $n = 3$ independent samples (chromosomes/metaphases: WT $n_1 = 843/16$, $n_2 = 952/16$, $n_3 = 580/11$; S/T→D $n_1 = 830/14$, $n_2 = 949/16$, $n_3 = 378/8$). Two-tailed unpaired $t$-test, ***$P = 0.001$. Scale bar, 1 μm. **j**, Model of RPA condensate formation by ssDNA-seeded RPA self-assembly.

phosphorylation and functionally co-operate in human cells to achieve RPA2 hyper-phosphorylation[57–62]. Our results suggest that multi-site phosphorylation of RPA2, without abolishing ssDNA binding[32,34], can gradually reduce the phase-separation capacity of RPA, indicating that phosphorylation may fine-tune RPA condensation properties and condensate-related functions in a context-dependent manner.

Functionally, unbiased proximity proteomics revealed that RPA condensation leads to an enrichment of the BTR complex and its associated proteins FANCM and RAD52. As part of the BTR complex, RAD52 is involved in ALT in cancer[63]. We found that RAD52 readily partitions into phase-separated RPA droplets in vitro, and that RAD52 enrichment at sites of ALT activation is reduced in cancer cells expressing phase-separation-impaired, phosphomimetic RPA2. Rather than loss of APB formation, RPA S/T→D cells with reduced RAD52 recruitment to telomeres showed elevated C-circles, in accordance with a shift towards RAD52-independent ALT associated with C-circle formation and progressive telomere shortening[64]. In line with this, we observed more frequent telomere loss in cells expressing phosphomimetic RPA2. Considering that we also observed signs of impaired telomere clustering, homology search and donor template usage might be altered by RPA hyper-phosphorylation, for example, towards more intratelomeric recombination. Further studies employing dedicated assays to interrogate ALT sub-pathway usage will be needed to investigate this hypothesis. In summary, our findings suggest that the self-assembly and condensation properties of RPA functionally contribute to telomere maintenance in ALT-positive cancer cells, consistent with an emerging implication of phase separation at telomeres[65] and specifically in ALT[40–42]. Given that telomere maintenance by ALT represents a vulnerability of certain cancers, understanding the condensation properties of the involved molecules may help to improve targeted therapies against ALT-dependent tumors. Beyond ALT, RPA condensation properties may also be involved in other cellular contexts, including DNA-repair compartments formed at DSBs that undergo resection for repair by HR and stressed replication factories. We observed fast recovery in FRAP experiments at these regions, and although RPA foci formation was not abrogated in RPA S/T→D cells, consistent with the mutant being proficient in ssDNA binding, the accumulated intensity of RPA in these regions was reduced. These findings are consistent with a more general role of RPA condensation in genome function, although testing the implications of RPA's phase-separation properties in additional cellular contexts will require further studies.

## Online content

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

## Methods

### Cell culture

All cell lines were grown at 37 °C under standard cell culture conditions (humidified atmosphere, 5% $CO_2$) in Dulbecco's modified Eagle's medium (DMEM) containing 10% fetal bovine serum (Gibco) and 1% penicillin–streptomycin antibiotics. Stable U-2 OS GFP-RPA2 WT and S/T→D cell lines (siRNA-resistant) were maintained in the presence of 400 μg/ml Geneticin (Gibco). For experiments with siRNA-resistant GFP-RPA2 WT and S/T→D cells, the endogenous RPA2 was transiently depleted by siRNA. Stable U-2 OS cells expressing GFP-RPA2 and TRF2-RFP were maintained in the presence of 400 μg/ml Geneticin (Gibco) and 100 μg/ml hygromycin B (Thermo Fisher Scientific). Stable U-2 OS TurboID-RPA2-mCherry-Cry2 and TurboID-mCherry-Cry2 cells were maintained in the presence of 100 μg/ml hygromycin B (Thermo Fisher Scientific) and expression of the constructs was induced by the addition of 1 μg/ml doxycycline (Sigma-Aldrich) for 24 hours. Stable U-2 OS GFP-RPA cells expressing a polycistronic GFP-RPA construct for stoichiometric expression of the three RPA subunits RPA1, RPA2, and RPA3 (ref. [20]) were maintained in presence of 400 μg/ml Geneticin (Gibco). U-2OS mScarlet-RPA1 cells, stably expressing endogenously tagged RPA1 and ectopic GFP-tagged 53BP1 (ref. [21]), were maintained in the presence of 0.5 μg/ml puromycin (Sigma-Aldrich) and 5 μg/ml blasticidin (InvivoGen). All cell lines used in this study were grown in sterile conditions and routinely tested for mycoplasma contamination and are listed in Supplementary Table 4. The generation of stable cell lines is described in the Supplementary Methods.

### Cloning

Cloning was performed using chemically competent DH5α generated in house, derived from Library Efficiency DH5α Competent Cells (Thermo Fisher Scientific). All constructs were generated by two-piece Gibson assembly or by site-directed mutagenesis. All primers and the cloning strategy used for expression constructs are provided in the Supplementary Methods (sections: cloning of Cry2-mCherry-fusion constructs; cloning of TurboID-RPA2-mCherry-Cry2, cloning of GFP-RPA2 mutants, cloning of TRF2-RFP, cloning of p11d-tRPA-32ΔN) and Supplementary Table 5. Correct cloning and integration into target vectors were confirmed by sequencing. All plasmids used in this study are listed in Supplementary Table 6.

### siRNA and plasmid transfections

Individual siRNA transfections (siRPA1, s12127; siRPA2, s12130; siFANCM, s33621; final concentration of 25 nM) were performed with Ambion Silencer Select siRNAs using Lipofectamine RNAiMAX (Thermo Fisher Scientific). Negative Silencer Select control Neg2 from Ambion was used as a non-targeting control. Plasmid transfections for transient expression were performed with TransIT-LT1 (Mirus Bio), according to the manufacturer's instructions.

### EdU labeling

For pulsed EdU (5-ethynyl-2′-deoxyuridine, Thermo Fisher Scientific) labeling, cells were incubated for 20 minutes in medium containing 10 μM EdU. The Click-iT EdU Alexa Fluor Imaging Kit (Thermo Fisher Scientific) was used for EdU detection.

### Immunofluorescence staining

Immunofluorescence staining was performed as previously described[21,67]. Specifically, cells were grown on sterile glass coverslips, fixed in 3% formaldehyde in PBS for 15 minutes at room temperature, and permeabilized for 5 minutes in 0.2% Triton X-100 (Sigma-Aldrich) in PBS. Primary and secondary antibodies were diluted in filtered DMEM containing 10% FBS and 0.02% sodium azide, and antibody incubations were performed for 1–2 hours at room temperature. Coverslips were incubated for 10 minutes with PBS containing 4′,6-diamidino-2-phenylindole dihydrochloride (DAPI, 0.5 μg/ml) at

room temperature, subsequently washed three times in PBS and briefly submerged in distilled water before being mounted on glass slides with Mowiol-based mounting medium (Mowiol 4.88 in glycerol/TRIS). To stain RAD52, cells were fixed in ice-cold methanol for 20 minutes at −20 °C and processed as described above without the permeabilization step. Antibody information is provided in the Reporting Summary.

### Quantitative image-based cytometry

Automated multichannel widefield microscopy for quantitative image-based cytometry (QIBC) was performed as described previously[18,21,67] on an Olympus ScanR Screening System (ScanR Image Acquisition 3.01) equipped with an inverted motorized Olympus IX83 microscope, a motorized stage, IR-laser hardware autofocus, a fast emission filter wheel with one set of bandpass filters for multi-wavelength acquisition (DAPI (ex BP 395/25, em BP 435/26), FITC (ex BP 470/24, em BP 511/23), TRITC (ex BP 550/15, em BP 595/40), Cy5 (ex BP 640/30, em BP 705/72)), and a Hamamatsu ORCA-FLASH 4.0 V2 sCMOS camera (2048 × 2048 pixel, pixel size 6.5 × 6.5 μm) with a ×20 UPLSAPO (NA 0.75) air objective. Image information of cell populations was acquired under non-saturating conditions, and identical settings were applied to all samples within one experiment. Images were analyzed with the Olympus ScanR Image Analysis Software (version 3.0.1), a dynamic background correction was applied, and nuclei segmentation was performed using an integrated intensity-based object-detection module based on the DAPI signal. Foci segmentation was performed using an integrated spot-detection module. Downstream analyses were focused on properly detected interphase nuclei containing a 2N-4N DNA content as measured by total and mean DAPI intensities with comparable quantified GFP expression. Fluorescence intensities are depicted as arbitrary units. Color-coded scatter plots of asynchronous cell populations were generated with Spotfire data visualization software (version 7.9.1 and 10.10.1, TIBCO). For visualization of discrete data in scatter plots, mild jittering (random displacement of data points along the discrete data axes) was applied to demerge overlapping data points. Representative scatter plots are shown.

### Cry2 optoDroplet experiments

Cry2 optoDroplet experiments were performed as described previously[18]. Specifically, U-2 OS cells were seeded into a 96-well plate (Greiner μclear), and 24 hours prior to imaging, they were transfected with plasmid DNA. During live-cell microscopy, FluoroBrite DMEM supplemented with 10% fetal bovine serum (Gibco) and Glutamax (Thermo Fisher Scientific) was used. Time-lapse microscopy of optoDroplet formation upon blue-light exposure was carried out in temperature- and $CO_2$-controlled conditions (37 °C, 5% $CO_2$) on a GE Healthcare IN Cell Analyzer 2500HS (V7.4) with a PCO sCMOS 16-bit camera (2048 × 2048 pixels, pixel size 6.5 × 6.5 μm) using a CFI Plan Apo Lambda (NA 0.75) ×20 air objective at 15-second intervals for 6 minutes (25 ms ex BP 475/28, em BP 526/52; 100 ms ex BP 575/25, em BP 607.5/19). For optoDroplet quantification in Figure 1c and Extended Data Figure 6h, images from time-lapse microscopy were analyzed with the Olympus ScanR Image Analysis Software (version 3.0.1), a dynamic background correction was applied, and single-cell segmentation was performed using an integrated intensity-based object-detection module based on the mCherry signal. Droplet segmentation was performed using an integrated spot-detection module. Upon blue-light exposure, transfected cells were either kept in the dark to serve as negative controls or were exposed to 20 cycles of 5 seconds of blue light and 15 seconds of dark in a custom-made blue-light box equipped with eight 1-W LED lamps with a power of 500 Lm at 10-cm distance from the cells. Cells were then fixed in 3% formaldehyde in PBS for 15 minutes at room temperature, and stained with DAPI. Imaging and image analyses were performed on the Olympus ScanR Screening System, as described above. Expression levels were normalized between samples for each experiment. The relative optoDroplet formation per construct was defined as the

relative fold change to its corresponding negative (dark) control. For determination of optoDroplet stability, optoDroplet formation was induced as described above for 6 minutes followed by mCherry detection (100 ms ex BP 575/25, em BP 607.5/19) for 15 minutes at 15-second intervals without further blue-light exposure.

## Trimeric RPA purification

The trimeric human RPA complex was expressed in chemocompetent BL21 cells using p11d-tRPA(123) for expression of the WT protein (Addgene plasmid no. 102613, kindly provided by M. Wold[25]), p11d-tRPA-32Asp8 for expression of the phosphomimetic RPA2 S/T→D mutant (Addgene plasmid no. 102617, kindly provided by M. Wold[32]), and p11d-tRPA-32ΔN for the expression of RPA2 lacking the disordered N-terminal domain. The multi-step purification procedure was performed as described previously[24,25]. Transformed BL21 cells were grown in LB medium supplemented with 100 µg/ml ampicillin (Sigma-Aldrich) at 37 °C to an optical density of 0.6. Expression was induced by 0.4 mM IPTG following overnight incubation at 18 °C. Bacteria were collected by centrifugation, resuspended in lysis buffer (30 mM HEPES pH 7.5, 0.01% NP-40, 25 µM EDTA, 1 mM DTT, 10% glycerol, 1× complete protease inhibitor cocktail (Roche)) and lysed using a French press. Protein purification was performed on an ÄKTA Purifier (GE Healthcare). The lysate was first loaded onto a HiTrap Blue column (GE Healthcare), then washed sequentially with lysis buffer at different salt concentrations (50 mM KCl; 800 mM KCl; 400 mM NaSCN), and finally eluted with lysis buffer containing 1.5 M NaSCN. Peak fractions were pooled and loaded onto a HiTrap Desalting column (GE Healthcare). After desalting, the peak fractions were loaded onto a HiTrap Q column (GE Healthcare) and washed sequentially with lysis buffer at different salt concentrations (50 mM KCl; 87.5 mM KCl; 200 mM KCl), before final fractions were eluted in lysis buffer using a salt gradient from 200 mM KCl to 500 mM KCl. The collected protein fractions were subjected to SDS–PAGE using a 12% gel for Coomassie staining. Amicon Ultra 3 kDa MWCO centrifugal tubes (Millipore) were used to concentrate the peak fractions and for buffer exchange to diluted Sørensen buffer (43.4 mM $Na_2HPO_4$, 6.6 mM $KH_2PO_4$, pH 7.6) supplemented with 150 mM KCl, 25 µM EDTA, and 1 mM DTT. Protein concentrations were measured by Bradford assay. Freshly purified protein, kept at 4 °C, was used for in vitro assays.

## Preparation of RAD52

The sequence encoding human RAD52 was ordered from GenScript as codon-optimized for *E. coli*, and was cloned into pMALT-P (a kind gift of the Kowalczykowski laboratory, UC Davis) using BamHI and PstI restriction sites. Additionally, a His-tag was inserted before the MBP-tag to yield the final construct pMALT-His-MBP-PP-hRAD52co. RAD52 was then expressed in *E. coli*, upon induction with 0.5 mM IPTG and incubation overnight at 18 °C. RAD52 was purified by affinity chromatography using amylose resin, the MBP-tag was subsequently cleaved by PreScission protease. RAD52 was then applied on HiTrap Heparin column in 20 mM Tris-HCl pH 7.5, 1 mM EDTA, 0.5 mM DTT, 10% glycerol, and 100 mM NaCl. The column was eluted using a salt gradient from 100 mM NaCl to 600 mM NaCl. The pooled eluted fractions were briefly incubated with NiNTA resin to remove uncleaved His-MBP-RAD52 from the cleaved RAD52 protein, and the final sample was dialyzed into 20 mM Tris-HCl pH 7.5, 1 mM DTT, 10% glycerol, and 100 mM NaCl. Four liters of bacterial culture yielded 1 ml of 39 µM RAD52.

## Protein labeling

For fluorescence labeling of trimeric RPA complexes, the purified protein complexes were incubated with Sulfo-Cyanine3 maleimide (Luminoprobe) using an excess molar ratio of 40:1 (label:protein) overnight at 4 °C. The labeled protein complex was then mixed 1:9 with the unlabeled protein complex for in vitro droplet assays. For Extended Data Figure 7a,b, proteins (RPA WT, RPA ΔN, RAD52) were incubated with Sulfo-Cyanine3 maleimide (Luminoprobe) using an excess molar

ratio of 100:1 (label:protein) overnight at 4 °C. Then, 1 µM labeled protein was added to preformed unlabeled RPA droplets.

## Oligonucleotide hybridization

To generate a double-stranded DNA oligonucleotide, the 40-nucleotide ssDNA (Supplementary Table 3) was mixed with its reverse and complementary oligonucleotide at a 1:1 molar ratio. For annealing of the two strands, the mixture was incubated for 10 minutes at 80 °C and then slowly cooled down. To verify oligonucleotide annealing, the product and the two ssDNA oligonucleotides were subjected to 2% agarose gel electrophoresis and RedSafe (Lucerna-Chem) detection. Redsafe signals were acquired with Infinity ST5 Xpress.

## In vitro droplet experiments

In vitro droplet experiments were performed in a 384-well imaging plate (Greiner µclear) at room temperature in diluted Sørensen buffer (43.4 mM $Na_2HPO_4$, 6.6 mM $KH_2PO_4$, pH 7.6) supplemented with 150 mM KCl, 25 µM EDTA, and 1 mM DTT. Images were acquired using a ×63 HC PL APO corr CS2 oil objective (NA 1.4) on a Leica SP5 UV-VIS or Leica SP8 inverse FALCON confocal laser scanning instruments (Leika Application Suite X 3.5.7.23225) equipped for simultaneous brightfield and fluorescence imaging (time-lapse acquisitions at 5-second intervals). For Figures 2d,g,h and 4h,i and Extended Data Figs. 5a,d–i and 6i, freshly purified trimeric RPA was used at a final concentration of 7.5 µM in 4% PEG-8000 (Sigma-Aldrich). For Figure 4f, Supplementary Figure 1c,f,g, and Extended Data Figure 7a,b, RPA was used at a final concentration of 7.5 µM in 5% PEG-8000. For Figure 2b,c and Supplementary Figure 1e, RPA was used at a final concentration of 12.5 µM in 10% PEG-8000. Single- and double-stranded oligonucleotides used for in vitro RPA droplet experiments are provided in Supplementary Table 3. For quantification of droplet formation, labeled tRPA (9:1 unlabeled:labeled protein) was incubated with nucleotides, and condensate formation was assessed in quadruplicates by automated droplet quantification of the Cy3-labeled tRPA signal at three matched time-points per condition and per replicate using Olympus ScanR Analysis Software (version 3.2).

## Turbidity measurements

For turbidity measurements, the absorbance at 600 nm was measured in quadruplicates using a Tecan microplate reader (Tecan i-control 2.0). Protein mixtures were prepared in diluted Sørensen buffer (43.4 mM $Na_2HPO_4$, 6.6 mM $KH_2PO_4$, pH 7.6) supplemented with 150 mM KCl, 25 µM EDTA, 1 mM DTT, and 4% PEG-8000 in a 384-well plate (Greiner µclear). The single-stranded oligonucleotides used for turbidity measurements are listed in Supplementary Table 3. Turbidity measurements were normalized to the negative control condition.

## Co-immunoprecipitation

U-2 OS and U-2 OS GFP-RPA2 WT and S/T→D cells were washed twice with PBS and directly lysed on ice in 500 µl TNE buffer (50 mM Tris-HCL pH 8.0, 150 mM NaCl, 0.1% Igepal CA630, 1 mM EDTA) supplemented with 2 mM $MgCl_2$, cOmplete inhibitor cocktail (Roche), phosphoSTOP (Roche), and 25 U/ml benzonase. Cell lysates were incubated for 5 minutes at room temperature and then centrifuged at 15,000g for 15 minutes. Then, 600 µg of cell lysate was incubated with 0.8 µg of rabbit anti-GFP antibody (Torrey Pines biolabs, TP401) for 3 hours at 4 °C. A 20-µl slurry of protein G-sepharose beads (GE Healthcare, 17-061801) was added per sample for a 1-hour incubation period at 4 °C. The beads were collected by centrifugation, washed five times in TNE buffer, and eluted by boiling in 10× SDS–PAGE loading buffer. Samples were subjected to SDS–PAGE and immunoblotting.

## Immunoblotting

Proteins were separated by standard SDS–PAGE and transferred onto PVDF membranes. Membranes were incubated for 1 minute at room temperature with Ponceau solution, washed three times with PBS-T

(PBS + 0.1% Tween-20) and then blocked with 5% milk in PBS-T for 2 hours at room temperature. Primary antibodies were incubated overnight at 4 °C in blocking solution. Membranes were then washed three times with PBS-T and incubated with HRP-conjugated secondary antibodies for 1 hour at room temperature. Membranes were washed again three times with PBS-T, and protein signals were detected using ECL Western Blotting Detection Reagent (Amersham) and acquired with Canon MP Navigator EX. Antibody information is provided in the Reporting Summary.

Additional methods are provided in the Supplementary Information.

### Statistics and reproducibility
No samples were measured repeatedly for statistical analysis. Two-tailed unpaired $t$-test, one-way analysis of variance (ANOVA) with Dunnett`s test, one-way ANOVA with Tukey test, or two-way ANOVA with Šidák's test was performed as indicated in the figure legends using GraphPad Prism (Versions 5, 8 and 9). Moderate $t$-test[68] was calculated using R package limma[69]. Sample sizes and statistical tests used are specified in the figure legends. All experimental findings were confirmed by independent repetitions. Data shown in Figures 1b–e, 2a–h, 3a–d, 4a–i, 5b, and 6a–i, Extended Data Figures 1a, 2c, 4a,b,i, 5a,b,f–h, 6i,j, 8a–j, 9a,b,g,i,j, and 10a–c,f, Supplementary Figures 1b,c, 2a, 3a–d, and 4a–d,f–i were confirmed in at least three independent experiments; data shown in Extended Data Figures 1b,c, 2b,d,e, 3b,c, 4c–h,j, 5c–e,i, 6b–h, 7a–c, 9c–f,h, 10d,e and Supplementary Figures 1d–g, 2b, 3e, and 4e were confirmed in at least two independent experiments. Label-free proximity labeling mass spectrometry was performed once with six technical replicates.

### Reporting summary
Further information on research design is available in the Nature Portfolio Reporting Summary linked to this article.

### Data availability
The mass spectrometry proteomics data were analyzed using Homo Sapiens UniProt reference proteome database (taxonomy 9606; canonical version from 20190709), reversed decoy-database, and database of common protein contaminants. The mass spectrometry proteomics data have been deposited together with the reversed decoy-database and the database of common protein contaminants to the ProteomeXchange Consortium via the PRIDE partner repository with the dataset identifier PXD036935. All other data are available from the corresponding author upon request. Source data are provided with this paper.

### Code availability
No custom code was used in this study.

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

### Acknowledgements
We acknowledge the Functional Genomics Center Zurich, the Center for Microscopy and Image Analysis, and the Flow Cytometry Facility of the University of Zurich for support. We would like to thank T. Halazonetis (University of Geneva, Switzerland), A. Constantinou (University of Montpellier, France), K. Gari (Zurich University of Applied Sciences, Switzerland), A. Sartori (University of Zurich, Switzerland), M. Hottiger (University of Zurich, Switzerland), and L. Toledo (University of Copenhagen, Denmark) for providing reagents, and C. Ambrosi, I. Karemaker, and T. Baubec for technical advice. This work was supported by the Swiss National Science Foundation (PP00P3_179057 & 310030_197003 to M.A. and 310030_205199 to P.C.) and the European Research Council (ERC) under the European Union's Horizon 2020 research and innovation program (ERC-2016-STG 714326 to M.A.). V.S. and A.P. received additional support from UZH Candoc and Postdoc grants.

### Author contributions
V.S. initiated the project, designed, and conducted most of the experiments, analyzed, interpreted, and visualized results, and developed the study. A.P. contributed to QIBC and live-cell experiments and data analysis. M.S. and A.K. contributed to data analysis. G.R. and P.C. expressed and purified RAD52 protein. R.I. contributed to cloning. B.R. performed LC/MS measurements and proteomics data analysis. M.A. conceived and supervised the study. V.S. and M.A. wrote the manuscript with input from all authors.

### Funding

### Competing interests
The authors declare no competing interests.

### Additional information
**Extended data** is available for this paper at https://doi.org/10.1038/s41594-023-00932-w.

**Correspondence and requests for materials** should be addressed to Matthias Altmeyer.

**Peer review information** *Nature Structural & Molecular Biology* thanks Jaewon Min and the other, anonymous, reviewer(s) for their contribution to the peer review of this work. Beth Moorefield, Carolina Perdigoto, Dimitris Typas, and Florian Ullrich were the primary editors on this article and managed its editorial process and peer review in collaboration with the rest of the editorial team. Peer reviewer reports are available.

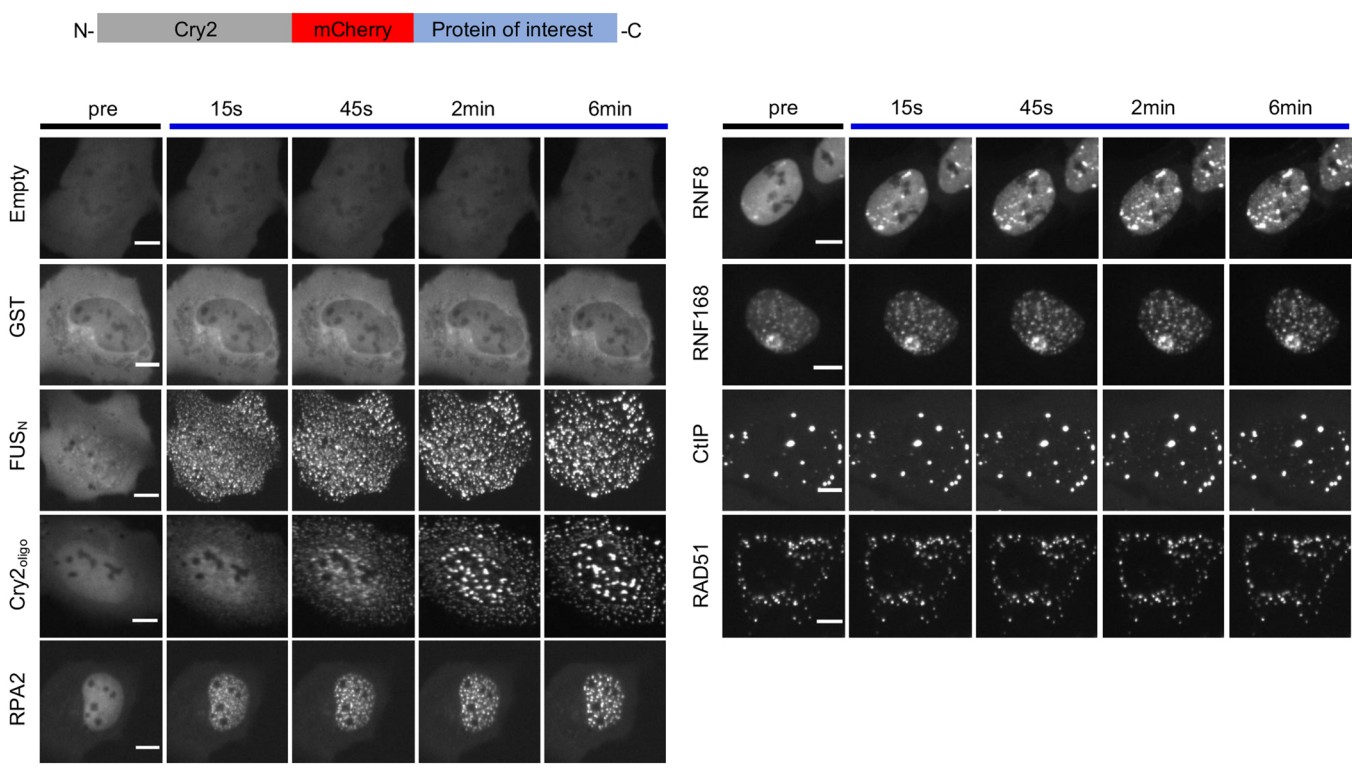

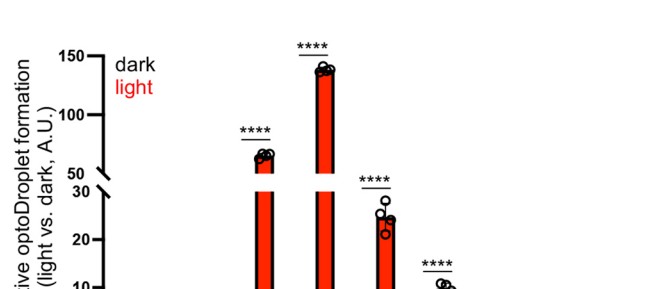

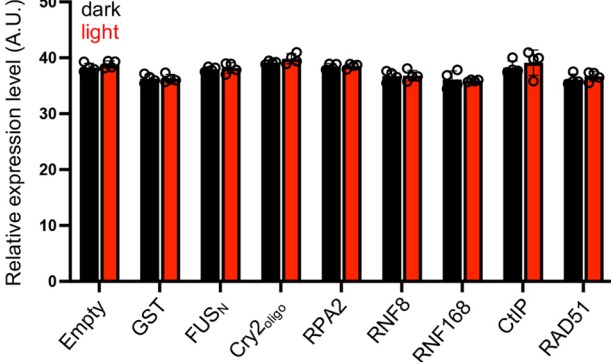

**Extended Data Fig. 1 | RPA2 forms dynamic intracellular optoDroplets. a**, Comparison of blue light-inducible optoDroplet formation by a panel of DNA damage response proteins. FUS_N fused to Cry2-mCherry and Cry2-mCherry E490A (Cry2olig) served as positive controls, GST fused to Cry2-mCherry and the empty Cry2-mCherry plasmid were included as negative controls. Representative stills from live-cell microscopy are shown. **b**, Accumulated optoDroplet intensity per nucleus of Cry2-mCherry constructs shown in (**a**) were analyzed and normalized to the average accumulated optoDroplet intensity of the corresponding dark condition. Two-way ANOVA with Šidák`s test, FUS_N, Cry2_oligo, RPA2, RNF8 **** $p < 0.0001$. **c**, Nuclear mean intensities of Cry2-mCherry in cells analyzed in (**b**). **b-c**, Averages and standard deviations are shown for $n = 4$

independent samples (cell number: emtpy_dark $n_1 = 604$, $n_2 = 834$, $n_3 = 785$, $n_4 = 510$; empty_light $n_1 = 587$, $n_2 = 696$, $n_3 = 781$, $n_4 = 606$; GST_dark $n_1 = 952$, $n_2 = 966$, $n_3 = 726$, $n_4 = 727$; GST_light $n_1 = 871$, $n_2 = 924$, $n_3 = 1015$, $n_4 = 863$; FUS_N dark $n_1 = 515$, $n_2 = 652$, $n_3 = 608$, $n_4 = 571$; FUS_N light $n_1 = 750$, $n_2 = 613$, $n_3 = 787$, $n_4 = 677$; Cry2_oligo dark $n_1 = 720$, $n_2 = 941$, $n_3 = 753$, $n_4 = 828$; Cry2_oligo light $n_1 = 506$, $n_2 = 889$, $n_3 = 766$, $n_4 = 952$; RPA2_dark $n_1 = 752$, $n_2 = 864$, $n_3 = 738$, $n_4 = 647$; RPA2_light $n_1 = 811$, $n_2 = 666$, $n_3 = 965$, $n_4 = 1028$; RNF8_dark $n_1 = 323$, $n_2 = 437$, $n_3 = 269$, $n_4 = 493$; RNF8_light $n_1 = 235$, $n_2 = 319$, $n_3 = 356$, $n_4 = 385$; RNF168_dark $n_1 = 459$, $n_2 = 278$, $n_3 = 532$, $n_4 = 334$; RNF168_light $n_1 = 369$, $n_2 = 506$, $n_3 = 395$, $n_4 = 424$; CtIP_dark $n_1 = 286$, $n_2 = 217$, $n_3 = 214$, $n_4 = 246$; CtIP_light $n_1 = 164$, $n_2 = 213$, $n_3 = 302$, $n_4 = 262$; RAD51_dark $n_1 = 305$, $n_2 = 245$, $n_3 = 275$, $n_4 = 124$; RAD51_light $n_1 = 256$, $n_2 = 214$, $n_3 = 243$, $n_4 = 177$). Scale bars 10 μm.

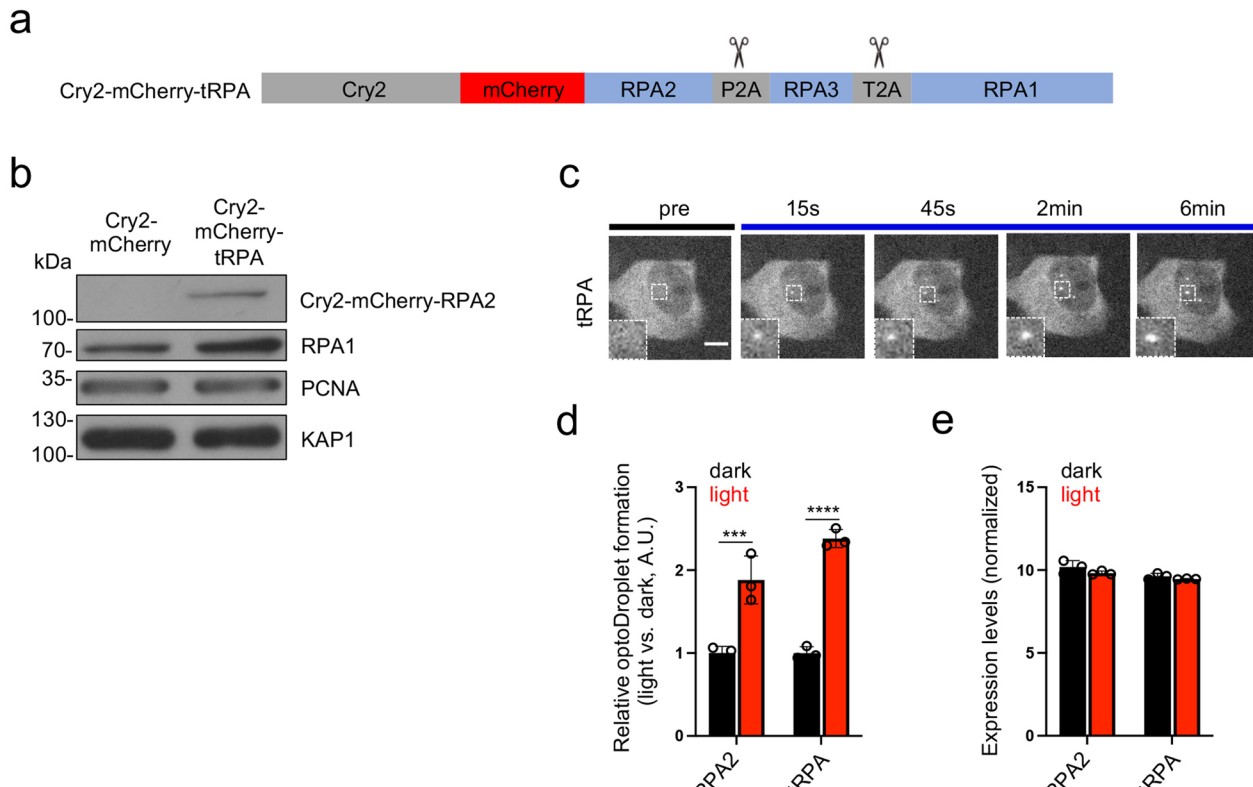

**Extended Data Fig. 2 | The trimeric RPA complex forms dynamic intracellular optoDroplets. a**, Schematic of the polycistronic Cry2-mCherry-tRPA optoDroplet construct possessing P2A and T2A cleavage sites for stoichiometric expression of all three RPA subunits. **b**, Western blot analysis of U-2 OS cells transiently transfected with Cry2-mCherry or Cry2-mCherry-tRPA as indicated. **c**, Time-resolved optoDroplet formation of trimeric RPA. Representative stills from live-cell microscopy are shown. **d**, Accumulated optoDroplet intensity per nucleus of trimeric RPA or RPA2 fused to Cry2-mCherry was analyzed and normalized to the average accumulated optoDroplet intensity of the corresponding dark condition. Two-way ANOVA with Šidák`s test, tRPA **** $p < 0.0001$, RPA2 *** $p = 0.0003$. **e**, Nuclear mean intensities of Cry2-mCherry in cells analyzed in (**d**). **d-e**, Averages and standard deviations for n = 3 independent samples are shown (cell number: RPA2$_{dark}$ $n_1 = 429$, $n_2 = 507$, $n_3 = 433$; RPA2$_{light}$ $n_1 = 361$, $n_2 = 452$, $n_3 = 411$; tRPA$_{dark}$ $n_1 = 305$, $n_2 = 412$, $n_3 = 304$; tRPA$_{light}$ $n_1 = 498$, $n_2 = 589$, $n_3 = 469$). Scale bars 10 µm.

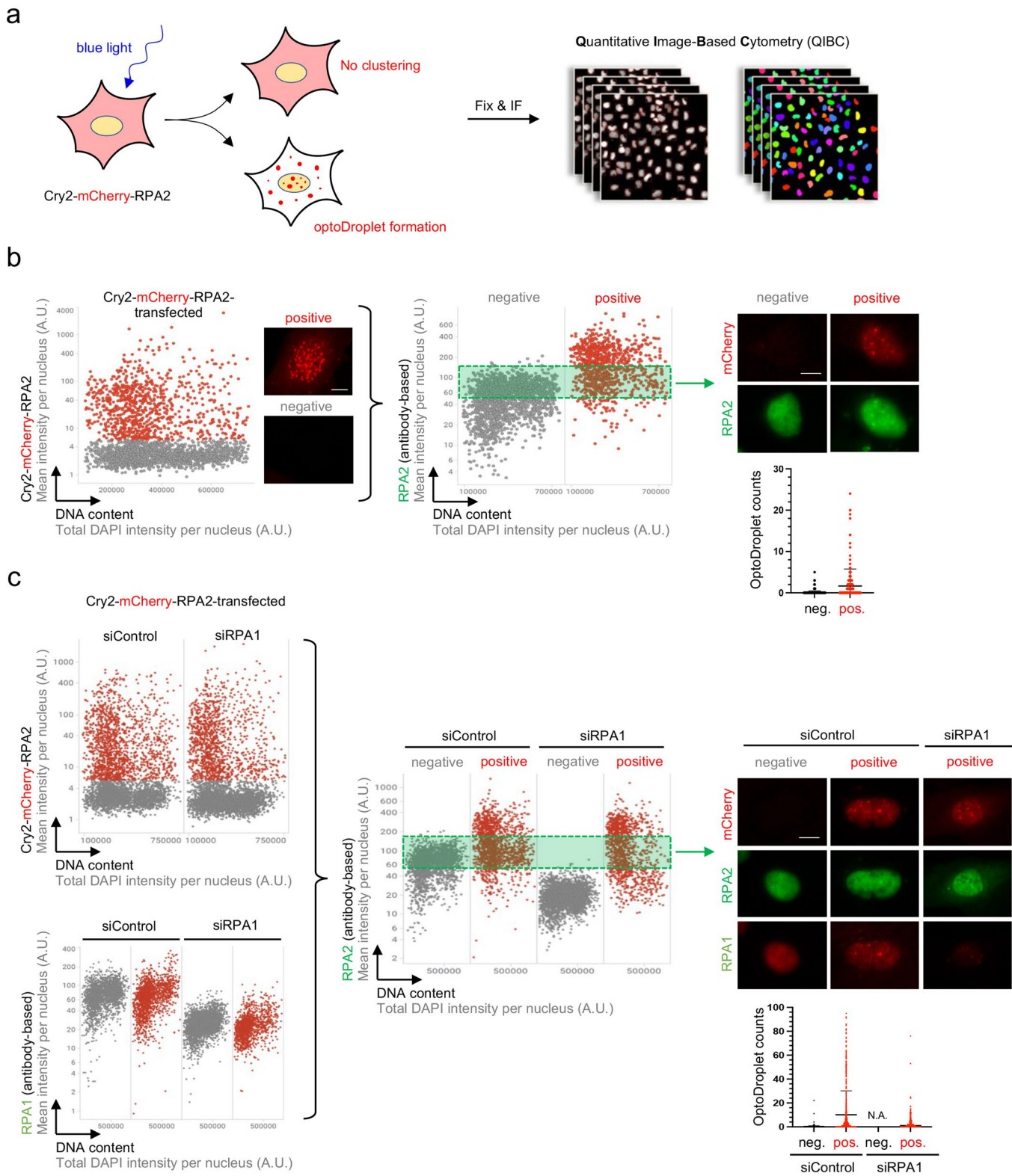

**Extended Data Fig. 3 | RPA2 optoDroplet formation occurs at endogenous RPA2 levels. a**, Schematic of the experimental pipeline used for QIBC analyses of blue light-induced clustering of Cry2-mCherry-RPA2. **b**, U-2 OS cells transfected with Cry2-mCherry-RPA2 were exposed to blue light and RPA2 levels analyzed by QIBC. OptoDroplet formation analyzed in Cry2-mCherry-RPA2 positive cells (n = 329 cells) was compared to Cry2-mCherry-RPA2 negative cells (n = 628 cells) of the same cell population with comparable RPA2 levels based on RPA2 staining (indicated by green box). **c**, U-2 OS cells were depleted of endogenous RPA1 prior to transfection with Cry2-mCherry-RPA2. Cells were exposed to blue light, fixed, and stained for RPA1 and RPA2. OptoDroplet formation was analyzed in siRPA1-transfected Cry2-mCherry-RPA2 positive cells with RPA2 levels comparable to endogenous (indicated by green box) in the control condition (siControl_negative n = 953 cells; siControl_positive n = 1470 cells; siRPA1_positive n = 788 cells). Scale bars 10 μm.

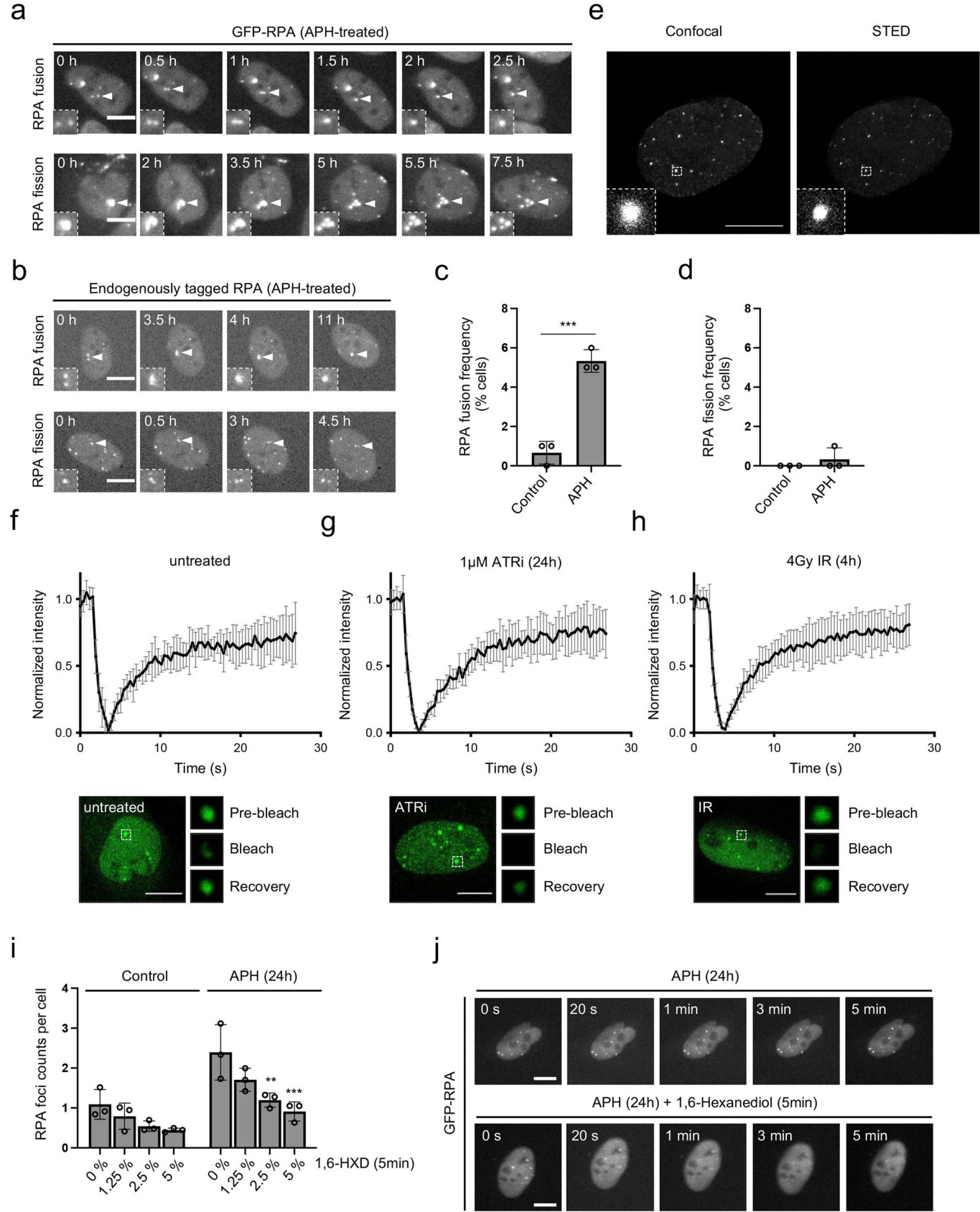

**Extended Data Fig. 4 | See next page for caption.**

**Extended Data Fig. 4 | Nuclear RPA condensates formed in response to genotoxic stress possess liquid-like properties. a**, U-2 OS GFP-RPA cells were treated with 0.2 µM APH for 24 h and analyzed by live-cell microscopy at 30 min intervals. Foci fusion and fission events are shown. **b**, U-2 OS cells expressing endogenously tagged mScarlet-RPA1 were treated with 0.2 µM APH and analyzed by live-cell microscopy at 20 min intervals. Foci fusion and fission events are shown. **c**, RPA foci fusion frequency from n = 3 independent 48 h time-lapse microscopy experiments. Averages and standard deviations from 100 cells per condition and replicate. Unpaired two-tailed t-test, *** p = 0.0006. **d**, RPA foci fission frequency analyzed as in (**c**). **e**, U-2 OS GFP-RPA2 cells, depleted for endogenous RPA2, were treated with 0.2 µM APH for 24 h and RPA ensembles were visualized by STED microscopy. **f**, FRAP analysis of RPA foci at endogenous DNA lesions in U-2 OS GFP-RPA cells. **g**, FRAP analysis as in (**f**) of RPA foci at ATR inhibitor (ATRi, 1 µM for 24 h) induced DNA lesions. **h**, FRAP analysis as in (f) of RPA foci at ionizing radiation (IR, 4 Gy with 4 h recovery) induced DNA lesions. **f-h**, Averages and standard deviations for 10–18 cells per condition. Half-recovery times 8–10 seconds. **i**, U-2 OS GFP-RPA cells were treated with 1,6-hexanediol in unchallenged conditions or after treatment with 0.2 µM APH for 24 h. RPA foci were analyzed by QIBC. Averages and standard deviation from n = 3 independent samples are shown (>100 cells for $APH_{5\%}$ and >1000 for all other samples). Two-way ANOVA with Šidák`s test, $APH_{5\%}$ *** p = 0.0004, $APH_{2.5\%}$ ** p = 0.0031. **j**, U-2 OS GFP-RPA cells were treated with 0.2 µM APH for 24 h followed by live-cell microscopy at 20 s intervals with and without addition of 2.5% 1,6-hexanediol. Representative stills are shown. Scale bars 10 µm.

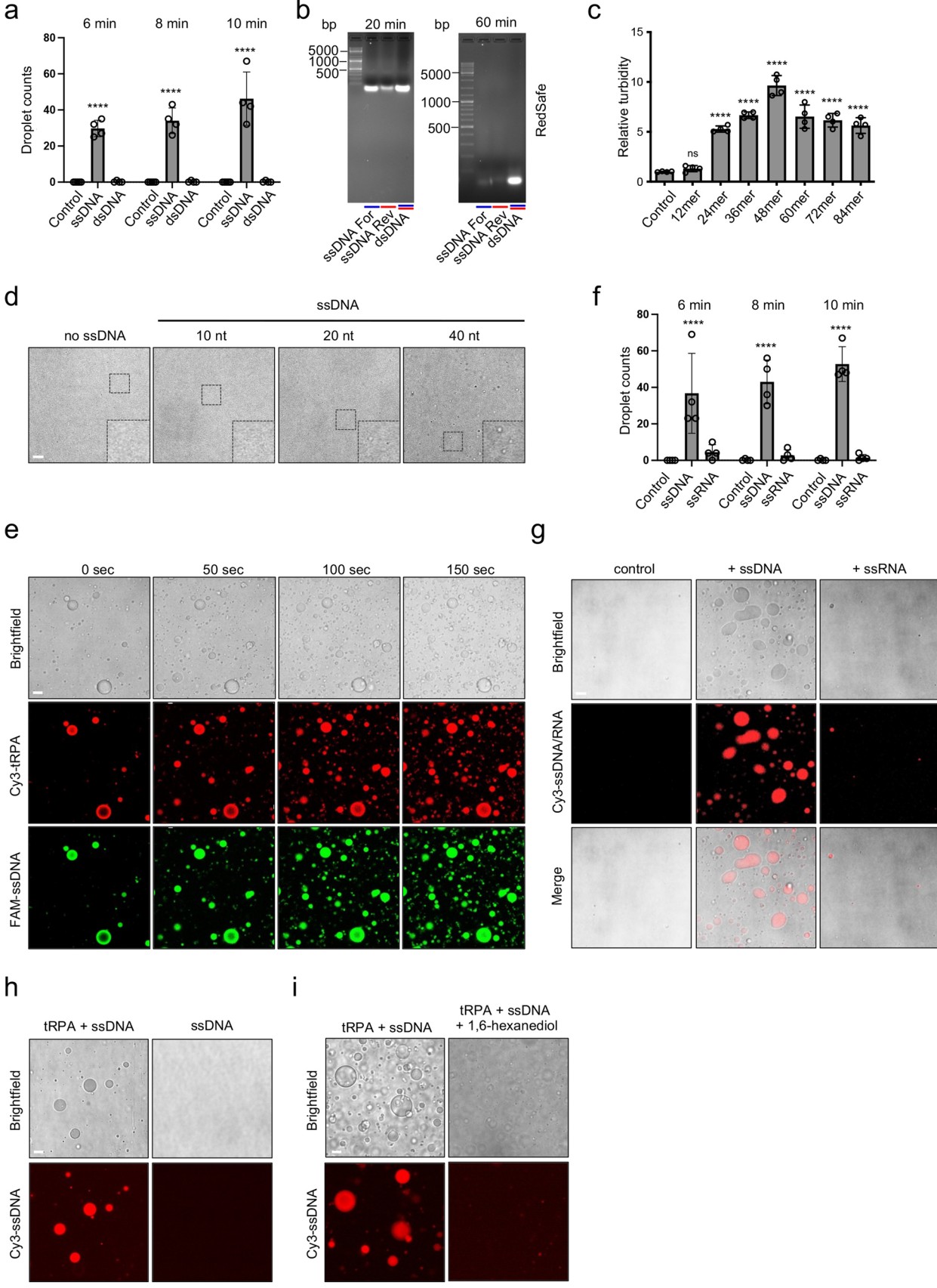

**Extended Data Fig. 5 | See next page for caption.**

**Extended Data Fig. 5 | ssDNA seeds RPA phase separation. a**, RPA droplet counts in the absence or presence of equimolar 40mer ssDNA or equimolar sequence-matched annealed 40mer dsDNA after different incubation periods of the samples. Averages and standard deviations from n = 4 replicates are shown. Two-way ANOVA with Šidák`s test compared to corresponding control, $6min_{ssDNA}$, $8min_{ssDNA}$, $10min_{ssDNA}$ **** $p < 0.0001$. **b**, 2% agarose gel electrophoresis of the 40mer ssDNA and the annealed 40mer dsDNA, detected by RedSafe at short and long running times. **c**, Turbidity measurements of purified trimeric RPA incubated with sub-stochiometric molar amounts of ssDNA (1:6 molar ratio ssDNA:RPA) from 12 to 84 nucleotides in length. Turbidity measurements were performed and normalized to the control. Averages and standard deviations from n = 4 replicates are shown. One-way ANOVA with Dunnett`s test compared to control, 24mer, 36mer, 48mer, 60mer, 72mer, 84mer **** $p < 0.0001$, 12mer ns $p = 0.9998$. **d**, In vitro droplet formation of purified trimeric RPA incubated with equimolar amount of ssDNA of the indicated lengths. **e**, Co-assembly of ssDNA into RPA droplets. Cy3-labeled purified trimeric RPA was incubated with equimolar amount of 40mer FAM-labeled ssDNA. Representative stills from time-lapse microscopy are shown. **f**, RPA droplet counts in the absence or presence of equimolar 40mer ssDNA or equimolar sequence-matched 40mer RNA after different incubation periods of the samples. Averages and standard deviations from n = 4 replicates are shown. Two-way ANOVA with Šidák`s test compared to corresponding control, $6min_{ssDNA}$ **** $p < 0.0001$, $8min_{ssDNA}$ **** $p < 0.0001$, $10min_{ssDNA}$ **** $p < 0.0001$. **g**, Purified trimeric RPA was incubated with equimolar Cy3-labeled 40mer ssDNA or equimolar sequence-matched Cy3-labeled ssRNA as indicated. Representative images of RPA droplets are shown. **h**, Purified trimeric RPA was incubated with equimolar Cy3-labeled 40mer ssDNA, with Cy3-labeled 40mer ssDNA without trimeric RPA serving as control. Representative images of RPA droplets are shown. **i**, Purified trimeric RPA was incubated with equimolar Cy3-labeled 40mer ssDNA without and with addition of 5% 1,6-hexanediol as indicated. Representative images of RPA droplets are shown. Scale bars 10 μm.

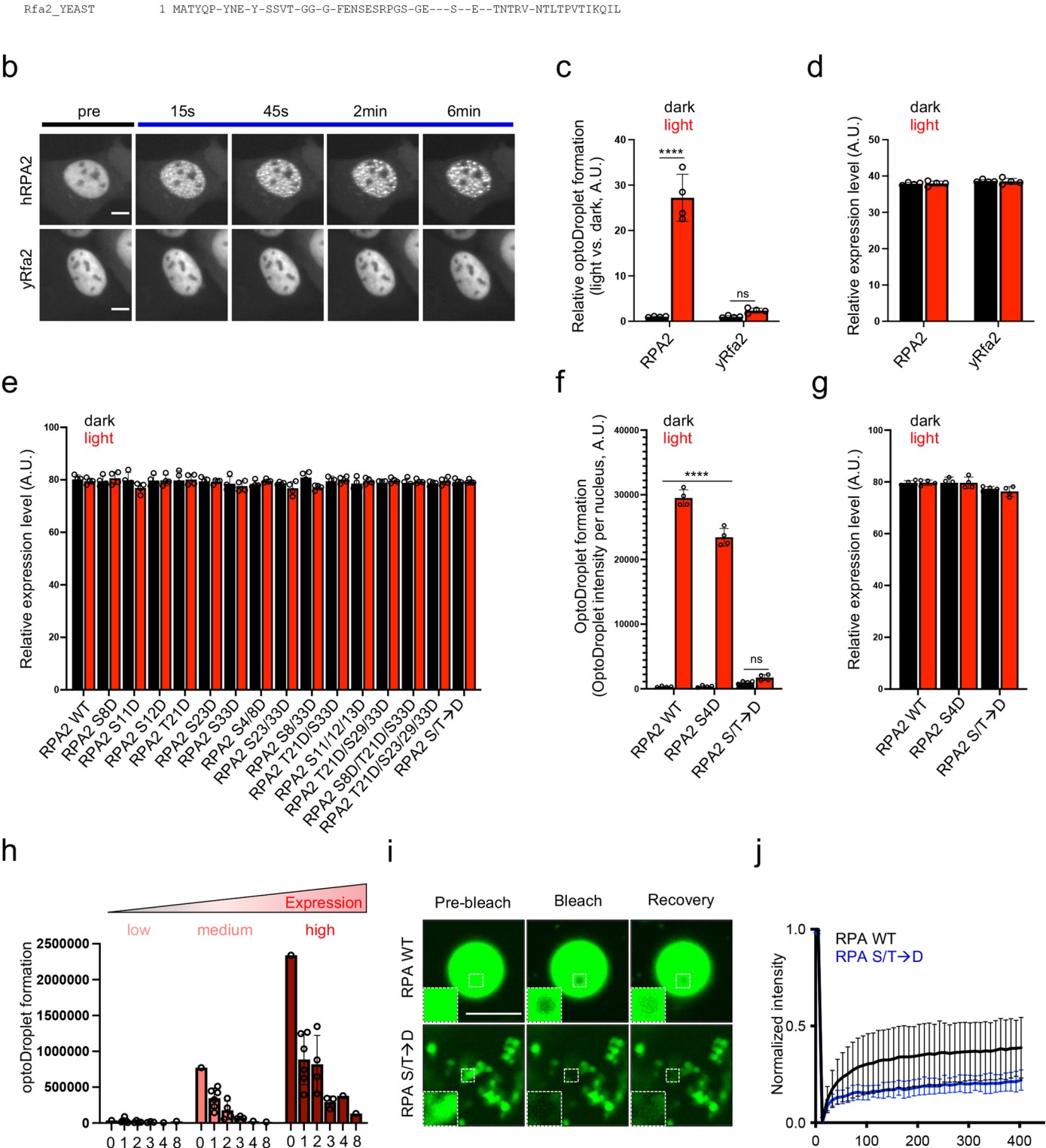

a

```
RPA2_HUMAN   1 M--WNSGF-ESYGSS-SYGGAGGY--TQS-PGGFGSPAPSQAEKKSRARAQH-IVPCTISQLL
               |  :..  : |  | || : || | :  ::| ||. |.   | |  ..|. : :.|.|.:|
Rfa2_YEAST   1 MATYQP-YNE-Y-SSVT-GG-G-FENSESRPGS-GE---S--E--TNTRV-NTLTPVTIKQIL
```

**Extended Data Fig. 6 | See next page for caption.**

**Extended Data Fig. 6 | Phosphorylation of the N-IDR of RPA2 regulates RPA phase separation. a**, Pairwise sequence alignment of the N-IDR of human RPA2 (hRPA2) with the *S. cerevisiae* homolog Rfa2 (yRfa2). **b**, Time-resolved optoDroplet formation of hRPA2 and yRfa2 fused to Cry2-mCherry. Representative stills from live-cell microscopy are shown. **c**, Accumulated optoDroplet intensity per nucleus of hRPA2 and yRfa2 fused to Cry2-mCherry was analyzed and normalized to the average accumulated optoDroplet intensity of the corresponding dark condition. Two-way ANOVA with Šidák`s test, RPA2 **** p < 0.0001, yRfa2 ns p = 0.8540. **d**, Nuclear mean intensities of Cry2-mCherry in cells analyzed in (**c**). **c-d**, Averages and standard deviations for n = 4 independent samples (>500 cells per sample). **e**, Nuclear mean intensities of Cry2-mCherry in cells analyzed in Fig. 4d. Averages and standard deviations are shown for n = 4 independent samples (>15 cells per sample, average 61 cells per sample). **f**, Accumulated optoDroplet intensity per nucleus of the indicated RPA2 mutants fused to Cry2-mCherry was analyzed. Two-way ANOVA with Šidák`s test, RPA2

WT, RPA2 S4D **** p < 0.0001; RPA2 S/T→D ns p = 0.7357. **g**, Nuclear mean intensities of Cry2-mCherry in cells analyzed in (**f**). **f-g**, Averages and standard deviation of n = 4 independent samples are shown (>80 cells per sample, average 192 cells per sample). **h**, Quantification of Cry2-mCherry-RPA2 optoDroplet formation of wildtype, single (S8D, S11D, S12D, T21D, S23D, S33D), double (S4/S8D, S23/S33D, S8/S33D, T21D/S33D), triple (S11/S12/S13D, T21D/S29/S33D, S8D/T21D/S33D), quadruple (T21D/S23/S29/S33D), and octuple S/T→D (S8/S11/S12/S13/T21/S23/S29/S33D) mutants. Cry2-mCherry expression was grouped from low to high. Single data points represent individual Cry2-mCherry-RPA2 mutants (>80 cells per sample, average 111 cells per sample). **i**, FRAP analysis of 40mer FAM-labelled ssDNA in phase separated RPA WT droplets and RPA S/T→D aggregates, respectively. **j**, Quantification of ssDNA FRAP experiments shown in (**i**). Average and standard deviation of n = 15 structures per condition are shown. Scale bars 10 μm.

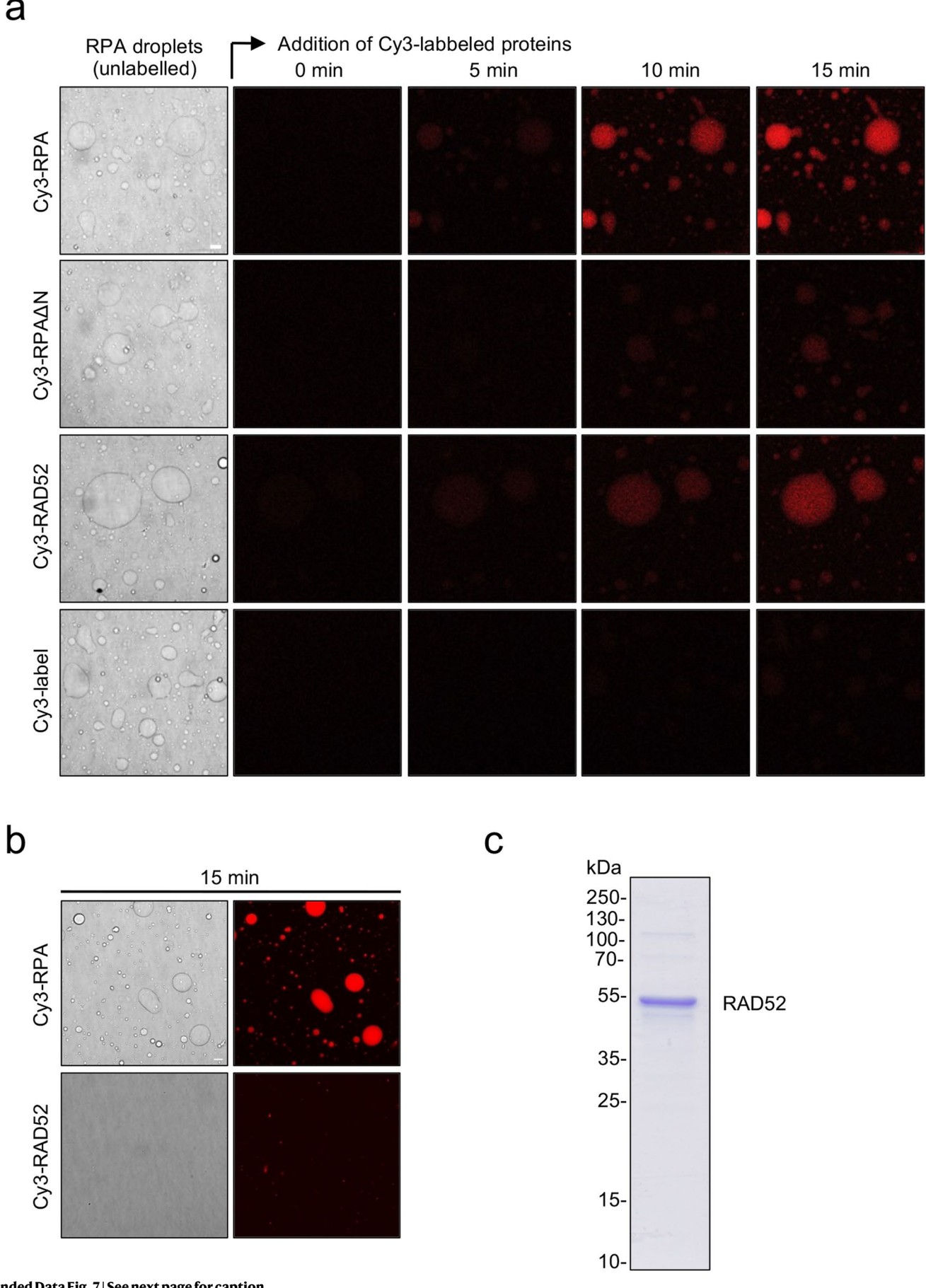

**Extended Data Fig. 7 | See next page for caption.**

**Extended Data Fig. 7 | RPA condensates concentrate RAD52 in vitro. a**, Purified RAD52 partitions into RPA droplets in vitro. Purified trimeric unlabeled RPA was allowed to form liquid droplets prior to incubation with Cy3-labeled purified RAD52. Time-resolved partitioning of RAD52 into RPA droplets was followed by time-lapse microscopy. Cy3-labeled RPA served as positive control, confirming homotypic RPA interactions in phase separated RPA droplets. The Cy3 label added to preformed RPA droplets served as negative control. Cy3-labeled RPAΔN, lacking the N-IDR of RPA2, served as additional negative control and confirmed the involvement of the N-IDR in homotypic RPA interactions. Representative stills from time-lapse microscopy are shown. **b**, Purified RAD52 alone is not forming phase separated droplets in the conditions used in (**a**). **c**, Coomassie staining of the purified human RAD52 after SDS-PAGE. Scale bars 10 μm.

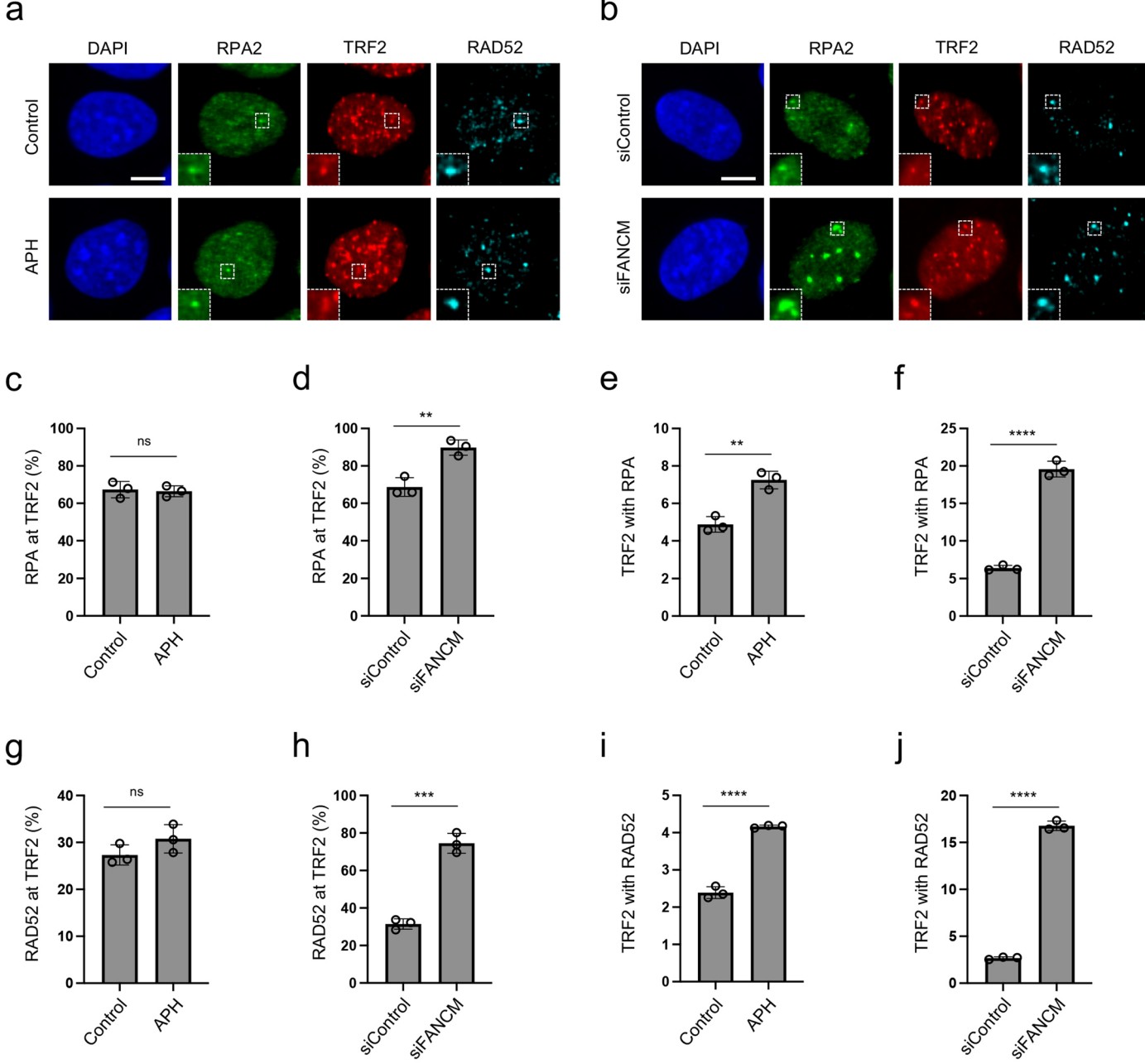

**Extended Data Fig. 8 | Localization of RPA and RAD52 at telomeres. a**, Co-localization of RPA2 and RAD52 at TRF2-marked telomeres in U-2 OS cells in unchallenged conditions and after treatment with 0.2 µM aphidicolin for 24 h. **b**, Enhanced co-localization of RPA2 and RAD52 at TRF2-marked telomeres in U-2 OS cells upon depletion of FANCM. **c**, Quantification of RPA at telomeres. Co-localization of RPA with TRF2-marked telomeres in U-2 OS cells in unchallenged conditions and after treatment with 0.2 µM aphidicolin for 24 h. Two-tailed unpaired t-test, ns p = 0.7898. **d**, As in (**c**) in U-2 OS cells transfected with negative control siRNA or siRNA against FANCM. Two-tailed unpaired t-test, ** p = 0.0049. **e**, Percentage of TRF2 foci co-localizing with RPA in cells analyzed in (**c**). Two-tailed unpaired t-test, ** p = 0.0028. **f**, Percentage of TRF2 foci co-localizing with RPA in cells analyzed in (**d**). Two-tailed unpaired t-test, **** p < 0.0001. **g**, Percentage of RAD52 foci co-localizing with TRF2-marked telomeres in cells

treated as in (**c**). Two-tailed unpaired t-test, ns p = 0.1824. **h**, Percentage of RAD52 foci co-localizing with TRF2-marked telomeres in cells treated as in (**d**). Two-tailed unpaired t-test, *** p = 0.0002. **i**, Percentage of TRF2 foci co-localizing with RAD52 in cells analyzed in (**g**). Two-tailed unpaired t-test, **** p < 0.0001. **j**, Percentage of TRF2 foci co-localizing with RAD52 in cells analyzed in (**h**). Two-tailed unpaired t-test, **** p < 0.0001. **c-f**, Averages and standard deviation from n = 3 independent samples with 50 analyzed cells per condition and replicate are shown. **g-j**, Averages and standard deviation from n = 3 independent samples with (**g,i**) $WT_{control}$ $n_1 = 92$, $n_2 = 96$, $n_3 = 107$; $WT_{APH}$ $n_1 = 80$, $n_2 = 73$, $n_3 = 101$; S/T→D$_{control}$ $n_1 = 149$, $n_2 = 108$, $n_3 = 90$; S/T→D$_{APH}$ $n_1 = 88$, $n_2 = 81$, $n_3 = 86$ (**h,j**) $WT_{siCON}$ $n_1 = 134$, $n_2 = 145$, $n_3 = 1115$; $WT_{siFANCM}$ $n_1 = 80$, $n_2 = 68$, $n_3 = 72$; S/T→D$_{siControl}$ $n_1 = 117$, $n_2 = 108$, $n_3 = 114$; S/T→D$_{siFANCM}$ $n_1 = 78$, $n_2 = 83$, $n_3 = 65$ cells are shown.

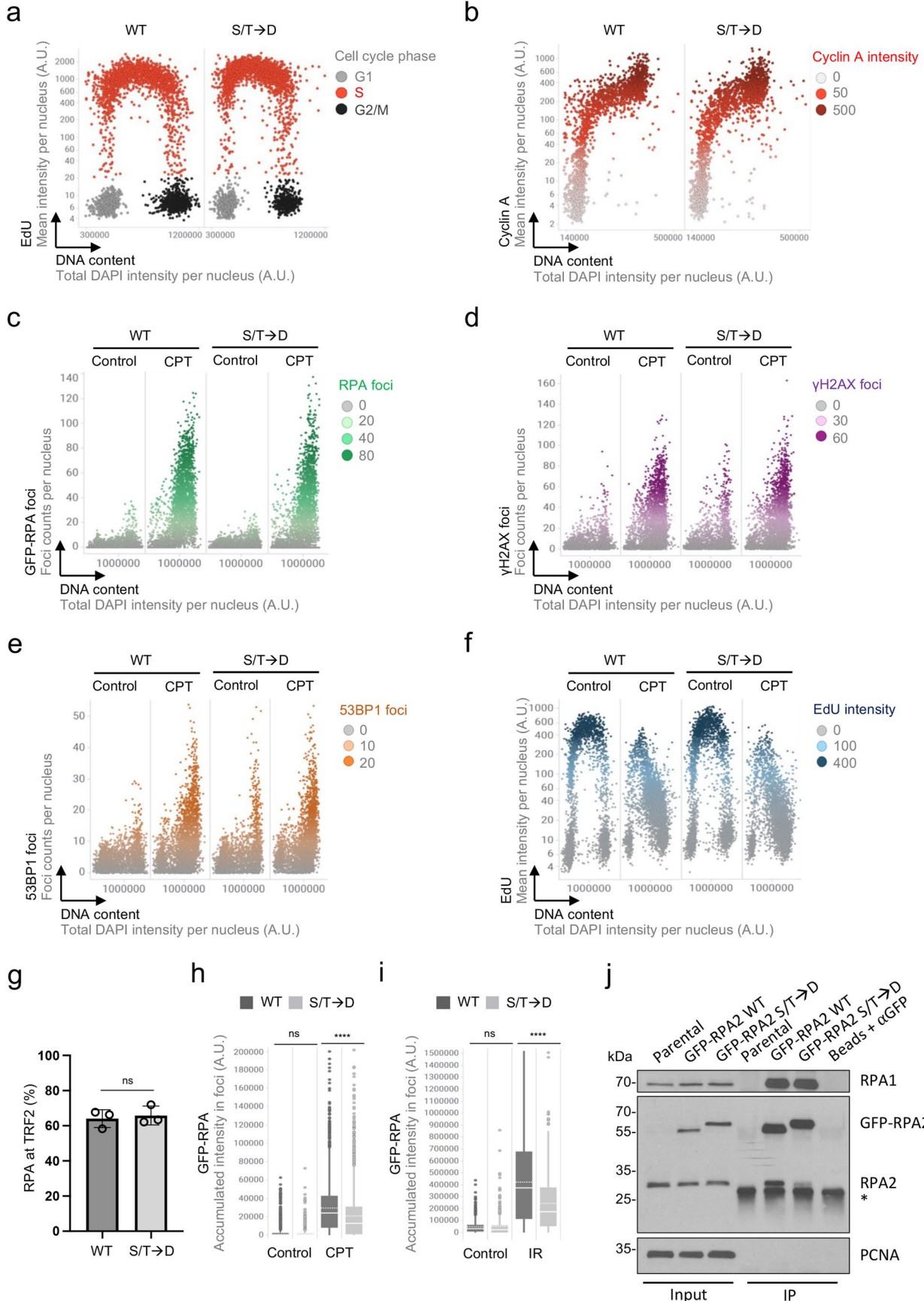

**Extended Data Fig. 9 | See next page for caption.**

**Extended Data Fig. 9 | Cell cycle and DNA damage response in siRNA-resistant GFP-RPA2 cell lines. a**, Cell cycle analysis by QIBC of RPA2-depleted U-2 OS cells stably expressing siRNA-resistant GFP-RPA2 WT or S/T→D . **b**, Cyclin A analysis by QIBC of cells treated as in (**a**) . **c**, Analysis of RPA recruitment to CPT-induced DNA damage (100 nM for 24 h) in RPA2-depleted U-2 OS cells stably expressing siRNA-resistant GFP-RPA2 WT or S/T→D, based on cell cycle resolved GFP-RPA2 measurements by QIBC. **d**, Analysis of γH2AX induction in cells treated as in (**c**) by QIBC. **e**, Analysis of 53BP1 recruitment to CPT-induced DNA damage in cells treated as in (**c**) by QIBC. **f**, Analysis of EdU profiles in cells treated as in (**c**) by QIBC. **a-f**, >2000 cells per sample. **g**, Analysis of RPA recruitment to telomeres in RPA2-depleted U-2 OS cells stably expressing siRNA-resistant GFP-RPA2 WT or S/T→D. Averages and standard deviations are shown for n = 3 independent samples with 50 cells analyzed per replicate. Two-tailed unpaired t-test, ns p = 0.7025. **h**, Analysis of the accumulated RPA intensity in CPT-induced nuclear foci (100 nM CPT for 24 h) in RPA2-depleted U-2 OS cells stably expressing siRNA-resistant GFP-RPA2 WT or S/T→D, based on cell cycle resolved GFP-RPA2 measurements by QIBC. One-way ANOVA with Tukey test, CPT **** p < 0.0001, control ns p = 0.4799. **i**, Analysis of the accumulated RPA intensity in IR-induced nuclear foci (10 Gy, 8 hours release) in RPA2-depleted U-2 OS cells stably expressing siRNA-resistant GFP-RPA2 WT or S/T→D, based on cell cycle resolved GFP-RPA2 measurements by QIBC. One-way ANOVA with Tukey test, IR **** p < 0.0001, control ns p > 0.9999. **h-i**, Box plots with medians (solid lines) and means (dashed lines). Box limits indicate 25th percentile (Q1) and 75th percentile (Q3); box represents interquartile range (IQR, Q3-Q1). Whiskers define lower and upper adjacent value; dots show outliers greater than Q3 + 1.5xIQR. >750 cells per sample. **(j)** GFP co-IP from parental U-2 OS cells and from U-2 OS cells stably expressing GFP-RPA2 WT or S/T→D to probe for RPA2 self-interaction. Endogenous RPA2 and GFP-RPA2 were detected with an anti-RPA2 antibody. *, unspecific band.

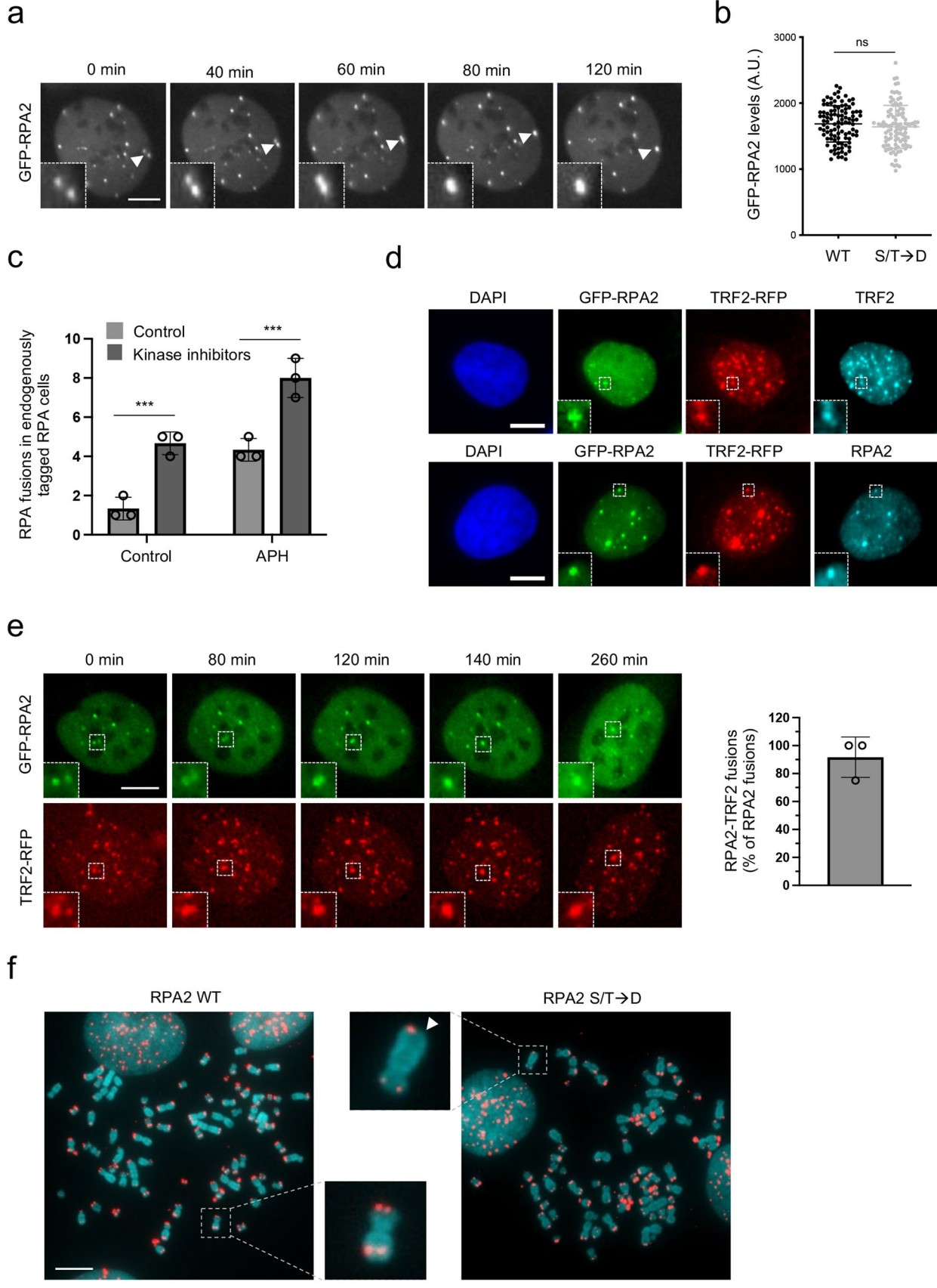

**Extended Data Fig. 10 | See next page for caption.**

**Extended Data Fig. 10 | RPA condensation is linked to telomere clustering.**
**a**, Example of RPA foci fusion event in U-2 OS cells stably expressing siRNA-resistant GFP-RPA2, depleted for endogenous RPA2, and treated with a 0.2 µM of aphidicolin for 48 h. **b**, GFP-RPA2 expression levels in the cells analyzed for RPA fusion events in Fig. 6h. Single cell measurements of n = 100 cells per condition, averages and standard deviation are shown. Two-tailed unpaired t-test was performed, ns p = 0.2815. **c**, RPA foci fusion frequency was analyzed in U-2 OS cells expressing endogenously tagged mScarlet-RPA1 from n = 3 independent 48 h time-lapse microscopy experiments with 100 cells per replicate. Cells were treated with 0.2 µM aphidicolin and kinase inhibitors (ATM/ATR/DNA-PK, CDK1/2) as indicated. Averages and standard deviation are shown. Two-way ANOVA with Šidák`s test, control *** p = 0.0008, APH *** p = 0.0004. **d**, RPA and TRF2 signals in a stable U-2 OS cell line expressing GFP-RPA2 and TRF2-RFP. Cells were stained for TRF2 (top panel) or RPA2 (lower panel) in Cy5 to confirm correct localization of GFP-RPA2 to RPA foci and TRF2-RFP to TRF2 foci, respectively. Representative images are shown. **e**, RPA foci fusions coincide with telomere fusions marked by TRF2-RFP in U-2 OS cells stably expressing GFP-RPA2 and TRF2-RFP, treated with 0.2 µM of aphidicolin for 36 h. The fraction of RPA foci fusions coinciding with TRF2 foci fusions is indicated to the right. Averages and standard deviation from n = 3 independent samples with 4 fusion events observed per replicate are shown. **f**, Representative metaphase spreads used to quantify telomere loss by metaphase telomere FISH analysis in Fig. 6i. Telomere signal in red, DNA signal in turquoise. Scale bars 10 µm.

# Reporting Summary

## Statistics

For all statistical analyses, confirm that the following items are present in the figure legend, table legend, main text, or Methods section.

| n/a | Confirmed | |
|---|---|---|
| ☐ | ☒ | The exact sample size (*n*) for each experimental group/condition, given as a discrete number and unit of measurement |
| ☐ | ☒ | A statement on whether measurements were taken from distinct samples or whether the same sample was measured repeatedly |
| ☐ | ☒ | The statistical test(s) used AND whether they are one- or two-sided<br>*Only common tests should be described solely by name; describe more complex techniques in the Methods section.* |
| ☒ | ☐ | A description of all covariates tested |
| ☒ | ☐ | A description of any assumptions or corrections, such as tests of normality and adjustment for multiple comparisons |
| ☐ | ☒ | A full description of the statistical parameters including central tendency (e.g. means) or other basic estimates (e.g. regression coefficient) AND variation (e.g. standard deviation) or associated estimates of uncertainty (e.g. confidence intervals) |
| ☐ | ☒ | For null hypothesis testing, the test statistic (e.g. *F*, *t*, *r*) with confidence intervals, effect sizes, degrees of freedom and *P* value noted<br>*Give P values as exact values whenever suitable.* |
| ☒ | ☐ | For Bayesian analysis, information on the choice of priors and Markov chain Monte Carlo settings |
| ☒ | ☐ | For hierarchical and complex designs, identification of the appropriate level for tests and full reporting of outcomes |
| ☒ | ☐ | Estimates of effect sizes (e.g. Cohen's *d*, Pearson's *r*), indicating how they were calculated |

*Our web collection on statistics for biologists contains articles on many of the points above.*

## Software and code

Policy information about availability of computer code

| Data collection | High-content microscopy: ScanR Image Acquisition 3.01 and 3.2, IN Cell Analyzer 2500 V7.4<br>Metaphase spreads: Leika Application Suite X 3.7.5.24914<br>FRAP: Leika Application Suite X 3.5.7.23225<br>STED imaging: Leika Application Suite X 3.5.7.23225<br>In vitro droplet assays: Leika Application Suite X 3.5.7.23225<br>Turbidity measurements:Tecan i-control 2.0<br>Western blot film developer: Canon MP Navigator EX<br>Agarose gel imaging: Infinity ST5 Xpress<br>qPCR: Qiagen Rotor-Gene Q Series Software 20.4.21 |
|---|---|
| Data analysis | Microscopy image analysis: Olympus ScanR Image Analysis Software (version 3.0.1 and version 3.2); Fiji/ImageJ 64-bit, Version 2.00-rc54/1.51h<br>Data visualization: TIBCO Spotfire data visualization software (version 7.9.1 and 10.10.1),  Graphpad Prism (Version 5, 8 and 9)<br>Statistical analysis: Graphpad Prism (Version 5, 8 and 9)<br>MS: MaxQuant (Version 1.6.2.3), R package SRM Service (https://github.com/protViz/SRMService) |

For manuscripts utilizing custom algorithms or software that are central to the research but not yet described in published literature, software must be made available to editors and reviewers. We strongly encourage code deposition in a community repository (e.g. GitHub). See the Nature Portfolio guidelines for submitting code & software for further information.

## Data

Policy information about availability of data

All manuscripts must include a data availability statement. This statement should provide the following information, where applicable:
- Accession codes, unique identifiers, or web links for publicly available datasets
- A description of any restrictions on data availability
- For clinical datasets or third party data, please ensure that the statement adheres to our policy

The mass spectrometry proteomics data was analyzed using Homo Sapiens UniProt reference proteome database (taxonomy 9606; canonical version from 20190709), reversed decoy-database, and database of common protein contaminants. The mass spectrometry proteomics data have been deposited together with the reversed decoy-database and the database of common protein contaminants to the ProteomeXchange Consortium via the PRIDE partner repository with the dataset identifier PXD036935. Other data are provided as source data file with this paper. No restrictions apply on data availability.

# Field-specific reporting

Please select the one below that is the best fit for your research. If you are not sure, read the appropriate sections before making your selection.

☒ Life sciences          ☐ Behavioural & social sciences          ☐ Ecological, evolutionary & environmental sciences

For a reference copy of the document with all sections, see nature.com/documents/nr-reporting-summary-flat.pdf

# Life sciences study design

All studies must disclose on these points even when the disclosure is negative.

| | |
|---|---|
| Sample size | This study did not include animal models or human participants and sample sizes were determined based on current standards in the field (e.g. Toledo et al., Cell 2013 Nov 21;155(5):1088-103; Michelena et al., Nat Commun. 2018 Jul 11;9(1):2678; Sedlackova et al., Nature, 2020 Oct 21; 587, 297-302; Lezaja et al., Nat Commun. 2021 Jun 22;12(1):3827). Exact sample sizes are provided in the methods and figure legends, and individual data points are provided in the Source Data files. |
| Data exclusions | No relevant data was excluded from this study. |
| Replication | Experiments were performed in at least 2-3 biological replicates and experimental findings were reliably reproduced. Turbo-ID proteomics was performed in six technical replicates for label-free quantification. |
| Randomization | Experiments were performed with asynchronously cycling cell populations and cultures serving as control or experimental groups, respectively, were randomly assigned. For each condition multiple non-overlapping fields of view using an evenly distributed standard grid were acquired. |
| Blinding | Data collection and analysis was conducted using automated unbiased image acquisition and analysis software. No further blinding was applied and no animals or human research participants or samples were involved in the study. |

# Reporting for specific materials, systems and methods

We require information from authors about some types of materials, experimental systems and methods used in many studies. Here, indicate whether each material, system or method listed is relevant to your study. If you are not sure if a list item applies to your research, read the appropriate section before selecting a response.

## Materials & experimental systems

| n/a | Involved in the study |
|---|---|
| ☐ | ☒ Antibodies |
| ☐ | ☒ Eukaryotic cell lines |
| ☒ | ☐ Palaeontology and archaeology |
| ☒ | ☐ Animals and other organisms |
| ☒ | ☐ Human research participants |
| ☒ | ☐ Clinical data |
| ☒ | ☐ Dual use research of concern |

## Methods

| n/a | Involved in the study |
|---|---|
| ☒ | ☐ ChIP-seq |
| ☒ | ☐ Flow cytometry |
| ☒ | ☐ MRI-based neuroimaging |

## Antibodies

| | |
|---|---|
| Antibodies used | Primary antibodies used in this study:<br>RPA70 (Abcam, ab79398, diluted 1:500 for IF and WB)<br>H2AX phospho S139 (Biolegend, 613401, diluted 1:1000 for IF) |

TRF2 (Novus, NB110-57130, diluted 1:300 for IF)
RAD52 (custom-made, kindly provided by Dr. Thanos Halazonetis, diluted 1:200 for IF)
RAD52 (Santa Cruz, sc-365341, diluted 1:100 for WB)
PML (Santa Cruz, sc-966, diluted 1:1000 for IF)
RPA32 (Abcam, ab2175, diluted 1:500 for IF and WB)
RPA32 pS4/8 (Bethyl, A300-245, diluted 1:500 for WB)
CyclinA (Abcam, ab16726, diluted 1:200 for IF)
53BP1 (Novus, NB100-304, diluted 1:1000 for IF)
PCNA (Santa Cruz, sc-56, diluted 1:2000 for WB)
KAP1 (Bethyl, A300-274A, diluted 1:1000 for WB)
GFP (Torrey Pines biolabs, TP401, 0.8ug/600ug whole cell lysate for IP)

Secondary antibodies used in this study:
Alexa Fluor 647 Goat Anti-Mouse (Life Technologies, A21325, diluted 1:500 for IF)
Alexa Fluor 647 Goat Anti-Rabbit (Life Technologies, A21244, diluted 1:500 for IF)
Alexa Fluor 568 Goat Anti-Mouse (Life Technologies, A11031, diluted 1:500 for IF)
Alexa Fluor 568 Goat Anti-Rabbit (Life Technologies, A11029, diluted 1:500 for IF)
Alexa Fluor 488 Goat Anti-Rabbit (Life Technologies, A11034, diluted 1:500 for IF)
Alexa Fluor 488 Goat Anti-Mouse (Life Technologies, A11029, diluted 1:500 for IF)
Donkey anti-Sheep IgG (H+L) Cross-Adsorbed Secondary AB Alexa 647 (Life Technologies, A21448, diluted 1:500 for IF)
Goat Anti-Rabbit IgG Antibody (H+L), Peroxidase (Vector Laboratories, PI-1000-1, diluted 1:10000 for WB)
Horse Anti-Mouse IgG Antibody (H+L), Peroxidase (Vector Laboratories, PI-2000-1, diluted 1:10000 for WB)

**Validation**

RPA70 (Abcam, ab79398) was previously validated by immunofluorescence staining, knockdown, and by endogenous RPA tagging (Toledo et al., Cell. 21,155(5):1088-103 (2013); Lezaja et al., Nat. Commun. 12, 3827 (2021)). Furthermore it was validated by microscopic analysis of co-localization with GFP-RPA2 signal in this study. H2AX phospho S139 (Biolegend, 613401) was previously validated by ATM inhibition (Lezaja et al., Nat. Commun. 12, 3827 (2021)). TRF2 (Novus, NB110-57130) was previously validated by knockdown and telomere FISH (Smogorzewska et al., Mol. Cell Biol. 20(5):1659-68 (2000); Lezaja et al., Nat. Commun. 12, 3827 (2021)). RAD52 (custom-made) was validated by immunofluorescence staining and RAD52 knockout (Sotiriou et al., Mol Cell, 64(6): 1127-1134 (2016); Lezaja et al., Nat. Commun. 12, 3827 (2021)). RAD52 (Santa Cruz, sc-365341) was validated by western blot by the manufacturer (https://datasheets.scbt.com/sc-365431) and by knockdown (van de Kooij et al., Nat Commun. 13, 5295 (2022)). PML (Santa Cruz, sc-966) was validated previously by knockdown and used for immunofluorescence (Vilotti et al., Cell Death Differ. 2012 Mar;19(3):488-500). RPA32 (Abcam, ab2175) was previously validated by knockdown (Mylers et al., Proc Natl Acad Sci U S A. 2016 Mar;113(9):E1170-9; Lezaja et al., Nat. Commun. 12, 3827 (2021)). RPA32 pS4/8 (Bethyl, A300-245) was previously validated by western blot (Zhang et al., Nat. Commun. 13, 6907 (2022)). Cyclin A (Abcam, ab16726) was previously validated by co-staining with other cell cycle markers and used for immunofluorescence (Lyman et al., PLoS One. 7;6(3):e17692 (2011); Moreno et al., Proc Natl Acad Sci U S A. 27;113(39):E5757-64 (2016); Somyajit et al., Science. 10;358(6364):797-802 (2017); Lezaja et al., Nat. Commun. 12, 3827 (2021)). 53BP1 (Novus, NB100-304) was previously validated by the manufacturer using gamma irradiation and immunofluorescence and by knockdown (Han et al., Sci Adv. 14;1(7):e1500454 (2015)) and further validated by assessing replication stress-induced 53BP1 nuclear bodies (Lezaja et al., Nat. Commun. 12, 3827 (2021)). PCNA (Santa Cruz, sc-56) was validated by knockout and western blot (Dietsch et al. BioTechniques. 62(2):80-82 (2017)). KAP1 (Bethyl, A300-274A) was validated by the manufacturer and used in recent studies (Uhlen et al. Nat. Methods. 13(10):823 (2016); (Spies et al. Nat. Cell. Bio. 21(4):487.497 (2019); Teloni et al. Mol. Cell. 72(4):670-683 (2019); Silva et al. Nat. Commun. 10(1),2253 (2019); Ali et al. Nat. Commun. 10(1),926 (2019)) and served as loading control. GFP (Torrey Pines biolabs, TP401) was validated by western blot (Bruhn et al., Nat. Commun. 11, 4154 (2020); https://www.amsbio.com/rabbit-anti-gfp-pab-tp401). All secondary antibodies were validated by the manufacture and used in recent studies (Lezaja et al., Nat. Commun. 12, 3827 (2021), Teloni et al. Mol. Cell. 72(4):670-683 (2019), Porro et al. Sci. Adv. 7:eabf7906 (2021), Gatti et al. Cell Reports. 32,107985 (2020).

# Eukaryotic cell lines

Policy information about cell lines

**Cell line source(s)**

U-2 OS cells: ATCC (HTB-96; RRID:CVCL_0042)
U-2 OS GFP-RPA2 WT: This study
U-2 OS GFP-RPA2 S>D: This study
U-2 OS GFP-RPA2 RFP-TRF2: This study
U-2 OS GFP-RPA: provided by Dr. Luis Toledo (University of Copenhagen, Denmark)
U-2 OS GFP-53BP1 RPA70-mScarlett: Lezaja et al., Nat Commun. 2021 Jun 22;12(1):3827 (University of Zurich, Switzerland)
U-2 OS Flp-IN Trex: provided by Dr. Kerstin Gari (Zurich University of Applied Sciences, Switzerland)
U-2 OS TurboID-RPA2-mCherry-Cry2: This study
U-2 OS TurboID-mCherry-Cry2: This study
HeLa cells: ATCC (CCL-2; RRID:CVCL_0030), kindly provided by Prof. Dr. Michael Hottiger (University of Zurich, Switzerland)

**Authentication**

The parental U-2 OS cell line was authenticated by STR profiling. HeLa cells used as negative control for C-circle analysis were not re-authenticated.

**Mycoplasma contamination**

All cell lines were tested every 4-6 weeks for mycoplasma contamination and always scored negative.

**Commonly misidentified lines**
(See ICLAC register)

No commonly misidentified cell lines were used in this study.

