## [Peer Review File · Nature Structural & Molecular Biology]

Peer Review Information

Manuscript Title: Phase separation properties of RPA combine high-affinity ssDNA binding with dynamic condensate functions at telomeres

Corresponding author name(s): Dr Matthias Altmeyer

Editorial Notes:

Transferred manuscripts This manuscript has been previously reviewed at another journal that is not operating a transparent peer review scheme. This document only contains reviewer comments, rebuttal and decision letters for versions considered at Nature Structural & Molecular Biology.

Reviewer Comments & Decisions:

Decision Letter, initial version:

Message: 31st Mar 2022

Dear Dr Altmeyer,

Thank you again for submitting your manuscript "Phase separation properties of RPA allow dynamic condensate formation despite ultrahigh-affinity ssDNA binding". I apologize for the delay while we awaited the full complement of comments from the 3 reviewers who evaluated your paper. These are copied below, and include the reports from the 2 referees that we previously shared with you by email. In light of these reports, we remain interested in your study and would like to see your response to the comments of the referees, in the form of a revised manuscript.

You will see that while all 3 reviewers are positive about potential interest and impact of the findings, they express some overlapping concerns about strength of the current data supporting the functional relevance of LLPS to RPA function and ALT maintenance, and suggest additional experimentation and analyses to address these concerns. Reviewer #1 finds that RPA localization at telomeres needs to be more clearly demonstrated, and that the current data do not establish how LLPS increases RPA's ssDNA binding affinity. Reviewer #3 requests additional assays to link RPA condensate formation, and its

regulation by DNA-PK, with characteristic ALT phenotypes in order to support the proposed regulatory model. Reviewer #2's detailed comments summarize and reinforce these concerns, and the reviewer offers specific suggestions of assays using combinations of RPA phosphomimetic mutants, and requests more rigorous statistical analysis and data reporting, to address these key points. Editorially, we agree that these suggestions would strengthen the findings, and ask that they be included in a revised manuscript.

Please be sure to address/respond to all concerns of the referees in full in a point-by-point response and highlight all changes in the revised manuscript text file.

We appreciate the requested revisions are extensive. We thus expect to see your revised manuscript within 6 months. If you cannot send it within this time, please let us know. We will be happy to consider your revision as long as nothing similar has been accepted for publication at NSMB or published elsewhere. Should your manuscript be substantially delayed without notifying us in advance and your article is eventually published, the received date would be that of the revised, not the original, version.

Reporting Summary:

When submitting the revised version of your manuscript, please pay close attention to our [href="https://www.nature.com/nature-research/editorial-policies/image-integrity"](https://www.nature.com/nature-research/editorial-policies/image-integrity)>Digital Image Integrity Guidelines. and to the following points below:

FOR MS WITH CROPPED GELS: Please note that all key data shown in the main figures as cropped gels or blots should be presented in uncropped form, with molecular weight markers. These data can be aggregated into a single supplementary figure. While these data can be displayed in a relatively informal style, they must refer back to the relevant

figures. These data should be submitted with the last revision, prior to acceptance, but you may want to start putting it together at this point.

Please use the link below to submit your revised manuscript and related files when they are ready:

[Redacted]

With kind regards,

Beth

Beth Moorefield, Ph.D.
Senior Editor
Nature Structural & Molecular Biology

Referee expertise:

Referee #1: DNA repair & genome stability/LLPS

Referee #2: DNA repair & genome stability/chromatin imaging

Referee #3: Telomere biology/ALT maintenance

Reviewers' Comments:

Reviewer #1:

Remarks to the Author:

Spegg et al. investigate the liquid-liquid phase separation (LLPS) properties and functions of the ssDNA binding protein RPA. The authors employed complementary techniques to assess the LLPS properties in vitro and in cells. First, the authors utilized the optogenetic system to interrogate the LLPS properties of a few selected DNA repair proteins. Of interest, they identified light-induced foci formation for the RPA2 protein, similar to the positive FUSN control. The propensity for RPA2 to phase separate was further investigated using purified proteins in vitro. The authors observed spontaneous condensation of the heterotrimeric RPA complex into liquid droplets. This process was stimulated by adding ssDNA, which localizes inside the RPA droplets. Finally, using a biotin proximity labeling approach coupled to the optogenetic system, the authors identified Cry2-specific light-activated interactors of RPA2, revealing a potential function for RPA2 LLPS in telomere maintenance via the ALT pathway.

Overall, Spegg et al. report important findings with the potential to impact our understanding of genome integrity. This manuscript should be of immediate interest to researchers in different areas due to the central role of RPA proteins in genome maintenance. The authors provided high-quality data to assess the LLPS properties of RPA2 both in vitro and in cells using a combination of complementary techniques. Notably, using the optogenetic system in combination with quantitative proximity proteomics enabled the identification of RPA2-dependent clustering-induced interactions. The authors were subsequently able to recapitulate one of these condensation-dependent interactions in vitro by combining purified tRPA and RAD52. The current version of the manuscript was of an appropriate length. However, several weaknesses remain. Of particular note, the overall message related to the functional importance of the findings requires more focused attention. Also, I often found myself trying to figure out the exact experimental conditions used across the manuscript, limiting my ability to interpret data accurately.

- The functional consequence and physiological impact of RPA2 LLPS seem preliminary and require further investigation.
- This manuscript did not clearly show how the LLPS properties of RPA are linked with its ultra-high affinity for ssDNA. Therefore, the title and key message appear over-reaching.
- In Figure 4 and extended Figures 6 and 7, it is unclear what the RPA and RAD52 foci are marking. In the case of FANCM depletion, which results in telomere replication stress/damage and induces damage elsewhere in the genome, the authors assume that the RPA/RAD52 foci faithfully mark telomeres. For example, in figure 4f, the authors monitored the RPA fusion frequency in cells treated with aphidicolin, and they did not clearly link it to telomeres; "the mutant cells showed a significant defect in replication stress-induced fusions" (lines 233-234). The authors then concluded that the "propensity of RPA to form dynamic condensates drives telomere clustering" (lines 235-236). Again, later in the discussion (line 288-291), the authors wrote, "... RAD52 enrichment at sites of ALT activation is greatly reduced in cancer cells expressing phase separation impaired, phospho-mimetic RPA2. Moreover, telomere clustering was impaired". One should use

telomere-specific markers in these experiments.

- The experimental conditions for figure 4e are unclear, making it difficult to interpret the data accurately. Were the cells subjected to aphidicolin treatment? FANCM depletion? No treatment? Is RPA always colocalizing with the telomeric marker TRF2 under these experimental conditions?
- Related to extended figure 2c, for the confocal and STED micrographs, the authors should highlight where the insets were taken from, similar to extended figure 2b.
- Related to figure 2h, is the panel mislabelled (Cy3-tRPA instead of Cy3-ssDNA)? Please also explain the peripheral localization of Cy3-tRPA in the droplets.
- In the legend of extended figure 4a, the authors refer to different incubation periods. Do the different colors of the dots in the plot refer to these different incubation periods?
- For extended figure 4b, if you don't see the ssDNA For and Rev on the gel, what is the relevance of this panel?
- On line 166, the authors cited reference 20 (Mine-Hattab et al., 2021, eLife) as consistent with their results that yeast Rfa2 "showed no light-induced clustering in optoDroplet experiments." However, Mine-Hattab et al. did not analyze the LLPS properties of Rpa2 and instead studied Rpa1.
- Related to extended figure 6a, one cannot exclude the possibility that RAD52 forms liquid droplets independently of RPA2. Only Rad52 is labeled in these experiments, and budding yeast Rad52 undergoes LLPS (Oshidari et al., 2020, Nat Commun; Mine-Hattab et al., 2021, eLife).
- The authors should comment on their observation that LLPS of RPA2 does not appear to be conserved in yeast (extended figure 5b). Why would humans have acquired and preserved the LLPS properties of RPA2?
- Line 105-106, the authors mentioned, "RPA spontaneously formed nuclear foci in these cells and time-lapse experiments revealed fusion and fission events." This sentence is confusing, and the authors should clearly state whether these cells were treated with aphidicolin.
- Line 131 "... ssDNA affects the phase behavior of RPA..." is missing the word "separation."

Reviewer #2:

Remarks to the Author:

This study by Spegg et al. shows that the human RPA complex has an intrinsic propensity to form condensates. A combination of in vitro and in vivo approaches reveals that RPA2 is the main driver of this condensation, ssDNA promote phase separation, and phospho-mimicking mutations in the RPA2 unstructured region interfere with condensation. Quantitative proximity proteomics identifies several RPA condensate components,

including proteins required for telomere maintenance in ALT (BTR complex, FANCM and RAD52). Following up on this observation, the authors also show that affecting RPA condensation (RPA2 S>D mutation) impairs telomere clustering and maintenance in cancer cells.

The phase separation properties of human RPA are overall well characterized in this study. Previous studies showed that the bacteria ssDNA binding protein SSB form condensates (Harami, 2020), perhaps taking away some of the novelty. That said, SSB and RPA are quite different components, and yeast RPA does not seem to display phase separation properties in single molecule studies (Mine'-Hattab, 2021). Thus, observing phase separation properties associated with human RPA is relevant and important. Establishing the role of phosphorylation in counteracting condensate formation is also an interesting observation, although not well characterized in the context of RPA functions in this study. Specifically, the importance of phase separation in biological contexts like DNA repair, cell cycle progression and protein interactions where RPA phosphorylation is known to be important, is not investigated.

A strength of this paper is the application of a recently developed tool (quantitative proximity proteomics) to probe for condensate composition in vivo (Alghoul et al, STAR protocol 2021). Given the identification of some ALT pathway components, the authors also follow up on this observation suggesting a role of RPA-mediated condensation in telomere maintenance. However, a link with ALT repair is not clearly established. ALT repair occurs at PML bodies (APBs), which are also phase separated structures, but the coexistence between APBs and RPA condensate is not investigated.

Despite the overall quality of the data and the importance of the discovery, a better characterization of the role of RPA-driven condensates in vivo is needed to significantly advance the field and for publication in NSMB.

We also note that some essential information is missing or difficult to retrieve (e.g., the phenotype of individual phospho-mutants, some quantifications and statistical validations), preventing the evaluation of parts of the data.

Major points:

1) Human RPA2 and its phosphorylation have known biological functions through RPA regulation, including in DNA repair, replication stress responses, cell cycle progression, and protein-protein interactions, in addition to telomere stability. Understanding the link between RPA condensation and these functions is a critical missing point in this study. This can be addressed by investigating how selected phosphomimic mutants affect these functions, correlating these phenotypes to effects on phase separation, and establishing the extent to which induction of phase separation mutants (S>D) re-establishes these functions.

2) Related to point 1), Fig S5f shows the average values for the effect of multiple combination of phosphomutants in ortodloplet formation, but it is unclear which individual mutation or combination leads to which phenotypes (low intensity thin pink lines in S2g). Some combinations of mutations seem just as effective as the S>D 8x mutant in affecting ortodloplets, but this is not highlighted or discussed. The rationale for the selection of 8S sites to mutagenize in the S>D mutant is unclear. Specifically, why T21 is excluded is not explained. Similarly, why certain combinations of mutations have been done in S5f,g is

not clear. Statistical comparisons between effects of different mutants is also missing. If a lower number of mutations results in phenotypes comparable to the S>D mutant, the lowest number of mutations resulting in a strong phenotype should be used for functional characterizations *in vivo*. This is important, as we know about the role of at least some phosphorylations in RPA functions, and clearly correlating these with phase separation properties is a critical point.

3) The authors initiate investigating the role of RPA condensation in telomere maintenance in ALT (figures 4e-g). Damaged telomeres leads to ALT-associated promyelocytic leukemia (PML) nuclear bodies (APBs) formation, telomere clustering within the APBs, and HR repair (Cho et al., 2014). The authors also previously showed that RPA foci co-localize with PML bodies in G1, in ALT+ cells (Lexaja et al., 2021). PML bodies are phase separated structures, but the coexistence/relationship between RPA and PML condensates is not investigated or discussed in this paper. For example, is PML at telomeres affected in S>D mutants? Are RPA condensates permeable to PML and vice versa *in vitro*?

4) RPA fusion frequency is reduced in S>D mutants (Fig 4f), and more telomeres are lost (Fig 4g). However, it is not clear how the authors reconcile these results with a model where 'in vivo nuclear RPA condensates provide a reservoir of highly concentrated free RPA (Fig S8)'. Is RPA condensation important to bring telomeres together for repair, or to increase the local concentration of RPA? The same applies to other functions of RPA.

5) The introduction would benefit from a better explanation of what is already known about bacteria and yeast ability to phase separate.

We also note that the authors show RPA at DSBs in their model, but DSB repair has not been investigated in the current version of the paper. The model should better reflect the data. The ability of ssDNA to enhance RPA condensation should also be integrated in the model.

Additional points on experiments and statistical analyses

- Figure S1a, b: The images presented do not clearly show clustering defects for CtIP and Rad51 (which is shown in the ranking), as foci that are already present in untreated conditions (pre) remain similar after light induction (post). Quantification of optodroplet number or intensity for CtIP and Rad51 pre and post light induction should be provided. Quantifications for RNF8, RNF168 (Fig. S1), RPA1-3 (Fig. 3b) and controls should also be shown.

- Why the tRPA show less ability to cluster compared to individual RPA2 ? The image provided shows only three light-induced foci (fig. S1e). Quantifications for these differences should be provided and the difference between RPA2 and tRPA condensation phenotypes should be discussed.

- In Fig. S2a, the authors analyze the ability of RPA to form condensates in cells stably expressing fluorescently labelled tRPA. It is unclear why data are shown only in response to APH here. APH is also only mentioned in S2 figure legend, but not in the main text or in the figure itself. A better explanation should be provided, information included in the figure, and a quantification of the fusion/fission events should be added. We note that the text says that foci 'spontaneously form', but this is misleading as foci are induced by APH treatment. Are spontaneous and APH-induced foci also sensitive to 1,6 Hexanediol?

- Figure S2: How do recovery rates from FRAP experiments relate with other proteins able to phase separate? This information should be included in the text. Also, I could not find an explanation for experiments shown in Figures S2e,f, in the main text.
- Figure S2: It is unclear why the authors focus on RPA phosphorylation by DNA-PK and not for example ATM/ATR. A better explanation should be provided.
- Figure 2: Why the authors provide STORM and STED microscopy data on tRPA GFP is unclear. The authors talk about sphericity of the signal, but isn't this always the case in single molecule imaging?
- Figure 3c suggests that RPA2 self-interact by CoIP. For this conclusion to be drawn, the authors need to show that the interaction is not mediated by ssDNA.
- Figure 3d: Confirmed phosphorylation sites within the N-IDR are indicated but the specific aminoacids are not shown. This information should be provided here. Are these the same aa mutated in figure s5g? If that is the case at least two sites are missing in 3d.
- Figure 4f: RPA signals in APH are not just marking replication damage at telomeres. A telomere marker should also be followed to conclude that telomere fusions are affected in S>D mutants.
- Figure 4f legend: How many cells were quantified from the 3 different experiments? How many foci analyzed?
- Figure 4g: Replicates for this experiment should be provided.
- Figure 4: Is the RPAmutS>D mutant still associated with telomeres? Is this association level the same observed in RPA WT?
- Why in Figure 2d the control with only tRPA does not show droplets formation, as expected from other experiments? Is a different incubation time used here? Please clarify in figure legend and in the figure.
- Fig S6e: Are Rad52 foci only marking telomeres in this experiment? Provide a reference or show that this is the case (e.g, by performing Rad52 IF + telomere DNA FISH).
- Fig. S6d,e Statistical validations are needed.
- Why ALT activity has been induced by FANCM depletion in U2OS cells (which are already ALT+) needs a better explanation.
- In the text the authors mention that among the clustering-induced interaction there are all components of the BLM-TOP3A-RMI (BTR). From the Excel file provided, BLM is found only in the group of common interactors. Which one is correct?

Minor Points

- Figure 1b: Why did the authors use FUS as a positive control rather than the mCh-Cry2olig previously described (Kilic et al., 2019)? An explanation in the methods or in the

main text should be provided.

- Figures 1b and S1a: Please provide schematic views of the Cry2-mCherry-fusion constructs used in the experiment next to the images, to help understand the experiments.
- RPA2 condensates seem to form in the nucleus in Fig1b but in the nucleus+cytoplasm in S1a. What is the common behavior?
- Figure 1d, Squares or arrowheads pointing to the foci showed in the magnified insets should be added
- Figure 1e: An experimental scheme should be included for a better understanding of how the experiment was conducted.
- Figure 2h: Cy3-ssDNA > Cy3-RPA
- Figure S5f,g: these data seem important enough that they should be moved to the main figure.
- Figure 3d legend: 'Consistent with x-ray crystallography data on the RPA tetramerization core' > trimerization
- Figure 3e: a Predicted aligned Error (PAE) plot for the structure prediction is shown, but this is not well discussed in the text. Describe this better.
- Figure 3f-g: Axis size for GlobPlot and PONDR graphs should be increased, as they are very difficult to read.
- Figure 3a: a description of the domain (PS, WHD) of RPA subunits should be provided in the legend.

Reviewer #3:

Remarks to the Author:

Replication protein A (RPA), a single-strand DNA (ssDNA) binding complex, is critical for DNA metabolism. RPA binding is required to protect the ssDNAs as well as recognize the ssDNAs in RNA/DNA loop (R loop) during DNA replication, repair, and recombination processes which are essential for the maintenance of genome integrity. In this manuscript, Spegg et al. show that RPA complex can form biomolecular condensates in vivo and in vitro through N-terminal intrinsically disordered region (N-IDR) in RPA2. They demonstrated that RPA condensation involves in ALT telomere synthesis through interaction with FANCM - BLM-TOP3-RMI (BTR) complex, and RAD52 proteins. This work provides not only a new understanding of the regulatory mechanism of RPA proteins but also novel insights of the role of biomolecular condensates in DNA metabolism. Overall, this manuscript has been written very clearly and cited previous work appropriately. However, there are some major points to be addressed.

1. Cell biology experiments (Fig 1, 3, 4); these are very stimulating results; however, they are all based on RPA2 overexpressing cells – either U2OS GFP-RPA2 or U2OS TurboID-RPA2-mCherry-Cry2 cells derived from U2OS Flp-In T-Rex system. Despite of the fact that

RPA proteins are highly abundant in cellular level, overexpression-based observations may not fully represent the endogenous-level/cellular physiology. For example, in Fig 3C and S4b, endogenous RPA and GFP-RPA should be represented in same blot, not separately, by using anti-RPA antibody. This could be addressed by generating endogenous level GFP-RPA2 by targeting RPA2 locus using CRISPR.

2. If the formation of RPA condensates is regulated by DNA-PK mediated phosphorylation in N-IDR of RPA2 as shown in RPA (S Δ D) mutant and in vitro condensate experiment with DNA-PK-mediated phosphorylation, it would be worth to examine the effect of DNA-PKc inhibitor in RPA-fusion frequency in vivo which potentially supports the author's idea.

3. How RPA condensates involve ALT telomere synthesis? What is the outcome of phosphomimetic/condensation-impaired RPA expression in ALT pathway, such as APB formation, C-circle levels, and G2-synthesis/MiDAS/post-MiDAS at telomeres?

4. Why RPA2 S Δ D cells displayed more frequent telomere loss than WT? Is this due to increase in unprotected ssDNA or ssDNA gap at telomeres which are processed/cleaved by nucleases?

5. Page 11 line 236-240: Cannot find the data supporting "RPA condensates drives telomere clustering" and "RPA2 S Δ D cells displayed elevated telomere fragility"

6. Fig 4e, GFP-RPA2 and RAD52 colocalization at telomeres (TRF2), is this specific to certain cell cycle phase, such as post-MiDAS/G1 phase?

7. As shown by the author, TurboID-Cry2-RPA2 is very innovative system for the quantitative proximity proteomics. However, TurboID is also useful tool for the quantitative proximity genomics, since it biotinylates proximal nucleotides- DNA, and RNA. Are RPA proteins in condensates less bound to ssDNAs or ALT telomeres as the author discussed (Page12 line 268-269 "RPA condensates provide a reservoir of highly concentrated free RPA")?

Author Rebuttal to Initial comments

We would like to thank the reviewers for their time and interest in our work and for their thoughtful and constructive feedback. All comments and suggestions were very helpful and inspired us to perform additional experiments and analyses that collectively corroborate and extend our study. All changes and new additions in our revised manuscript, which now contains 6 main and 14 extended data figures, are detailed in our point-by-point response below and are highlighted in the revised manuscript text file.

Reviewers' Comments:

Reviewer #1:

Remarks to the Author:

Spegg et al. investigate the liquid-liquid phase separation (LLPS) properties and functions of the ssDNA binding protein RPA. The authors employed complementary techniques to assess the LLPS properties in vitro and in cells. First, the authors utilized the optogenetic system to interrogate the LLPS properties of a few selected DNA repair proteins. Of interest, they identified light-induced foci formation for the RPA2 protein, similar to the positive FUSN control. The propensity for RPA2 to phase separate was further investigated using purified proteins in vitro. The authors observed spontaneous condensation of the heterotrimeric RPA complex into liquid droplets. This process was stimulated by adding ssDNA, which localizes inside the RPA droplets. Finally, using a biotin proximity labeling approach coupled to the optogenetic system, the authors identified Cry2-specific light-activated interactors of RPA2, revealing a potential function for RPA2 LLPS in telomere maintenance via the ALT pathway.

Overall, Spegg et al. report important findings with the potential to impact our understanding of genome integrity. This manuscript should be of immediate interest to researchers in different areas due to the central role of RPA proteins in genome maintenance. The authors provided high-quality data to assess the LLPS properties of RPA2 both in vitro and in cells using a combination of complementary techniques. Notably, using the optogenetic system in combination with quantitative proximity proteomics enabled the identification of RPA2-dependent clustering-induced interactions. The authors were subsequently able to recapitulate one of these condensation-dependent interactions in vitro by combining purified tRPA and RAD52. The current version of the manuscript was of an appropriate length. However, several weaknesses remain. Of particular note, the overall message related to the functional importance of the findings requires more focused attention. Also, I often found myself trying to figure out the exact experimental conditions used across the manuscript, limiting my ability to interpret data accurately.

We thank this reviewer for their time to evaluate our study and we were delighted to read that the reviewer considers our data of high quality and our findings important and of immediate interest to a broad audience. This was extremely encouraging feedback, and we took the reviewer's helpful suggestions to further improve our study by providing addition

information on experimental conditions and by strengthening the functional implications of our findings, as detailed below.

- The functional consequence and physiological impact of RPA2 LLPS seem preliminary and require further investigation.

We added a significant amount of experimental data on the functional consequences and physiological impact of RPA's phase separation capability. Specifically, we provide additional data to strengthen the connection between RPA foci in cells and telomeres undergoing ALT (new Extended Data Figures 9 and 13). Moreover, we provide a more thorough and quantitative analysis of RPA2 phosphosite mutants and their clustering defects (new Figure 4d,e and new Extended Data Figure 7e). Importantly, we also provide a set of new results that exclude over-expression artefacts (new Extended Data Figure 3a-c and new Extended Data Figure 4b-d). Along these lines, we provide additional characterization of the siRNA-resistant RPA2 replacement cells (WT and S/T→D) and their telomere fragility phenotypes, including formation of APBs, DNA synthesis at telomeres outside of S-phase, ssDNA at telomeres, extrachromosomal C-circle formation, and RAD52 enrichment at telomeres (new Figure 6 and Extended Data Figure 12). Collectively, these analyses corroborate a RAD52 enrichment defect at telomeres in RPA S/T→D cells with consistently elevated signs of telomere fragility.

- This manuscript did not clearly show how the LLPS properties of RPA are linked with its ultra-high affinity for ssDNA. Therefore, the title and key message appear over-reaching.

We agree with the reviewer that this point was not sufficiently developed in our original manuscript and, at least in part, was based primarily on published work, which had demonstrated facilitated exchange of RPA on ssDNA by an excess of free RPA (Gibb et al. 2014, PMID: 24498402), and which had established that RPA2 phosphosite mutations do not interfere with ssDNA binding (Binz et al. 2003, PMID: 12819197). These two independent findings, when combined with our results, indicate that the high concentration of RPA in nuclear RPA foci facilitates dynamic exchange between RPA and ssDNA, despite RPA's high affinity for ssDNA. In our revised manuscript, we provide new data that directly support this notion: We show that whereas wild-type trimeric RPA forms dynamic condensates with ssDNA *in vitro*, the phosphomimetic S/T→D mutant forms amorphous aggregates with ssDNA (new Extended Data Figure 7f). By FRAP experiments we show that the ssDNA in these aggregates is less mobile compared to the ssDNA in WT RPA condensates, suggesting that while both WT and S/T→D RPA can bind to ssDNA with high affinity, the dynamic exchange is impaired in the S/T→D mutant (new Extended Data Figure 7g,h). To better reflect these findings and the extended functional characterization in the title of the revised manuscript, we suggest "Phase separation properties of RPA combine high-affinity ssDNA binding with dynamic condensate functions at telomeres".

- In Figure 4 and extended Figures 6 and 7, it is unclear what the RPA and RAD52 foci are marking. In the case of FANCM depletion, which results in telomere replication

stress/damage and induces damage elsewhere in the genome, the authors assume that the RPA/RAD52 foci faithfully mark telomeres. For example, in figure 4f, the authors monitored the RPA fusion frequency in cells treated with aphidicolin, and they did not clearly link it to telomeres; “the mutant cells showed a significant defect in replication stress-induced fusions” (lines 233-234). The authors then concluded that the “propensity of RPA to form dynamic condensates drives telomere clustering” (lines 235-236). Again, later in the discussion (line 288-291), the authors wrote, “... RAD52 enrichment at sites of ALT activation is greatly reduced in cancer cells expressing phase separation impaired, phospho-mimetic RPA2. Moreover, telomere clustering was impaired”. One should use telomere-specific markers in these experiments.

We thank the reviewer for raising this important point. In our revised manuscript, by careful co-localization analyses, we now show that around 70% of spontaneous and replication stress-induced RPA foci co-localize with telomeres. Upon FANCM depletion, we find around 90% of RPA foci to co-localize with TRF2. These numbers indicate that not all, but a clear majority of RPA foci mark telomeres under these conditions (new Extended Data Figure 9).

To substantiate this point further, we generated stable cell lines in which both RPA2 and TRF2 are fluorescently labeled. Consistent with our co-localization analysis from IF staining, GFP-RPA2 co-localizes predominantly with RFP-TRF2 in these cells, and replication stress-induced GFP-RPA2 fusions coincide in space and time with RFP-TRF2 fusions (new Extended Data Figure 13d,e).

The RAD52 defect, which we observe in RPA2 S/T→D cells, has now also been linked directly to telomeres, by quantifying RAD52-positive TRF2 foci (new Figure 6f,g and Extended Data Figure 12f,g).

- The experimental conditions for figure 4e are unclear, making it difficult to interpret the data accurately. Were the cells subjected to aphidicolin treatment? FANCM depletion? No treatment? Is RPA always colocalizing with the telomeric marker TRF2 under these experimental conditions?

We apologize for the confusion and have replaced Figure 4e by a clearer and more comprehensive analysis of RPA co-localization with the telomeric marker TRF2 in different experimental conditions (new Extended Data Figure 9). We find around 70% of RPA foci to co-localize with TRF2, and this number increases to around 90% in FANCM-depleted cells, indicating a strong overlap between these two markers.

- Related to extended figure 2c, for the confocal and STED micrographs, the authors should highlight where the insets were taken from, similar to extended figure 2b.

We thank the reviewer for pointing out the missing inset highlight and have included it in the revised manuscript (Extended Data Figure 4e).

- Related to figure 2h, is the panel mislabelled (Cy3-tRPA instead of Cy3-ssDNA)? Please also explain the peripheral localization of Cy3-tRPA in the droplets.

We apologize for the confusion caused by this panel. The panel was in fact not mislabeled, but instead two differentially labeled ssDNA oligos of identical sequence (one FAM-labelled and one Cy3-labeled) had been used in this experiment together with unlabeled RPA. While ‘dye-only’ controls do not form droplets and do not get enriched in pre-formed RPA droplets (Extended Data Figure 8a), we cannot formally exclude that the peripheral localization correctly observed by the reviewer was caused by the physicochemical properties of the differentially labeled competing ssDNA oligos themselves. We therefore replaced the panel by another representative example of droplet fusion, in which only a single ssDNA species was used (Figure 2h).

- In the legend of extended figure 4a, the authors refer to different incubation periods. Do the different colors of the dots in the plot refer to these different incubation periods?

Indeed, we had analyzed droplet formation at three different time-points after mixing RPA with nucleic acids. To depict the data in a clearer manner, we replaced the figure panel by a new experiment showing replicates for each of the three time-points (new Extended Data Figure 6a). Similarly, we also replaced the panel comparing ssDNA to sequence-matched ssRNA by a new experiment showing replicates for each of the three time-points (new Extended Data Figure 6f).

- For extended figure 4b, if you don’t see the ssDNA For and Rev on the gel, what is the relevance of this panel?

The panel was meant to serve as a control for successful annealing of the two oligos, but we agree that input controls for the two ssDNA oligos were not present. To circumvent this limitation and visualize the unlabeled ssDNAs and the annealed dsDNA on the same gel, we made use of the DNA dye RedSafe, which binds to both ssDNA and dsDNA and has a higher affinity for the latter. We now show a short and a long run of the same agarose gel, in which the input ssDNAs are visible at the early time-point (new Extended Data Figure 6b). At the later time-point the preference of RedSafe for dsDNA prevails, consistent with the panel that was shown in our original submission. The efficiency of annealing is further supported by the inability of the annealed DNA to induce RPA condensation (new Extended Data Figure 6a).

- On line 166, the authors cited reference 20 (Mine-Hattab et al., 2021, eLife) as consistent with their results that yeast Rfa2 “showed no light-induced clustering in optoDroplet experiments.” However, Mine-Hattab et al. did not analyze the LLPS properties of Rpa2 and instead studied Rpa1.

Has been corrected, thank you very much.

- Related to extended figure 6a, one cannot exclude the possibility that RAD52 forms liquid

droplets independently of RPA2. Only Rad52 is labeled in these experiments, and budding yeast Rad52 undergoes LLPS (Oshidari et al., 2020, Nat Commun; Mine-Hattab et al., 2021, eLife).

We agree and have included an additional panel showing that RAD52 alone does not form liquid droplets under these conditions (new Extended Data Figure 8b).

- The authors should comment on their observation that LLPS of RPA2 does not appear to be conserved in yeast (extended figure 5b). Why would humans have acquired and preserved the LLPS properties of RPA2?

We speculate in the revised discussion that “the larger size of the human genome and the bigger volume of the cell nucleus may have required additional layers of control for the spatial and temporal regulation of sub-cellular compartmentalization. This requirement could be one of the reasons why human cells have acquired and preserved phase separation properties of RPA2 that can be modulated by multi-site phosphorylation of its N-IDR”.

- Line 105-106, the authors mentioned, “RPA spontaneously formed nuclear foci in these cells and time-lapse experiments revealed fusion and fission events.” This sentence is confusing, and the authors should clearly state whether these cells were treated with aphidicolin.

Has been corrected, thank you.

- Line 131 “... ssDNA affects the phase behavior of RPA...” is missing the word “separation.”

Has been corrected, thank you.

Reviewer #2:

Remarks to the Author:

This study by Spegg et al. shows that the human RPA complex has an intrinsic propensity to form condensates. A combination of in vitro and in vivo approaches reveals that RPA2 is the main driver of this condensation, ssDNA promote phase separation, and phospho-mimicking mutations in the RPA2 unstructured region interfere with condensation. Quantitative proximity proteomics identifies several RPA condensate components, including proteins required for telomere maintenance in ALT (BTR complex, FANCM and RAD52). Following up on this observation, the authors also show that affecting RPA condensation (RPA2 S>D mutation) impairs telomere clustering and maintenance in cancer cells.

The phase separation properties of human RPA are overall well characterized in this study. Previous studies showed that the bacteria ssDNA binding protein SSB form condensates (Harami, 2020), perhaps taking away some of the novelty. That said, SSB and RPA are quite different components, and yeast RPA does not seem to display phase separation properties in single molecule studies (Mine'-Hattab, 2021). Thus, observing phase separation properties associated with human RPA is relevant and important. Establishing the role of phosphorylation in counteracting condensate formation is also an interesting observation, although not well characterized in the context of RPA functions in this study. Specifically, the importance of phase separation in biological contexts like DNA repair, cell cycle progression and protein interactions where RPA phosphorylation is known to be important, is not investigated.

A strength of this paper is the application of a recently developed tool (quantitative proximity proteomics) to probe for condensate composition in vivo (Alghoul et al, STAR protocol 2021). Given the identification of some ALT pathway components, the authors also follow up on this observation suggesting a role of RPA-mediated condensation in telomere maintenance. However, a link with ALT repair is not clearly established. ALT repair occurs at PML bodies (APBs), which are also phase separated structures, but the coexistence between APBs and RPA condensate is not investigated.

Despite the overall quality of the data and the importance of the discovery, a better characterization of the role of RPA-driven condensates in vivo is needed to significantly advance the field and for publication in NSMB.

We also note that some essential information is missing or difficult to retrieve (e.g., the phenotype of individual phospho-mutants, some quantifications and statistical validations), preventing the evaluation of parts of the data.

We thank this reviewer for the time to evaluate our study and were delighted to read that the reviewer appreciates the overall quality of the data and considers the RPA phase separation

properties well characterized and our discoveries relevant and important. This was very encouraging feedback.

We have done our best to address the points raised by the reviewer and to further investigate and characterize the role of RPA condensation properties for ALT telomere maintenance. Specifically, we now provide additional data to strengthen the connection between RPA foci in cells and telomeres undergoing ALT (new Extended Data Figures 9 and 13). Moreover, we provide a more thorough and quantitative analysis of RPA2 phosphosite mutants and their clustering defects (new Figure 4d,e and new Extended Data Figure 7e). We also provide a set of new results that exclude over-expression artefacts (new Extended Data Figure 3a-c and new Extended Data Figure 4b-d). We provide additional characterization of the siRNA-resistant RPA2 replacement cells (WT and S/T→D), including cell cycle and DNA damage response analyses (new Extended Data Figure 11), and we provide an extended characterization of their telomere fragility phenotypes, including formation of APBs, DNA synthesis at telomeres outside of S-phase, ssDNA at telomeres, extrachromosomal C-circle formation, and RAD52 enrichment at telomeres (new Figure 6 and Extended Data Figure 12). Collectively, these analyses corroborate a RAD52 enrichment defect at telomeres in RPA S/T→D cells with consistently elevated signs of telomere fragility using multiple orthogonal readouts, altogether supporting our initial conclusions, and strengthening the link between RPA condensation and telomere maintenance.

Major points:

1) Human RPA2 and its phosphorylation have known biological functions through RPA regulation, including in DNA repair, replication stress responses, cell cycle progression, and protein-protein interactions, in addition to telomere stability. Understanding the link between RPA condensation and these functions is a critical missing point in this study. This can be addressed by investigating how selected phosphomimic mutants affect these functions, correlating these phenotypes to effects on phase separation, and establishing the extent to which induction of phase separation mutants (S>D) re-establishes these functions.

To address this point, we have investigated a relatively comprehensive set of 15 different RPA2 phosphosite mutants, including single, double, triple, and quadruple mutants. Their nuclear expression levels were normalized and their light-induced clustering potential was quantified (new Figure 4d,e and Extended Data Figure 7e). These analyses did not point to a single critical phosphorylation site, but rather supported our initial interpretation that multi-site phosphorylation gradually decreases the phase separation potential of RPA2. A complete or nearly complete “loss-of-function” in terms of RPA phase separation modulated by the N-terminal disordered region of RPA2 is therefore only achieved by combined mutation of the phosphorylation sites. The octuple S/T→D mutant (RPA2 S8/S11/S12/S13/T21/S23/S29/S33D) showed nearly complete loss of light-induced clustering (Figure 4d) and loss of droplet formation in the purified trimeric RPA complex *in vitro* (Figure 4f,g), and was therefore used for *in vivo* characterization. To this end, we had generated siRNA-resistant cell lines, in which the endogenous RPA2 can be depleted and is

replaced by the ectopic WT or S/T→D mutant RPA. The depletion of endogenous RPA2 is efficient in this system and the levels of WT and S/T→D RPA are very similar and only slightly above the endogenous levels (Extended Data Figure 10a). As these isogenic cell lines seem well suitable as replacement system, we used them for functional analyses. Cell cycle progression and EdU profiles are similar in WT and S/T→D cells and both cell lines respond similarly to CPT-induced DNA damage (new Extended Data Figure 11a-g). However, the S/T→D cells show a defect in RPA self-interaction in immunoprecipitation experiments (new Extended Data Figure 11h), reduced RAD52 recruitment to telomeres (new Figure 6g and Extended Data Figure 12g), and at multiple levels signs of telomere maintenance problems including increased ssDNA at telomeres, elevated telomeric DNA synthesis outside S-phase, and increased extrachromosomal C-circles (Figure 6 and Extended Data Figure 12). While these results do not exclude additional functions, the finding that RPA foci are frequently found at telomeres (70%-90%, depending on the condition, see Extended Data Figure 9) argues that the observed ALT phenotypes are relevant in the context of our study.

2) Related to point 1), Fig S5f shows the average values for the effect of multiple combination of phosphomutants in ortodloplet formation, but it is unclear which individual mutation or combination leads to which phenotypes (low intensity thin pink lines in S2g). Some combinations of mutations seem just as effective as the S>D 8x mutant in affecting ortodloplets, but this is not highlighted or discussed. The rationale for the selection of 8S sites to mutagenize in the S>D mutant is unclear. Specifically, why T21 is excluded is not explained. Similarly, why certain combinations of mutations have been done in S5f,g is not clear. Statistical comparisons between effects of different mutants is also missing. If a lower number of mutations results in phenotypes comparable to the S>D mutant, the lowest number of mutations resulting in a strong phenotype should be used for functional characterizations in vivo. This is important, as we know about the role of at least some phosphorylations in RPA functions, and clearly correlating these with phase separation properties is a critical point.

We apologize for the confusion and have clarified these issues in the revised manuscript. The clustering- and droplet-deficient mutant contains the T21D mutation and is identical to the version used previously by Marc Wold and colleagues (Binz et al. 2003, PMID: 12819197). For reasons of (over-)simplification we had denoted this mutant as “S→D”, although in fact it is an “S/T→D” mutant. We have changed this throughout the manuscript text and in the manuscript figures. We also retested and reanalyzed a panel of 15 phosphosite mutants that we had generated, including 6 single, 4 double, 3 triple, 1 quadruple, and 1 octuple (Binz et al. 2003, PMID: 12819197) mutant. The results are depicted in Figure 4d,e and Extended Data Figure 7e. Although additional combinations of phosphosite mutations could in principle be conceived, the panel of 15 constructs that was tested and analyzed at comparable expression levels strongly suggests that, instead of a single site or few specific sites, the phosphorylation-mimicking mutations and the negative charges introduced functionally cooperate to tune self-assembly properties, and that only the combined mutations lead to a near-complete loss of clustering.

3) The authors initiate investigating the role of RPA condensation in telomere maintenance in ALT (figures 4e-g). Damaged telomeres leads to ALT-associated promyelocytic leukemia (PML) nuclear bodies (APBs) formation, telomere clustering within the APBs, and HR repair (Cho et al., 2014). The authors also previously showed that RPA foci co-localize with PML bodies in G1, in ALT+ cells (Lexaja et al., 2021). PML bodies are phase separated structures, but the coexistence/relationship between RPA and PML condensates is not investigated or discussed in this paper. For example, is PML at telomeres affected in S>D mutants? Are RPA condensates permeable to PML and vice versa *in vitro*?

We thank the reviewer for these interesting and thoughtful questions. PML, unfortunately, has escaped all purification attempts so far and we are unaware of any lab that has achieved this task and of any publication reporting on PML expression and purification, precluding *in vitro* experiments with RPA droplets and their interaction with PML. However, we analyzed ALT-associated PML bodies (APBs) and found increased APBs in RPA S/T→D cells, together with increased telomeric DNA synthesis in G2 and G1, elevated ssDNA, and increased C-circles (new Figure 6 and Extended Data Figure 12). These results suggest that APBs can still form in RPA S/T→D cells, however RAD52 enrichment at telomeres is impaired. Together with the elevated C-circle formation in RPA S/T→D cells, the data are consistent with a shift towards RAD52-independent BIR associated with increased telomere shortening, as was shown previously by Zhang et al. (PMID: 30673617). We have included these considerations in the discussion.

4) RPA fusion frequency is reduced in S>D mutants (Fig 4f), and more telomeres are lost (Fig 4g). However, it is not clear how the authors reconcile these results with a model where ‘*in vivo* nuclear RPA condensates provide a reservoir of highly concentrated free RPA (Fig S8)’. Is RPA condensation important to bring telomeres together for repair, or to increase the local concentration of RPA? The same applies to other functions of RPA.

This is a great question and although we do not have a definite answer our revised manuscript provides some additional clues, which suggest that both features – a reservoir of highly concentrated free RPA to facilitate RPA exchange on ssDNA, and telomere clustering for repair – are connected: We observed that the RPA S/T→D mutant forms aggregates with ssDNA *in vitro*, instead of the dynamic liquid droplets that are formed by RPA WT (new Extended Data Figure 7f). The ssDNA in the RPA S/T→D aggregates is less mobile, as measured by FRAP, than in the WT RPA droplets. We consider these results in agreement with single molecule experiments by the Greene lab, which had revealed that rapid exchange between RPA and ssDNA requires an excess of free RPA (Gibb et al. 2014, PMID: 24498402). This rapid exchange, which we think happens in and is promoted by dynamic RPA condensates, facilitates RAD52 binding, which in turn can promote telomere clustering and repair as shown by Min et al. (PMID: 31171703).

5) The introduction would benefit from a better explanation of what is already known about bacteria and yeast ability to phase separate.

We agree that the work on bacterial SSB and yeast Rfa1 deserves more attention. We found it difficult, however, to include this in a coherent manner in the introduction, where introducing the marked differences between bacterial SSB and the trimeric human RPA complex, but also between yeast Rfa1 and human RPA2 would require extensive explanations. We found it more intelligible to include these aspects in the discussion, where now two new paragraphs address the differential phase separation behavior of bacterial SSB and yeast Rfa1, and how this relates to human RPA and our findings.

We also note that the authors show RPA at DSBs in their model, but DSB repair has not been investigated in the current version of the paper. The model should better reflect the data. The ability of ssDNA to enhance RPA condensation should also be integrated in the model.

We thank the reviewer for this helpful suggestion and have revised the model accordingly. We did not intend to depict DSBs, but rather indicate telomere clustering in RPA- and RAD52-containing condensates. We have clarified this in the figure legend.

Additional points on experiments and statistical analyses

- Figure S1a, b: The images presented do not clearly show clustering defects for CtIP and Rad51 (which is shown in the ranking), as foci that are already present in untreated conditions (pre) remain similar after light induction (post). Quantification of optodroplet number or intensity for CtIP and Rad51 pre and post light induction should be provided. Quantifications for RNF8, RNF168 (Fig. S1), RPA1-3 (Fig. 3b) and controls should also be shown.

We now provide the requested quantifications (new Extended Data Figure 1b,c, Figure 3b,c, Figure 4b,c, and Extended Data Figure 7c,d). In all cases the analyses are focused on comparable expression levels and on the blue light-induced formation of dynamic optoDroplets as they are formed by the positive controls FUS_N and Cry2_{oligo}.

- Why the tRPA show less ability to cluster compared to individual RPA2 ? The image provided shows only three light-induced foci (fig. S1e). Quantifications for these differences should be provided and the difference between RPA2 and tRPA condensation phenotypes should be discussed.

This difference is due to the lower expression level of the large tRPA construct compared to the smaller RPA2 construct. When normalized to comparable expression levels and quantified, the differences disappear, indicating that tRPA has a similar potential to form optoDroplets as RPA2 (new Extended Data Figure 2d,e).

- In Fig. S2a, the authors analyze the ability of RPA to form condensates in cells stably expressing fluorescently labelled tRPA. It is unclear why data are shown only in response to APH here. APH is also only mentioned in S2 figure legend, but not in the main text or in the figure itself. A better explanation should be provided, information included in the figure, and

a quantification of the fusion/fission events should be added. We note that the text says that foci ‘spontaneously form’, but this is misleading as foci are induced by APH treatment. Are spontaneous and APH-induced foci also sensitive to 1,6 Hexanediol?

We indeed see RPA foci fusions more prominently when cells are treated with a low dose of aphidicolin (0.2 μ M during live cell microscopy) and have clarified this in the figure panel and in the main text. We also repeated live cell experiments with CRISPR-engineered cells that express endogenously tagged RPA, and quantified fusion and fission events in untreated and APH-treated conditions. The results confirmed fusion and rare fission events after replication stress and were included as new Extended Data Figure 4b-d.

We also performed the suggested experiments with 1,6-Hexanediol using relatively mild conditions. The results, which show sensitivity of spontaneous and APH-induced RPA foci to 1,6-Hexanediol, both in fixed and live cell conditions, have been included as new Extended Data Figure 4i,j.

- Figure S2: How do recovery rates from FRAP experiments relate with other proteins able to phase separate? This information should be included in the text. Also, I could not find an explanation for experiments shown in Figures S2e,f, in the main text.

The half-recovery times that we measured for RPA in FRAP experiments are below 10 seconds and thus in the same range as was measured for other proteins with a high propensity to phase separate, including FUS, hnRNPA1, and DDX4 (as summarized by McSwiggen et al. Genes Dev. 2019, Table 1, PMID: 31594803). We have included this information together with better explanations of the conditions used for FRAP experiments in the main text.

- Figure S2: It is unclear why the authors focus on RPA phosphorylation by DNA-PK and not for example ATM/ATR. A better explanation should be provided.

For the *in vitro* experiments we focused on DNA-PK because of its known ability to antagonize the self-assembly of FUS through phosphorylation of its intrinsically disordered low complexity domain (Han et al. 2012, PMID: 22579282). We included this explanation and the reference to DNA-PK-regulated FUS assembly in the main text.

- Figure 2: Why the authors provide STORM and STED microscopy data on tRPA GFP is unclear. The authors talk about sphericity of the signal, but isn't this always the case in single molecule imaging?

In both approaches we looked at molecule ensembles rather than single molecules, which can be inferred from the heterogenous foci sizes and intensities. The goal of these experiments was merely to obtain microscopic images of RPA ensembles at higher spatial resolution than can be obtained by conventional confocal microscopy. We agree, however, that the use of STORM can be confusing in this context. We therefore removed the STORM images, described the STED imaging in more detail in the methods section, and clarified in the main text that molecule ensembles and not single molecules are visualized.

- Figure 3c suggests that RPA2 self-interact by CoIP. For this conclusion to be drawn, the authors need to show that the interaction is not mediated by ssDNA.

To exclude that the interaction is mediated by ssDNA the cell lysates were treated with benzonase, which degrades both dsDNA and ssDNA. This information is included in the methods section and has been added to the main text. In addition, we now provide an additional co-IP experiment, which shows that the S/T→D mutant, which was previously shown to bind to ssDNA (Binz et al. 2003, PMID: 12819197), has a self-interaction defect compared to WT (new Extended Data Figure 11h).

- Figure 3d: Confirmed phosphorylation sites within the N-IDR are indicated but the specific aminoacids are not shown. This information should be provided here. Are these the same aa mutated in figure s5g? If that is the case at least two sites are missing in 3d.

We apologize for this imprecision; we had indicated only one phosphorylation marker for the three neighboring sites S11/12/13. The corrected Figure 3e now shows specific amino acids and all sites. The amino acids that were mutated are now indicated in Figure 4d,e and the S/T→D mutant is defined in the main text and in the methods section.

- Figure 4f: RPA signals in APH are not just marking replication damage at telomeres. A telomere marker should also be followed to conclude that telomere fusions are affected in S>D mutants.

The analysis of RPA fusions by live cell microscopy requires stable cell lines that express RPA2 WT or RPA2 S/T→D at comparable expression levels and in a manner that they replace the endogenous RPA2 (Extended Data Figure 10). While we did not succeed to generate a second pair of equally well controlled cell lines, which in addition to siRNA-resistant GFP-RPA2 WT and S/T→D also express RFP-TRF2, we did succeed in generating a stable cell line that expresses both GFP-RPA2 and RFP-TRF2. GFP-RPA2 co-localizes with RFP-TRF2 in these cells (new Extended Data Figure 13d), and GFP-RPA2 fusions observed by live cell microscopy coincide with RFP-TRF2 fusions (new Extended Data Figure 13e). These results, together with the quantification that 70%-90% of RPA foci co-localize with telomeres (new Extended Data Figure 9) suggest that the majority of RPA foci fusions represent telomere clustering events.

- Figure 4f legend: How many cells were quantified from the 3 different experiments? How many foci analyzed?

RPA fusions in 100 cells for each of the 3 biological replicates were analyzed over a time-course of 48h. This information is included in the methods section and was added to the figure legend. We also quantified the number of RPA foci that were present at the end of the experiments. On average, the untreated cells had 2 RPA foci per nucleus and the APH-treated ones around 4-5. We provide the data for the reviewers:

Average RPA foci count in live imaging (48h)

Figure R1. GFP-RPA foci counts in the three live cell experiments that were used to quantify GFP-RPA fusion events. U-2 OS cells stably expressing siRNA-resistant GFP-RPA2 WT or S/T→D, depleted for endogenous RPA2, and treated with a low dose of aphidicolin (Aph). Averages and standard deviation from 3 independent 48h time-lapse experiments are indicated. 100 cells per replicate were analyzed.

- Figure 4g: Replicates for this experiment should be provided.

Replicates from three independent experiments are provided in Figure 6i.

- Figure 4: Is the RPAmutS>D mutant still associated with telomeres? Is this association level the same observed in RPA WT?

We quantified the association of RPA WT and S/T→D with telomeres and found a comparable recruitment of RPA to TRF2 foci (Extended Data Figure 11g). However, the self-interaction of RPA was impaired (Extended Data Figure 11h), and RAD52 enrichment at telomeres was reduced (Figure 6g and Extended Data Figure 12g).

- Why in Figure 2d the control with only tRPA does not show droplets formation, as expected from other experiments? Is a different incubation time used here? Please clarify in figure legend and in the figure.

The stimulation of tRPA droplet formation by ssDNA can be overcome by higher PEG-8000 concentrations (Extended Data Figure 5d). The exact PEG-8000 concentrations that were used in all experiments, with and without ssDNA, have been defined in the methods section.

- Fig S6e: Are Rad52 foci only marking telomeres in this experiment? Provide a reference or show that this is the case (e.g, by performing Rad52 IF + telomere DNA FISH).

As requested, we quantified RAD52 co-localization with telomeres. These data are provided in new Extended Data Figure 9.

- Fig. S6d,e Statistical validations are needed.

As requested, we added statistics (Extended Data Figure 12i).

- Why ALT activity has been induced by FANCM depletion in U2OS cells (which are already ALT+) needs a better explanation.

We refer to the relevant publications, which have shown that FANCM depletion increases replication stress at telomeres and ALT activity in U-2 OS cells and provide additional sets of experiments which corroborate this point (comparison of Figure 6 and Extended Data Figure 12).

- In the text the authors mention that among the clustering-induced interaction there are all components of the BLM-TOP3A-RMI (BTR). From the Excel file provided, BLM is found only in the group of common interactors. Which one is correct?

As can be seen in Figure 5d, BLM is enriched after light-induced clustering and has a very low p-value, but in terms of fold change is just below the threshold that we set. Figure 5d is therefore consistent with the Excel table, and the interaction with BLM is nevertheless induced by RPA clustering, although more mildly than the interaction with other components of the BTR complex. We now point out clearly in the figure legend that BLM scored just below the threshold.

Minor Points

- Figure 1b: Why did the authors use FUS as a positive control rather than the mCh-Cry2olig previously described (Kilic et al., 2019)? An explanation in the methods or in the main text should be provided.

Both are valid positive controls, and we have included both in the revised manuscript (Extended Data Figure 1).

- Figures 1b and S1a: Please provide schematic views of the Cry2-mCherry-fusion constructs used in the experiment next to the images, to help understand the experiments.

We provide a clearer scheme of the experiments and the constructs in Figure 1a, and a general scheme of the constructs in Extended Data Figure 1a.

- RPA2 condensates seem to form in the nucleus in Fig1b but in the nucleus+cytoplasm in S1a. What is the common behavior?

In the revised manuscript we now consistently use an NLS-containing version of RPA2, which shows exclusively nuclear localization.

- Figure 1d, Squares or arrowheads pointing to the foci showed in the magnified insets should be added

Arrowheads have been added as suggested.

- Figure 1e: An experimental scheme should be included for a better understanding of how the experiment was conducted.

A scheme has been included and the sequential light activation is now easier to understand.

- Figure 2h: Cy3-ssDNA > Cy3-RPA

The figure has been replaced to avoid confusion.

- Figure S5f,g: these data seem important enough that they should be moved to the main figure.

The mutant analyses has been improved and is now included as main Figure 4d,e and Extended Data Figure 7e.

- Figure 3d legend: 'Consistent with x-ray crystallography data on the RPA tetramerization core' > trimerization

Thank you very much, has been corrected.

- Figure 3e: a Predicted aligned Error (PAE) plot for the structure prediction is shown, but this is not well discussed in the text. Describe this better.

We now explain the PAE plot as prediction error estimate in more detail in the figure legend.

- Figure 3f-g: Axis size for GlobPlot and PONDR graphs should be increased, as they are very difficult to read.

We have increased the axis labels as suggested.

- Figure 3a: a description of the domain (PS, WHD) of RPA subunits should be provided in the legend.

For space reasons and because this schematic did not seem essential for this figure, we removed it from the revised manuscript.

Reviewer #3:

Remarks to the Author:

Replication protein A (RPA), a single-strand DNA (ssDNA) binding complex, is critical for DNA metabolism. RPA binding is required to protect the ssDNAs as well as recognize the ssDNAs in RNA/DNA loop (R loop) during DNA replication, repair, and recombination processes which are essential for the maintenance of genome integrity. In this manuscript, Spegg et al. show that RPA complex can form biomolecular condensates in vivo and in vitro through N-terminal intrinsically disordered region (N-IDR) in RPA2. They demonstrated that RPA condensation involves in ALT telomere synthesis through interaction with FANCM - BLM-TOP3-RMI (BTR) complex, and RAD52 proteins. This work provides not only a new understanding of the regulatory mechanism of RPA proteins but also novel insights of the role of biomolecular condensates in DNA metabolism. Overall, this manuscript has been written very clearly and cited previous work appropriately. However, there are some major points to be addressed.

We thank this reviewer for the time to evaluate our study and were excited to read that the reviewer appreciates the novel insights and how they advance our understanding of the regulation and function of RPA. We were also happy to read that the manuscript had been written very clearly and covers previous work appropriately. Thank you very much for this positive and encouraging feedback.

1. Cell biology experiments (Fig 1, 3, 4); these are very stimulating results; however, they are all based on RPA2 overexpressing cells – either U2OS GFP-RPA2 or U2OS TurboID-RPA2-mCherry-Cry2 cells derived from U2OS Flp-In T-Rex system. Despite of the fact that RPA proteins are highly abundant in cellular level, overexpression-based observations may not fully represent the endogenous-level/cellular physiology. For example, in Fig 3C and S4b, endogenous RPA and GFP-RPA should be represented in same blot, not separately, by using anti-RPA antibody. This could be addressed by generating endogenous level GFP-RPA2 by targeting RPA2 locus using CRISPR.

The reviewer raises a very valid point, and we completely agree that observations based on over-expression may not fully represent cellular physiology. This has been a limitation in many research areas, especially in contexts where protein concentrations have a strong influence on protein behavior, including phase separation.

We have therefore significantly extended our study in this direction, inspired by the comments of the reviewer. First, the Western blots that show RPA2 and GFP-RPA2 came from the same membrane and the levels of RPA2 and GFP-RPA2 can thus be compared directly, using the anti-RPA antibody signal. The uncropped/unseparated blots are now shown in Figure 3d and Extended Data Figure 10a. GFP-RPA2 levels are slightly but not excessively higher than endogenous RPA2 and can efficiently replace endogenous RPA2 after RPA2 knockdown (Extended Data Figure 10a).

Second, we had generated a CRISPR-engineered cell line in which endogenous RPA is fluorescently labeled (Lezaja et al. Nat Commun. 2021). We used this cell line to validate and quantify RPA foci fusion and fission events (new Extended Data Figure 4b-d). And third, we performed experiments in the Cry2 system to verify that even in this system near-endogenous expression levels are sufficient to obtain light-induced RPA2 clustering. This was achieved by carefully comparing RPA2 levels (based on antibody IF staining) in Cry2-mCherry-RPA2 transfected and untransfected cells and showing that overexpression is not needed for optoDroplet formation (new Extended Data Figure 3b), and by additionally depleting endogenous RPA1 (and consequently losing endogenous RPA2 as shown by antibody staining) to verify that even under these conditions Cry2-mCherry-RPA2 at nuclear concentrations very comparable to endogenous still forms optoDroplets (new Extended Data Figure 3c). Although the experimental setup is arguably a bit complicated to explain, we think that these experiments clearly and conclusively demonstrate that the observed clustering is not due to unphysiological overexpression but instead occurs at physiological nuclear protein concentrations.

2. If the formation of RPA condensates is regulated by DNA-PK mediated phosphorylation in N-IDR of RPA2 as shown in RPA (S \rightarrow D) mutant and *in vitro* condensate experiment with DNA-PK-mediated phosphorylation, it would be worth to examine the effect of DNA-PKc inhibitor in RPA-fusion frequency *in vivo* which potentially supports the author's idea.

We thank the reviewer for this suggestion. While *in vitro* the DNA-PK mediated phosphorylations are sufficient to reduce RPA condensation, multiple DDR kinases and CDKs cooperate *in vivo* to phosphorylate the RPA2 N-IDR. At least partially, they are also redundant and can compensate for each other. With combined DDR kinase and CDK1/2 inhibition, however, we observed an increased RPA fusion frequency, consistent with phosphorylation being a negative regulator of RPA fusions. These results have been included as new Extended Data Figure 13c.

3. How RPA condensates involve ALT telomere synthesis? What is the outcome of phosphomimetic/condensation-impaired RPA expression in ALT pathway, such as APB formation, C-circle levels, and G2-synthesis/MiDAS/post-MiDAS at telomeres?

These are great questions and we have significantly extended the analyses of ALT phenotypes based on these suggestions (new Extended Data Figure 11i, new Figure 6 and new Extended Data Figure 12). We observed increased APB formation, higher levels of DNA synthesis at telomeres outside S-phase, elevated levels of ssDNA at telomeres, and increased formation of C-circles in the S/T \rightarrow D cells compared to WT. At the level of RAD52 recruitment, however, consistent with our original observations, we observed an enrichment defect at telomeres. Collectively, these results corroborate an ALT-associated telomere maintenance defect in the phosphomimetic RPA S/T \rightarrow D cells, and suggest a shift towards RAD52-independent BIR associated with increased telomere shortening, as was shown previously by Zhang et al. (PMID: 30673617). We have included these considerations in the discussion.

4. Why RPA2 S/D cells displayed more frequent telomere loss than WT? Is this due to increase in unprotected ssDNA or ssDNA gap at telomeres which are processed/cleaved by nucleases?

Indeed, we see higher levels of ssDNA at telomeres (measured by native telomere FISH performed under non-denaturing conditions) in the S/T→D cells (new Figure 6d and Extended Data Figure 12d). Based on the RAD52 enrichment and RPA fusion defects in the S/T→D cells, we suspect that the balance may be shifted towards RAD52-independent BIR associated with increased telomere shortening, as was shown previously by Zhang et al. (PMID: 30673617). We have included these considerations in the discussion.

5. Page 11 line 236-240: Cannot find the data supporting “RPA condensates drives telomere clustering” and “RPA2 S/D cells displayed elevated telomere fragility”

We have rephrased these sentences. Our revised manuscript more clearly links RPA condensates and RPA fusions to telomeres and telomere clustering in ALT cells (new Extended Data Figure 9, 12, 13 and Figure 6).

6. Fig 4e, GFP-RPA2 and RAD52 colocalization at telomeres (TRF2), is this specific to certain cell cycle phase, such as post-MiDAS/G1 phase?

The triple co-localization of RPA2, RAD52 and TRF2 is technically difficult due to the staining requirements and the reduced GFP-RPA2 signal after methanol fixation required for the RAD52 antibody. While this poses a limitation for a precise determination of co-localization frequencies and how they may change in different cell cycle phases, we can say that co-localization at telomeres can be seen both in G2 and G1, consistent with a role of RPA and RAD52 in both MiDAS and post-MiDAS at telomeres. We provide these data for the reviewers:

Figure R2. RPA and RAD52 co-localization at TRF2-marked telomeres in G1, S and G2. Cells were stained sequentially for the indicated markers (sheep anti-RAD52 followed by mouse anti-RPA2 and rabbit anti-TRF2) and analyzed by QIBC. G1, S-phase, and G2 cells were selected based on DNA content using the nuclear total DAPI intensity.

7. As shown by the author, TurboID-Cry2-RPA2 is very innovative system for the quantitative proximity proteomics. However, TurboID is also useful tool for the quantitative proximity genomics, since it biotinylates proximal nucleotides- DNA, and RNA. Are RPA proteins in condensates less bound to ssDNAs or ALT telomeres as the author discussed (Page12 line 268-269 “RPA condensates provide a reservoir of highly concentrated free RPA”)?

This is an intriguing question, and we spent a considerable amount of time and effort to address it. While we considered a dedicated proximity genomics experiment to map RPA interactions genome-wide without and with light-induced condensation and correlate this with ssDNA at telomeres beyond the scope of this study, we followed two alternative approaches to address condensation-dependent ssDNA binding: One approach that we took to measure ssDNA binding to RPA WT vs S/T→D in cells by FRAP was unfortunately not successful despite many systematic attempts to bring fluorescently labeled ssDNA into the nucleus of RPA WT and S/T→D cells (presumably due to problems with the transfection of fluorescently labeled ssDNA oligos and/or degradation of the transfected ssDNA inside cells). With the second approach, however, we made the interesting observation that trimeric RPA containing the RPA2 S/T→D mutations forms aggregates *in vitro* together with ssDNA, instead of the dynamic liquid droplets formed by wild-type trimeric RPA (new Extended Data Figure 7f). FRAP experiments to analyze the mobility of ssDNA revealed that the ssDNA in RPA S/T→D aggregates was much less mobile compared to the ssDNA in RPA WT droplets (Extended Data Figure 7g,h). We consider these results in agreement with single molecule experiments performed previously by the Greene lab, which had shown that rapid exchange between RPA and ssDNA requires an excess of free RPA (Gibb et al. 2014, PMID: 24498402). This excess of free RPA is provided in RPA condensates, in which RPA molecules are highly concentrated yet can undergo dynamic exchange. We consider the new ssDNA FRAP data a great addition to the manuscript and would like to thank the reviewer for inspiring us to perform these experiments.

Decision Letter, first revision:**Message:** 15th Nov 2022

Dear Matthias,

Thank you again for submitting your revised manuscript "Phase separation properties of RPA combine high-affinity ssDNA binding with dynamic condensate functions at telomeres". I apologize for the delay in responding, which resulted from the difficulty in obtaining suitable referee reports. Nevertheless, we now have comments (below) from the 3 reviewers who had originally evaluated your paper. I hope you will be pleased to see that all reviewers find the manuscript much improved. However, reviewer #2 still has some remaining concerns that we feel should be addressed before we can make a final decision. Thus, we invite you to submit a revised manuscript. Please be sure to address/respond to all concerns in full in a point-by-point response and highlight all changes in the revised manuscript text file. If you have comments that are intended for editors only, please include those in a separate cover letter.

We expect to see your revised manuscript within 6 weeks. If you cannot send it within this time, please contact us to discuss an extension; we would still consider your revision, provided that no similar work has been accepted for publication at NSMB or published elsewhere.

Reporting Summary:

When submitting the revised version of your manuscript, please pay close attention to our <https://www.nature.com/nature-portfolio/editorial-policies/image-integrity>>Digital Image Integrity Guidelines.

Finally, please ensure that you retain unprocessed data and metadata files after publication, ideally archiving data in perpetuity, as these may be requested during the

peer review and production process or after publication if any issues arise.

SOURCE DATA: we urge authors to provide, in tabular form, the data underlying the graphical representations used in figures. This is to further increase transparency in data reporting, as detailed in this editorial

(<http://www.nature.com/nsmb/journal/v22/n10/full/nsmb.3110.html>). Spreadsheets can be submitted in excel format. Only one (1) file per figure is permitted; thus, for multi-paneled figures, the source data for each panel should be clearly labeled in the Excel file; alternately the data can be provided as multiple, clearly labeled sheets in an Excel file. When submitting files, the title field should indicate which figure the source data pertains to. We encourage our authors to provide source data at the revision stage, so that they are part of the peer-review process.

Data availability: this journal strongly supports public availability of data. All data used in accepted papers should be available via a public data repository, or alternatively, as Supplementary Information. If data can only be shared on request, please explain why in your Data Availability Statement, and also in the correspondence with your editor. Please note that for some data types, deposition in a public repository is mandatory - more information on our data deposition policies and available repositories can be found below: <https://www.nature.com/nature-research/editorial-policies/reporting-standards#availability-of-data>

[Redacted]

Kind regards,
Florian

Dr Florian Ullrich
Associate Editor, Nature
Consulting Editor, Nature Structural & Molecular Biology
ORCID 0000-0002-1153-2040

Reviewers' Comments:

Reviewer #1:

Remarks to the Author:

The authors have satisfactorily addressed my initial concerns. The additional experiments have strengthened the manuscript, especially concerning the clarity of experimental approaches and claims related to the impact of protein phase separation on biological function. The new title is also appropriate. This study will enrich our understanding of the increasingly recognized role of protein phase separation in maintaining genome stability.

Reviewer #2:

Remarks to the Author:

In this revised version, Spegg et al. provide a much improved set of experiments supporting the conclusion that RPA2 promotes phase separation, that this property is promoted by ssDNA and is counteracted by phosphorylation of RPA2 in the unstructured N-terminal region. They also provide evidence that RPA2-dependent phase separation is important for telomere maintenance in ALT cells. Together, this is an important study that will be well received in the field. The paper reads well and the conclusions are overall well supported by the results. The data are more clearly presented and information is generally easier to retrieve relative to the first submission. However, the writing style of this paper is still quite minimalist, and this sometimes gets in the way of clarity. Several experiments that need a better explanation are highlighted below. Some new results generated additional concerns, although we remain enthusiastic about this work and we commend the authors for the rigor and the significant effort in the revision.

We believe most of the remaining points can easily be addressed with text changes or relatively straightforward experiments. However, the possibility that RPA-mediated phase separation is required for DSB repair and replication stress responses remains a critical point that should be addressed experimentally before publication.

Main points:

1) A critical point brought up in the previous submission and not addressed in this revision is the extent to which the pathway characterized here is a general property relevant to RPA functions in genome stability or a specific role in ALT biology. This is critical not just from the mechanistic perspective, but also for the relevance of this process to the biology and treatment of cancers. ^{SEP}SEP

Specifically, the authors should provide some basic analysis to address whether the phase separation properties of RPA are important for the role of RPA in DSB repair by HR or in fork processing in the presence of replication stress. Providing a clear statement in support of a general role of RPA-mediated phase separation in ALT or in additional RPA-dependent roles in genome stability is critical, as it affects the interpretation of the data and their applicability to different contexts.

2) In Figure S3b,c, Cry2-mCh-RPA2 foci are formed but only one cell is shown. Quantifications should be provided. Additionally, in these experiments and in Fig 1 and S1, the authors used a NLS-containing version of RPA2 and they correspondently see RPA2 only in the nucleus. Shouldn't RPA2 already have its own NLS? Is this additional NSL potentially increasing the amount of nuclear RPA, skewing the results on this experiment toward cluster formation? This potential caveat should be addressed or at least discussed.

3) The authors show DNA-PK-dependent phase separation (Fig. 4i) of RPA2, but then they inhibit a panel of kinases in Fig S13c. I think they are missing an opportunity to show whether some kinases are more important than others in the observed phenotypes. Could the experiment shown in Fig S13c (or a different assay) be performed with specific kinase inhibitors to clarify which kinases are important for telomere clustering?

Additional points:

4) The authors introduce previous data on SSB and yeast RPA in the discussion, but these should be also clearly stated in the introduction, to provide a more comprehensive context to this work and a fairer acknowledgment of previous work in the field. Introducing yRPA data early on is also necessary for a better understanding of why yRfa2 is characterized in S7a-d.

5) Figure S1a-c: Quantifications show that CtIP and Rad51 do not phase separate. However, these proteins already show droplets before light exposure, which could result from protein over expression, for example. The authors provide a nice analysis of Rpa2 protein level and normalizations supporting the result that Rpa2-induced phase separate structures do not require RPA2 over expression (Fig. S3), as these also form at physiological levels of Rpa2. However, we understand that this was not done for the experiments showed in Figure S1. If this is correct, the caveats of potential protein over expression in these experiments should be clearly discussed, and the possibility that CtIP and Rad51 actually phase separate should be introduced.

6) Figure 2b-c: show in 2b (with arrows or squares) what droplets the cutouts (Fig 2c) refer to. Lines 138-9: RPA droplets displayed gravity flow resulting in surface wetting at plate bottoms: indicate this phenotype in the figure 2b with an arrow for non specialists.

7) Lines 142-3: The difference between 1.6 hexanediol and hydrotrope ATP should be explained. It should also be clarified with literature references that the behaviors described (surface wetting, partial droplet reformation upon remixing, droplet fusions, etc)

are consistent with phase separated structures.

8) Fig S4 i,j: from the quantification, 1.6 Hexanediol doesn't seem to completely dissolve the RPA foci. Do authors have an explanation for this? Do Hexanediol-resistant foci correspond to non-telomeric signals?

9) Fig S4h: I am not sure why the FRAP was done also in response to IR condition in panel h, given that no other experiments in this paper address IR-induced responses. Also (even though interesting), the authors don't provide any discussion of this result in the main text.

10) Fig. S4i: It is unclear why after 24h treatment with APH (long-term replication stress) there are only ~2 RPA foci/cell. What is the biological relevance of such a small count of RPA signals?

11) Fig. S7: yRfa2 is characterized in human cells rather than in yeast cells. The results could be different in yeast where endogenous regulatory mechanisms are present, and this should be addressed or at least this caveat should be clearly discussed in the text.

12) What is shown in Figure S7g is not clear and I am not convinced the experiments shown in the WT vs S/T>D mutants are comparable. RPA WT shows a FRET analysis of a large droplet with a small bleached surface, while in the S>T mutant a small droplet is entirely bleached and less/delayed recovery is shown. Droplets of similar size should be compared in this experiment.

13) The meaning of the analyses shown in Figures 3f-h needs to be better explained in the text for non-experts, and how they differ from each other in the type of information they convey is unclear.

14) Figure 4d and Extended figure 7e > The Y axis scale is different between the main figure 4d and extended figure 7e making difficult to compare these. Figure S7e seems to indicate that different clones with differential expression level for RPA2S/T>D were analyzed, but it is not clear in the text or the figure. Please clarify.

15) It is not necessary to show the protein expression level in 4e, this result can be moved to extended figure. Also, here the expression level is around 80%, differently from the other constructs, where is this variability coming from?

16) Fig. 6: Please provide representative images in the main figure for: i) increased numbers of ALT-associated PML bodies (APBs); ii) more DNA synthesis at telomeres outside S-phase in G2 and in G1; and iii) reduced telomeric Rad52 foci and RPA fusions.

17) Fig. 6a: The result that ALT-associated PML bodies ABPs are increased in RPA S/T>D mutants is interesting and should be better elaborated in the discussion

18) Figure S7f > This is an interesting result and should be moved to the main figure

19) Line 242-245: 'RPA and RAD52 co-localized at telomeres of ALT-positive cancer cells marked by telomere repeat binding factor 2 (TRF2), and their association with telomeres was further enhanced by replication stress or when ALT activity was increased by FANCM depletion', yet Fig S9c and S9g show no increased RPA or Rad52 at telomeres after APH

treatment. The images in Fig S9a-b also do not clearly show such enrichments.

20) Line 263: associated > association

21) Line 269 > even though the kinase inhibitors (ATM/ATR/DNA-PK, CDK1/2) are listed in the figure legend 13, please include this information also in the main text for clarity.

22) Figure S11i is not explained in the main text

23) Fusions shown in Fig S13d-e need to be quantified to show how many tel fusions contain RPA and how many do not.

24) An important value of a figure with a model is that readers can look through it and have a summary at a glance of the main conclusions of the paper and the authors' interpretation of the results. For this purpose, additional text needs to be added to both the figure and the legend. For example, add 'telomere' to the figure or it can still be confused with resected DSBs. Add 'phase separation' next to the yellow circle. Add RPA phosphorylation (not just 'P' to indicate what the modification is acting on. Add the kinase(s) responsible. What the green protein exclusion to the right indicates needs to be specific in the model and in the legend. Why the condensate has a different color in the rightmost model vs the one in the center needs to be explained. Further, in the model, the role of RPA2 phosphorylation in phase separation (not just telomere clustering) should be included by adding the inhibitory sign to the phase separated structure in the middle.

Reviewer #3:

Remarks to the Author:

The authors have adequately addressed all of my concerns. No further questions need to be asked. This is a novel study that is well supported by experimental data. The revised manuscript has been significantly improved and suitable for publication in NSMB.

Author Rebuttal, first revision:

Reviewers' Comments:

Reviewer #1:

Remarks to the Author:

The authors have satisfactorily addressed my initial concerns. The additional experiments have strengthened the manuscript, especially concerning the clarity of experimental approaches and claims related to the impact of protein phase separation on biological function. The new title is also appropriate. This study will enrich our understanding of the increasingly recognized role of protein phase separation in maintaining genome stability.

We were happy to read that the initial concerns were addressed in the revised manuscript and that the reviewer endorses publication.

Reviewer #2:

Remarks to the Author:

In this revised version, Spegg et al. provide a much improved set of experiments supporting the conclusion that RPA2 promotes phase separation, that this property is promoted by ssDNA and is counteracted by phosphorylation of RPA2 in the unstructured N-terminal region. They also provide evidence that RPA2-dependent phase separation is important for telomere maintenance in ALT cells. Together, this is an important study that will be well received in the field. The paper reads well and the conclusions are overall well supported by the results. The data are more clearly presented and information is generally easier to retrieve relative to the first submission. However, the writing style of this paper is still quite minimalist, and this sometimes gets in the way of clarity. Several experiments that need a better explanation are highlighted below. Some new results generated additional concerns, although we remain enthusiastic about this work and we commend the authors for the rigor and the significant effort in the revision.

We believe most of the remaining points can easily be addressed with text changes or relatively straightforward experiments. However, the possibility that RPA-mediated phase separation is required for DSB repair and replication stress responses remains a critical point that should be addressed experimentally before publication.

We were happy to read that the reviewers consider the revised manuscript much improved, acknowledge the rigor of our work, and state that this is an “important study that will be well received in the field” and that “the conclusions are overall well supported by the results”. This is exactly what we were aiming for! We address the remaining comments below.

Main points:

1) A critical point brought up in the previous submission and not addressed in this revision is the extent to which the pathway characterized here is a general property relevant to RPA functions in genome stability or a specific role in ALT biology. This is critical not just from the mechanistic perspective, but also for the relevance of this process to the biology and treatment of cancers.

Specifically, the authors should provide some basic analysis to address whether the phase separation properties of RPA are important for the role of RPA in DSB repair by HR or in fork processing in the presence of replication stress. Providing a clear statement in support of a general role of RPA-mediated phase separation in ALT or in additional RPA-dependent roles in genome stability is critical, as it affects the interpretation of the data and their applicability to different contexts.

We appreciate the interest in a potential role of the phase separation properties of RPA beyond telomere maintenance by ALT, and we agree that such a role in different cellular and genomic contexts is possible and perhaps likely and could even be relevant for the treatment of cancers that do not rely on ALT.

Assessing the functional role of RPA phase separation in other cellular contexts in a conclusive manner (which would be needed for any “clear statement in support of a general role of RPA-mediated phase separation”) is far from trivial, unfortunately, and a “basic analysis” to address whether (or not) “the phase separation properties of RPA are important for the role of RPA in DSB repair by HR or in fork processing in the presence of replication stress”, in the form of a simple test, does not exist, as far as we know. We base our analysis of RPA functions at ALT telomeres and ALT-associated PML bodies on multiple independent lines of evidence, including the defective *in vitro* and *in vivo* clustering of the ssDNA-binding proficient RPA S/T→D mutant, the enrichment of the ALT factor RAD52 into RPA droplets and nuclear foci, and the reduced fusion frequency of RPA- and TRF2-marked telomeres. Assessing the functional contribution of phase separation properties of RPA in other cellular contexts and reaching a level where a clear statement on this would be justified are, if one wants to do this thoroughly, full projects on their own and beyond the scope of this study.

However, certain experimental observations do in fact indicate that RPA condensation around DSBs and stalled/broken replication forks is impaired in RPA S/T→D mutant cells. For instance, the accumulated intensity of RPA in nuclear foci at these sites (stressed replication forks induced by a low dose of CPT; DSBs induced by IR) is lower in RPA S/T→D cells compared to WT (newly included Extended Data Figures 11h and i), suggesting that although RPA recruitment is not abrogated in RPA S/T→D cells, consistent with the mutant being proficient in ssDNA binding, its condensation is impaired. We have included these additional data as the reviewers suggested, because together with the FRAP results on RPA dynamics at ATRi-induced stressed replication forks and at IR-induced DSBs (Extended Data Figures 4g and h), they do provide additional support beyond the context of ALT telomere maintenance. The results are discussed on page 7 lines 134-135, page 13 lines 270-281, and in the extended discussion on page 17 lines 381-389.

2) In Figure S3b,c, Cry2-mCh-RPA2 foci are formed but only one cell is shown. Quantifications should be provided. Additionally, in these experiments and in Fig 1 and S1, the authors used a NLS-containing version of RPA2 and they correspondently see RPA2 only in the nucleus. Shouldn't RPA2 already have its own NLS? Is this additional NSL potentially increasing the amount of nuclear RPA, skewing the results on this experiment toward cluster formation? This potential caveat should be addressed or at least discussed.

We demonstrate in Figure S3b,c that RPA can form light-induced clusters in cells that do not show RPA over-expression, to reject the hypothesis that the observed clustering is an over-expression artefact. Nevertheless, we reanalyzed the data as suggested to test if the mCherry-positive cells make more RPA clusters than the mCherry-negative ones and included the additional quantifications in the revised figures (Extended Data Figures 3b and c).

Import of trimeric RPA is independent of importin alpha and of a classical NLS (Jullien et al. EMBO J. 1999 Aug 2;18(15):4348-58). When RPA2 is expressed as a fusion with Cry2-mCherry, it needs an NLS for nuclear import. In our first submission, we had included results obtained without NLS, where RPA2 as fusion with Cry2-mCherry readily formed optoDroplets in the cytoplasm (previous Extended Data Figure 1a). Endogenous RPA and ectopic GFP-tagged RPA are exclusively nuclear (Extended Data Figure 4a,b,f,g,h,j, 9a,b, 14a,d,e), indicating that RPA condensation behavior is best studied in the nucleus, as the reviewer rightly pointed out previously. Importantly, we verified that Cry2-mCherry-RPA2 with an NLS forms clusters in conditions where it is expressed at endogenous levels, with the endogenous RPA2 being depleted (Extended Data Fig. 3b,c), excluding potential caveats associated with enforced nuclear localization.

3) The authors show DNA-PK-dependent phase separation (Fig. 4i) of RPA2, but then they inhibit a panel of kinases in Fig S13c. I think they are missing an opportunity to show whether some kinases are more important than others in the observed phenotypes. Could the experiment shown in Fig S13c (or a different assay) be performed with specific kinase inhibitors to clarify which kinases are important for telomere clustering?

The DDR kinases ATM, ATR, and DNA-PK have partly redundant substrates and overlapping functions. This is well documented in the literature (reviewed for instance in Blackford & Jackson, ATM, ATR, and DNA-PK: The Trinity at the Heart of the DNA Damage Response, Mol Cell. 2017 Jun 15;66(6):801-817) and mentioned in the manuscript. Inhibition of individual kinases by inhibitors did not increase the fusion frequency in our experiments, probably due to their redundancy and overlapping functions. This is a negative result, however, and as such not very informative. Moreover, all our data with the phosphomimetic mutants suggest that multiple phosphosites, which are targeted by the DDR kinases and CDKs, functionally cooperate (Figure 4d and new Extended Data Figures 7f and g, which we added to include the previously missing single S4D mutant and thereby complement the panel of mutants). Among those, S4, S8, S12, and T21 can be targeted by both DNA-PK and ATM, whereas S33 is targeted by ATR and S23 and S29 are targeted by CDK1 and CDK2 (Oakley & Patrick. Front Biosci (Landmark Ed). 2010 Jun 1;15(3):883-900). Since neither of these sites individually nor in double, triple, or quadruple combinations affected RPA2 clustering in a significant manner, targeting either of the three DDR kinases individually would not be expected to affect clustering.

Additional points:

4) The authors introduce previous data on SSB and yeast RPA in the discussion, but these should be also clearly stated in the introduction, to provide a more comprehensive context to this work and a fairer acknowledgment of previous work in the field. Introducing yRPA

data early on is also necessary for a better understanding of why yRfa2 is characterized in S7a-d.

As requested, we included bacterial SSB and yeast RPA already in the introduction (page 4, lines 60-65): *“In bacteria, the essential ssDNA binding protein SSB was recently reported to phase separate in vitro and in bacterial extracts¹⁴. In yeast, however, the ssDNA binding protein Rfa1 showed less dynamic behavior and tighter ssDNA binding compared to Rad52, for which condensation behavior consistent with phase separation had been proposed^{15,16}. Here, we reveal that mammalian RPA ...”*.

5) Figure S1a-c: Quantifications show that CtIP and Rad51 do not phase separate. However, these proteins already show droplets before light exposure, which could result from protein over expression, for example. The authors provide a nice analysis of Rpa2 protein level and normalizations supporting the result that Rpa2-induced phase separate structures do not require RPA2 over expression (Fig. S3), as these also form at physiological levels of Rpa2. However, we understand that this was not done for the experiments showed in Figure S1. If this is correct, the caveats of potential protein over expression in these experiments should be clearly discussed, and the possibility that CtIP and Rad51 actually phase separate should be introduced.

We agree and mention this limitation in the main text (page 5, lines 84-93): *“Other DNA damage response (DDR) factors showed weaker or no light-induced clustering when over-expressed as Cry2-mCherry fusions, and neither expression of the Cry2-mCherry module alone (Empty), nor as a fusion with dimerization-prone glutathione S-transferase (GST) resulted in optoDroplet formation (Extended Data Figure 1a-c). We conclude that light-induced seeding requires self-assembly-driven amplification of protein condensation to cause discernable optoDroplet formation, and that RPA2 carries these features. Although other DDR factors did not show these features in the optoDroplet system, we do not exclude the possibility of their dynamic clustering at endogenous expression levels and in other cellular contexts”*.

6) Figure 2b-c: show in 2b (with arrows or squares) what droplets the cutouts (Fig 2c) refer to. Lines 138-9: RPA droplets displayed gravity flow resulting in surface wetting at plate bottoms: indicate this phenotype in the figure 2b with an arrow for non specialists.

The droplet fusion shown in Figure 2c is not part of the images shown in Figure 2b. We clarified this in the figure legend. The droplet formation and surface wetting on the plate bottom is now indicated more clearly by dashed lines and circles, explained in the figure legend: *“For illustration purposes, one RPA droplet per image is marked by an orange dotted circle, and examples of surface wetting at the plate bottom are marked by a red dotted curved line”*.

7) Lines 142-3: The difference between 1.6 hexanediol and hydrotrope ATP should be explained. It should also be clarified with literature references that the behaviors described (surface wetting, partial droplet reformation upon remixing, droplet fusions, etc) are consistent with phase separated structures.

As requested, we provide additional explanations and references. We write: “RPA foci were sensitive to the aliphatic alcohol 1,6-hexanediol, which interferes with weak hydrophobic interactions (Extended Data Figure 4i,j)” (page 7, lines 137-1140) and “RPA droplets formed *in vitro* were dissolved by 1,6-hexanediol (Extended Data Figure 5f) and by millimolar concentrations of ATP, which was previously reported to act as biological hydrotrope that solubilizes aggregation-prone proteins in aqueous solutions²⁸ (Extended Data Figure 5g)”. We also provide additional references on surface tension, droplet fusions, and surface wetting behavior in the context of phase separation (page 8, line 148-155).

8) Fig S4 i,j: from the quantification, 1.6 Hexanediol doesn't seem to completely dissolve the RPA foci. Do authors have an explanation for this? Do Hexanediol-resistant foci correspond to non-telomeric signals?

As Hexanediol treatment of cells is toxic and has pleiotropic effects, we limited the treatment to a short period of only 5 minutes and used relatively low concentrations (1.25-5%). We stated this more clearly in the manuscript (page 7, line 138). Low levels of residual foci are likely linked to this. The strong inhibitory effect of the treatment is clear, however, both from the quantification of fixed cell populations and also from the images obtained by live cell imaging.

9) Fig S4h: I am not sure why the FRAP was done also in response to IR condition in panel h, given that no other experiments in this paper address IR-induced responses. Also (even though interesting), the authors don't provide any discussion of this result in the main text.

Together with the new results on RPA accumulation at CPT- and IR-induced DNA lesions (new Extended Data Figure 11h and i) and the new paragraph in the discussion on RPA condensation functions beyond ALT (page 17), we feel that the FRAP results on stressed replication forks and IR-induced DSBs can remain included. Additional description and discussion are provided in the main text (page 7 lines 130-140, page 13 lines 270-281, page 17 lines 381-389).

10) Fig. S4i: It is unclear why after 24h treatment with APH (long-term replication stress) there are only ~2 RPA foci/cell. What is the biological relevance of such a small count of RPA signals?

RPA foci under these conditions form stochastically at fragile genomic regions and their numbers are heterogenous due to asynchronous cell cycle progression. These conditions have been established as a general standard in the field to induce low levels of physiological replication stress, which results in fragility of common fragile sites (CFSs), frequent breakpoints in cancer. From the many publications that use similarly low doses of APH (0.2-0.4 μ M), some of the more recent ones include:

Bhowmick et al. *Mol Cell*. 2022 Sep 15;82(18):3366-3381.e9.

Groelly et al. *Mol Cell*. 2022 Sep 15;82(18):3382-3397.e7.

Lezaja et al. *Nature Comm*. 2021 Jun 22;12(1):3827.

Macheret et al. *Cell Res*. 2020 Nov;30(11):997-1008.

Blin et al. *Nat Struct Mol Biol*. 2019 Jan;26(1):58-66.

Minocherhomji et al. *Nature*. 2015 Dec 10;528(7581):286-90.

11) Fig. S7: yRfa2 is characterized in human cells rather than in yeast cells. The results could be different in yeast where endogenous regulatory mechanisms are present, and this should be addressed or at least this caveat should be clearly discussed in the text.

We agree that the situation could be different in yeast and pointed out this caveat in the text: *“Consistently, yeast Rfa2 (corresponding to human RPA2) failed to form optoDroplets in our experiments, although it remains possible that this would be different in a yeast cell environment”* (page 16, lines 345-350).

12) What is shown in Figure S7g is not clear and I am not convinced the experiments shown in the WT vs S/T>D mutants are comparable. RPA WT shows a FRET analysis of a large droplet with a small bleached surface, while in the S>T mutant a small droplet is entirely bleached and less/delayed recovery is shown. Droplets of similar size should be compared in this experiment.

These results show FRAP analyses, and identical bleach areas have been used to analyze RPA WT and S/T→D. The comparison is as good as it gets, considering that RPA WT forms spherical droplets whereas RPA S/T→D forms amorphous aggregates of varying size and dimensions. The droplets and aggregates were chosen to be of similar overall size, see also Main Figure 4i and the response to point 18.

13) The meaning of the analyses shown in Figures 3f-h needs to be better explained in the text for non-experts, and how they differ from each other in the type of information they convey is unclear.

As suggested, we provided additional explanations and references in the main text and in the figure legends: *“AlphaFold predicted the N-IDR to be flexible and unconstrained by the two globular domains of RPA2 (Figure 3e), with a high expected position error for the first 40-45 amino acids and a low expected position error for the two globular domains (Figure 3f). Two sequence-based online prediction tools for protein folding and disorder, GlobPlot³⁰ and PONDR (Predictor of Natural Disordered Regions)³¹, agreed on the N-IDR being unstructured (Figure 3g,h)”* (page 10 lines 195-200) and *“(g) RPA2 protein disorder prediction by GlobPlot 2.3, a tool plotting the tendency within query proteins for order/globularity and disorder³⁰. Disorder propensity on the y-axis, amino acid position of RPA2 on the x-axis. (h) RPA2 protein disorder prediction by PONDR (Predictor of Natural Disordered Regions)³¹. PONDR score on the y-axis, amino acid position of RPA2 on the x-axis. Highly disordered regions are marked by the black bar”* (page 46 lines 1065-1070).

14) Figure 4d and Extended figure 7e > The Y axis scale is different between the main figure 4d and extended figure 7e making difficult to compare these. Figure S7e seems to indicate that different clones with differential expression level for RPA2S/T>D were analyzed, but it is not clear in the text or the figure. Please clarify.

The mutants analyzed in this figure are described in detail in the legend: *“Quantification of Cry2-mCherry-RPA2 optoDroplet formation of wildtype, single (S8D, S11D, S12D, T21D, S23D, S33D), double (S4/S8D, S23/S33D, S8/S33D, T21D/S33D), triple (S11/S12/S13D,*

T21D/S29/S33D, S8D/T21D/S33D), quadruple (T21D/S23/S29/S33D), and octuple S/T → D (S8/S11/S12/S13/T21/S23/S29/S33D) mutants. Cry2-mCherry expression was grouped from low to high. Single data points represent individual Cry2-mCherry-RPA2 mutants". Fluorescence intensity values from microscopy images are in arbitrary units, and differences between experiments reflect differences in expression levels, exposure times, and microscopes. Please see also next point.

15) It is not necessary to show the protein expression level in 4e, this result can be moved to extended figure. Also, here the expression level is around 80%, differently from the other constructs, where is this variability coming from?

We moved the results previously shown in Figure 4e to new Extended Data Figure 7e, as suggested by the reviewer. The optoDroplet experiments with the panel of RPA mutants were performed by transient plasmid transfections, which explains the inter-experiment variability in expression and intensity measurements in arbitrary units. Of note, values depict intensity measurements, not normalized percentages: Within each experiment, expression levels were quantified and cells with nearly identical expression levels were analyzed. This is possible due to the high-content imaging, image analysis and cell population gating pipeline that we use, described in the Methods. Measured differences in optoDroplet formation are therefore independent of variations in expression and reflect the intrinsic propensity of the expressed protein to form light-induced condensates.

16) Fig. 6: Please provide representative images in the main figure for: i) increased numbers of ALT-associated PML bodies (APBs); ii) more DNA synthesis at telomeres outside S-phase in G2 and in G1; and iii) reduced telomeric Rad52 foci and RPA fusions.

As suggested, we included representative single cell images for APBs (defined as PML-TRF2 co-localizations), EdU incorporation at telomeres in G2 and G1, and RAD52 at telomeres (new Extended Data Figure 12a-d). It is not possible to provide a representative image of a decreased fusion *frequency* (Figure 6h). How fusion frequencies were determined is demonstrated in Extended Data Figure 14, together with representative example images from time-lapse experiments, and described in the text and methods. The newly added quantification of RPA and TRF2 fusions (new Extended Data Figure 14e) strongly suggests that RPA fusions represent telomere clustering events.

17) Fig. 6a: The result that ALT-associated PML bodies ABPs are increased in RPA S/T>D mutants is interesting and should be better elaborated in the discussion

In the discussion we write: *"We found that RAD52 readily partitions into phase separated RPA droplets in vitro, and that RAD52 enrichment at sites of ALT activation is reduced in cancer cells expressing phase separation-impaired, phospho-mimetic RPA2. Rather than loss of APB formation, RPA S/T → D cells with reduced RAD52 recruitment to telomeres showed elevated C-circles, consistent with a shift towards RAD52-independent ALT associated with C-circle formation and progressive telomere shortening. In line, we observed more frequent telomere loss in cells expressing phospho-mimetic RPA2. Considering that we also observed signs of impaired telomere clustering, homology search and donor template usage might be altered by RPA hyper-phosphorylation, e.g., towards more intratelomeric recombination.*

Further studies employing dedicated assays to interrogate ALT sub-pathway usage will be needed to investigate this hypothesis". As the manuscript with its 6 Main and now 15 Extended Data Figures is quite extensive already, and to avoid too far-reaching speculations on the role of APB for different ALT-associated telomere repair pathways in mammalian cells, we would prefer to keep this section in its current form. In case we may have missed to cite and discuss relevant related studies, we would of course be happy to include them.

18) Figure S7f > This is an interesting result and should be moved to the main figure

As suggested, we moved this result to main Figure 4i. From these images, the overall size distribution of droplets and amorphous aggregates can be appreciated (see also point 12).

19) Line 242-245: 'RPA and RAD52 co-localized at telomeres of ALT-positive cancer cells marked by telomere repeat binding factor 2 (TRF2), and their association with telomeres was further enhanced by replication stress or when ALT activity was increased by FANCM depletion', yet Fig S9c and S9g show no increased RPA or Rad52 at telomeres after APH treatment. The images in Fig S9a-b also do not clearly show such enrichments.

Fig. S9c and S9g depict percentages rather than foci counts. That more telomeres are bound by RPA and RAD52 after replication stress or FANCM depletion is evident from Fig S9e,f and S9i,j. The images in S9a,b also show more RPA and RAD52 foci after APH and siFANCM. That the percentage of RPA and RAD52 foci at telomeres after APH does not increase (S9c,g), despite the fact that more RPA and RAD52 foci form and that more telomeres are positive for these two proteins, indicates that APH not only causes replication stress at telomeres, but also at other genomic sites (which is expected and in line with the previous literature).

20) Line 263: associated> association

Has been corrected, thank you for spotting this mistake.

21) Line 269> even though the kinase inhibitors (ATM/ATR/DNA-PK, CDK1/2) are listed in the figure legend 13, please include this information also in the main text for clarity.

Has been included in the main text as suggested (page 13 lines 293-294).

22) Figure S11i is not explained in the main text

This Figure panel has been moved to new Extended Data Figure 12e. The figure shows a control for the C-circle analysis, using ALT-positive versus ALT-negative cells as reference. The Figure is referred to in the main text (page 13 line 281) and the newly extended figure legends provides the necessary information: *"Relative C-circle levels in ALT-positive U-2 OS cells compared to ALT-negative HeLa cells as control for the C-circle assay. C-circle levels were determined by qPCR in triplicates and normalized to the average C-circle level of U-2 OS cells. Two-tailed unpaired t-test, ** $p \leq 0.01$ "* (page 58 lines 1356-1357).

23) Fusions shown in Fig S13d-e need to be quantified to show how many tel fusions contain RPA and how many do not.

As requested, we quantified the fusions and added the results to Extended Data Figure 14e. The results confirm the tight association between RPA and ALT-associated telomere clustering.

24) An important value of a figure with a model is that readers can look through it and have a summary at a glance of the main conclusions of the paper and the authors' interpretation of the results. For this purpose, additional text needs to be added to both the figure and the legend. For example, add 'telomere' to the figure or it can still be confused with resected DSBs. Add 'phase separation' next to the yellow circle. Add RPA phosphorylation (not just 'P' to indicate what the modification is acting on. Add the kinase(s) responsible. What the green protein exclusion to the right indicates needs to be specific in the model and in the legend. Why the condensate has a different color in the rightmost model vs the one in the center needs to be explained. Further, in the model, the role of RPA2 phosphorylation in phase separation (not just telomere clustering) should be included by adding the inhibitory sign to the phase separated structure in the middle.

As suggested, we added additional text to the model figure and the legend to facilitate interpretation. Specifically, ssDNA at telomers is now explicitly shown, RPA condensation and RPA-RAD52 co-condensation are more clearly depicted using a unified color code, the undefined exclusion of proteins from these condensates has been omitted, and the multisite phosphorylation of RPA and its role have been clarified. Together, these changes make the model figure clearer and more concise and the key messages easier to interpret at a glance. We would like to thank the reviewers for these helpful suggestions, and for the time to review our manuscript so thoroughly and help enhance its clarity.

Reviewer #3:

Remarks to the Author:

The authors have adequately addressed all of my concerns. No further questions need to be asked. This is a novel study that is well supported by experimental data. The revised manuscript has been significantly improved and suitable for publication in NSMB.

We were happy to read that all concerns were addressed in the revised manuscript and that the reviewer recommends publication in NSMB.

Decision Letter, second revision:

Message: Our ref: NSMB-A45955B

15th Dec 2022

Dear Dr. Altmeyer,

Thank you for submitting your revised manuscript "Phase separation properties of RPA combine high-affinity ssDNA binding with dynamic condensate functions at telomeres" (NSMB-A45955B). It has now been seen by the remaining, original referee and her/his comments are below. The reviewer finds that all the issues raised have been addressed, and therefore we are ready to accept in principle to publish it in Nature Structural & Molecular Biology, pending minor revisions and compliance with our editorial and formatting guidelines.

To facilitate our work at this stage, we would appreciate if you could send us the main text as a word file. Please make sure to copy the NSMB account (cc'ed above).

Sincerely,

Dimitris Typas
Associate Editor
Nature Structural & Molecular Biology
ORCID: 0000-0002-8737-1319

Reviewer #2 (Remarks to the Author):

The authors did a commendable job addressing all the points raised during the revision. New quantifications and text changes significantly improved the rigor and readability of the manuscript.

This study provides new important insights on the role and regulation of RPA -mediated phase separation, significantly extending our understanding of the contribution of phase separation to genome stability. I fully endorse its publication in NSMB.

Final Decision Letter:**Message** 27th Jan 2023

:

Dear Dr. Altmeyer,

We are now happy to accept your revised paper "Phase separation properties of RPA combine high-affinity ssDNA binding with dynamic condensate functions at telomeres" for publication as an Article in Nature Structural & Molecular Biology.

As soon as your article is published, you can generate your shareable link by entering the DOI of your article here: http://authors.springernature.com/share. Corresponding authors will also receive an automated email with the shareable link

Your paper will be published online soon after we receive proof corrections and will appear

in print in the next available issue. You can find out your date of online publication by contacting the production team shortly after sending your proof corrections. Content is published online weekly on Mondays and Thursdays, and the embargo is set at 16:00 London time (GMT)/11:00 am US Eastern time (EST) on the day of publication. Now is the time to inform your Public Relations or Press Office about your paper, as they might be interested in promoting its publication. This will allow them time to prepare an accurate and satisfactory press release. Include your manuscript tracking number (NSMB-A45955C) and our journal name, which they will need when they contact our press office.

About one week before your paper is published online, we shall be distributing a press release to news organizations worldwide, which may very well include details of your work. We are happy for your institution or funding agency to prepare its own press release, but it must mention the embargo date and Nature Structural & Molecular Biology. If you or your Press Office have any enquiries in the meantime, please contact press@nature.com.

Please note that *Nature Structural & Molecular Biology* is a Transformative Journal (TJ). Authors may publish their research with us through the traditional subscription access route or make their paper immediately open access through payment of an article-processing charge (APC). Authors will not be required to make a final decision about access to their article until it has been accepted. <https://www.springernature.com/gp/open-research/transformative-journals> Find out more about Transformative Journals

Authors may need to take specific actions to achieve [compliance](https://www.springernature.com/gp/open-research/funding/policy-compliance-faqs) with funder and institutional open access mandates. If your research is supported by a funder that requires immediate open access (e.g. according to [Plan S principles](https://www.springernature.com/gp/open-research/plan-s-compliance)) then you should select the gold OA route, and we will

direct you to the compliant route where possible. For authors selecting the subscription publication route, the journal's standard licensing terms will need to be accepted, including [self-archiving policies](https://www.springernature.com/gp/open-research/policies/journal-policies). Those licensing terms will supersede any other terms that the author or any third party may assert apply to any version of the manuscript.

Sincerely,

Dimitris Typas
Associate Editor
Nature Structural & Molecular Biology
ORCID: 0000-0002-8737-1319
